# When is Multicalibration Post-Processing Necessary?

**Dutch Hansen**[*]
University of Southern California
jmhansen@usc.edu

**Siddartha Devic**[*]
University of Southern California
devic@usc.edu

**Preetum Nakkiran**
Apple
preetum@nakkiran.org

**Vatsal Sharan**
University of Southern California
vsharan@usc.edu

## Abstract

Calibration is a well-studied property of predictors which guarantees meaningful uncertainty estimates. Multicalibration is a related notion — originating in algorithmic fairness — which requires predictors to be simultaneously calibrated over a potentially complex and overlapping collection of protected subpopulations (such as groups defined by ethnicity, race, or income). We conduct the first comprehensive study evaluating the usefulness of multicalibration post-processing across a broad set of tabular, image, and language datasets for models spanning from simple decision trees to 90 million parameter fine-tuned LLMs. Our findings can be summarized as follows: (1) models which are calibrated out of the box tend to be relatively multicalibrated without any additional post-processing; (2) multicalibration post-processing can help inherently uncalibrated models and also large vision and language models; and (3) traditional calibration measures may sometimes provide multicalibration implicitly. More generally, we distill many independent observations which may be useful for practical and effective applications of multicalibration post-processing in real-world contexts. We also release a python package implementing multicalibration algorithms, available via 'pip install multicalibration'.

## 1 Introduction

A popular approach to ensuring that probabilistic predictions from machine learning algorithms are *meaningful* is model calibration. Intuitively, calibration requires that amongst all samples given score $p \in [0, 1]$ by an ML algorithm, exactly a $p$-fraction of those samples have positive label. Calibration ensures that a predictor has an accurate estimate of its own predictive uncertainty, and is a fundamental requirement in applications where probabilities may be taken into account for high-stake decisions such as disease diagnosis (Dahabreh et al., 2017) or credit/lending decisions (Bequé et al., 2017). Miscalibration can result in undesirable downstream consequences when probabilistic predictions are *thresholded* into decisions: if a predictor has high calibration error in disease diagnosis, for example, the individuals assigned lower predicted probabilities may be unfairly denied treatment. Calibration has a long history in the machine learning community (Guo et al., 2017; Minderer et al., 2021; Niculescu-Mizil and Caruana, 2005; Platt et al., 1999), but was arguably first introduced in *fairness* contexts by Cleary (1968). More recently, it has appeared in the algorithmic fairness community via the seminal works of Chouldechova (2017); Kleinberg et al. (2017).

Although calibration ensures meaningful uncertainty estimates aggregated over the entire population, it does *not* preclude potential discrimination at the level of *groups* of individuals: a model may

38th Conference on Neural Information Processing Systems (NeurIPS 2024).

---

* Equal contribution.

be well calibrated overall but systematically underestimate the risk or qualification probability on historically underrepresented subsets of individuals. For example, Obermeyer et al. (2019) show differing calibration error rates across groups defined by race for prediction in high-risk patient care management systems. As pointed out by Obermeyer et al. (2019), in the downstream task of patient intervention based on thresholds over probabilistic predictions, this can inadvertently lead to differing rates of healthcare access based on group membership.

To combat these issues, the notion of *multicalibration* was proposed as a refinement of standard calibration (Hébert-Johnson et al., 2018). Multicalibration requires that a model be simultaneously calibrated on an entire collection of (efficiently) identifiable and potentially overlapping subgroups of the data distribution. A plethora of recent theoretical work has studied and utilized multicalibration to obtain interesting and important guarantees in algorithmic fairness (Bastani et al., 2022; Devic et al., 2024; Dwork et al., 2021; Gopalan et al., 2022b,c; Jung et al., 2021; Shabat et al., 2020), learning theory (Gollakota et al., 2024; Gopalan et al., 2023, 2022a), and cryptography (Dwork et al., 2023). Desirable consequences of multicalibrated predictors abound: multicalibration can provide provable guarantees on the transferability of a model's predictions to different loss functions (omniprediction, Gopalan et al. (2022a)), the ability of a model to do meaningful conformal prediction (Jung et al., 2023), and universal adaptability or domain adaptation (Kim et al., 2022).

Although there is a host of theoretical results surrounding multicalibration and related notions, there is little systematic empirical study of the latent multicalibration error of popular machine learning models, the effectiveness of multicalibration post-processing algorithms, or even best practices for practitioners who wish to apply ideas and algorithms from the multicalibration literature. In particular, theoretical results are often concerned with multicalibration towards subgroups defined by potentially *infinite* hypothesis classes (Haghtalab et al., 2023; Hébert-Johnson et al., 2018). In contrast, fairness practitioners may prioritize the equitable performance of a model over a *finite* number of protected subgroups of interest. These groups are typically defined by attributes and meta-data such as race, sex, ethnicity, etc. (Chen et al., 2023) which are normatively deemed as important. Furthermore, most existing works applying multicalibration in practical settings only focus on one-off datasets or examples, and do not validate the algorithm(s) across a variety of datasets and models or with realistic finite sample restrictions (Barda et al., 2020; La Cava et al., 2023; Liu et al., 2019).

To address these, we consider a "realistic" setup where a practitioner only has a finite amount of data, and must choose how to partition this data between *learning* and *post-processing* in order to achieve a suitable accuracy and multicalibration error rate over a finite set of subgroups. This allows us to investigate many important questions pertaining to the practical usage of multicalibration concepts and algorithms, which, to the best of our knowledge, have not been *systematically* considered by the theoretical or practical communities. For example, we use this setup to investigate the effectiveness of multicalibration post-processing algorithms and hyperparameter choices, as well as the latent multicalibration properties of popular machine learning models at a large scale.

More broadly, we initiate a systematic empirical study of multicalibration with the goal of answering two salient questions:

**Question 1.** In practice, how often and for what machine learning models is multicalibration an expected consequence of empirical risk minimization?

**Question 2.** Conversely, when must additional steps be taken to multicalibrate models, how difficult is this to do in practice, and what steps can be taken to make this easier?

The conventional wisdom is that multicalibration is something that is not naturally achieved by ML algorithms—this is precisely why many in the community have focused on creating post-processing algorithms which do achieve it (see, e.g., Gopalan et al. (2022b); Hébert-Johnson et al. (2018), and Section 1.2). However, recent theoretical results suggest that multicalibration may in fact be an *inevitable consequence* of certain empirical risk minimization (ERM) methods with proper losses (Błasiok et al., 2023; Liu et al., 2019). This apparent conflict between conventional wisdom and recent results has not been tested in practice. We propose studying Question 1 since we believe that the current state of multicalibration in ML models should be systematically studied to better understand the implications for modern learning setups involving large models and fine-tuning. Question 2 is complementary and focused on investigating the effectiveness of current multicalibration algorithms on real datasets and illuminating challenges which can guide the development of future algorithms. A

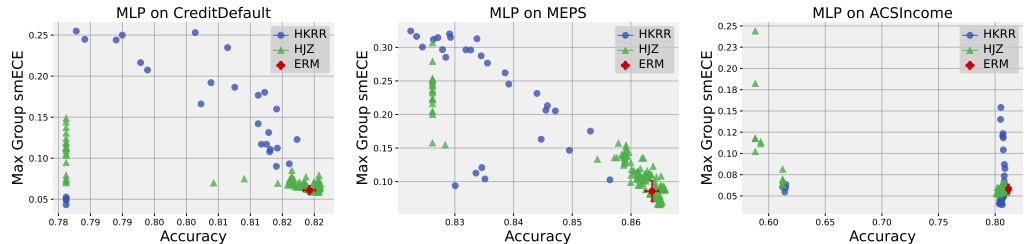

Figure 1: Test accuracy vs. maximum group-wise calibration error (smECE) averaged over five train/validation splits for simple neural networks (MLPs) trained on Credit Default, MEPS, and ACS Income. Each point corresponds to the performance of the multicalibration post-processing algorithm HKRR (Hébert-Johnson et al., 2018) or HJZ (Haghtalab et al., 2023) with a different choice of hyperparameters. Standard empirical risk minimization (ERM) for MLPs achieves nearly optimal accuracy and multicalibration error. Similar plots for each dataset are in Appendix H.

partial answer to one or both of these questions could help practitioners concerned about fairness understand when they should or should not expect multicalibration algorithms to help.

## 1.1 Our Contributions

We conduct a large-scale evaluation of multicalibration methods, comparing three families of methods: (1) standard ERM, (2) ERM followed by a classical recalibration method (e.g. Platt scaling), and (3) ERM followed by an explicit multicalibration algorithm (e.g. that of Hébert-Johnson et al. (2018)).

We find that in practice, this comparison is surprisingly subtle: multicalibration algorithms do not always improve worst group calibration error (relative to the ERM baseline), for example. From the results of our extensive experiments on tabular, vision, and language tasks (involving running multicalibration algorithms more than 45K times), we extract a number of observations clarifying the utility of multicalibration algorithms. Most significantly, we find:

1. ERM alone is often a strong baseline, and can often be remarkably multicalibrated without further post-processing. In particular, on tabular datasets, multicalibration post-processing does not improve upon worst group calibration error of ERM for simple NNs.

2. Multicalibration algorithms are very sensitive to hyperparameter choices, and can require large parameter sweeps to avoid overfitting. Furthermore, these algorithms tend to be most effective in regimes with large amounts of available data, such as image and language datasets.

3. Traditional calibration methods such as Platt scaling or isotonic regression can sometimes give nearly the same performance as multicalibration algorithms, and are hyperparameter-free. Furthermore, compared to multicalibration post-processing, they are extremely computationally efficient.

We also present numerous practical takeaways for users of multicalibration algorithms, which are not apparent from the existing theoretical literature, but are crucial considerations in practice. We believe that our investigations will not only broaden the practical applicability of multicalibration as a concept and algorithm, but also provide valuable information to the theoretical community as to what barriers multicalibration faces in practice. To both of these ends, all code used in our experiments is publicly accessible, and we also release a python package implementing two multicalibration algorithms which we make available via 'pip install multicalibration'. [1]

**Organization.** In Section 1.2, we begin with a brief review of related theoretical and experimental work in the multicalibration literature. We then detail our key experimental design choices in Section 2, before discussing our results on tabular data in Section 3. We extend our results to more complex image and language datasets in Section 4. Finally, we conclude with limitations of our experiments as well as practical takeaways for practitioners of fair machine learning in Section 5.

---

[1]Experiment code is available at https://github.com/dutchhansen/empirical-multicalibration, while code for the python package is available at https://github.com/sid-devic/multicalibration.

## 1.2 Related Works: Theory and Practice

The theory of multicalibration is rife with theoretical results investigating the sample complexity (Shabat et al., 2020), learnability, and computational efficiency of multicalibrated predictors. Hébert-Johnson et al. (2018) initiated this study by showing that achieving multicalibration over a hypothesis class $\mathcal{C}$ defining protected subgroups requires access to a *weak agnostic learner* for that class (Shalev-Shwartz and Ben-David, 2014). From a fairness perspective, however, we are oftentimes—but not always (Sahlgren and Laitinen, 2020)—interested in subgroups defined by features or metadata, rather than a generic (and potentially infinite) hypothesis class. Subgroups in practical applications of algorithmic fairness are often given as input to the machine learning algorithm and intrinsic to a particular dataset of interest.

Although there are results describing and proving links between ERM and multicalibration in theory (Błasiok et al., 2024, 2023; Liu et al., 2019), we systematically evaluate when this link holds in practice across a broad range of models. To the best of our knowledge, only Barda et al. (2021); La Cava et al. (2022) consider issues when applying multicalibration in practice. Both works are limited to small models or only run experiments with one or two datasets. Pfohl et al. (2022) measure subgroup calibration, but do not discuss it at length. In recent work, Detommaso et al. (2024) utilize multicalibration as a tool to improve the overall uncertainty and confidence calibration of language models but, to our knowledge, do not focus on or report fairness towards protected subgroups. We provide additional discussion of related works in Appendix C.

## 2 Preliminaries

We work in the binary classification setting with a domain $\mathcal{X}$ and binary label set $\mathcal{Y} = \{0, 1\}$, and assume data is drawn from a distribution $\mathcal{D}$ over $\mathcal{X} \times \mathcal{Y}$. We consider arbitrary risk predictors $f : \mathcal{X} \to \Delta(\mathcal{Y})$, which return probability distributions over the binary label space. We will measure the calibration of $f$ on a dataset $S \in (\mathcal{X} \times \mathcal{Y})^n$ with the binned variant of the well-known and standard *Expected Calibration Error*, which we refer to as ECE (Guo et al., 2017). Throughout, we measure ECE with 10 bins of equal width 0.1.

Recent work has questioned ECE as a calibration measure, due to consistency and continuity issues that come with relying on a fixed bin width. To address these, we also report calibration as measured by *smoothed* ECE (smECE, Błasiok and Nakkiran (2023)), which (1) can be roughly thought of as the ECE after applying a suitable kernel-smoothing to the predictions, and (2) satisfies desirable continuity and consistency guarantees. Importantly, unlike binned ECE, there are no hyperparameters associated with measuring the smoothed calibration error. A full description of smECE is beyond the scope of our work—we refer the interested reader to Błasiok and Nakkiran (2023).

Multicalibration requires that a predictor have not only small calibration error overall, but also when restricted to marginal subgroup distributions of the data. In particular, we assume that there is a (finite) collection of groups $G = \{g_1, g_2, \dots\}$, where $g_i \subseteq \mathcal{X}$. We operationalize *measuring* multicalibration by reporting the *maximum* calibration error over a given collection of subgroups $G$.[2] Taking the max avoids fairness concerns associated with the (weighted) mean of groups of varying size and/or degree of overlap. Note that the subgroup collection $G$ is context and dataset dependent, and that the groups within $G$ may be overlapping, capturing desirable *intersectionality* notions (Ovalle et al., 2023).

### 2.1 Multicalibration Post-Processing Algorithms and Hyperparameter Selection

In theory, standard calibration post-processing methods like Platt scaling (Platt et al., 1999) or temperature scaling (Guo et al., 2017) do not guarantee that predictions will be well-calibrated on protected subgroups. Therefore, in order to achieve multicalibration, Hébert-Johnson et al. (2018) propose an iterative boosting-style post-processing algorithm which we refer to as `HKRR`. The algorithm works by iteratively searching for and removing subgroup calibration violations until convergence. We detail the algorithm's hyperparameters and the values we choose for them in Appendix F.1, and note that we perform a relatively wide parameter sweep.

The recent work of Haghtalab et al. (2023) also provide a *family* of alternative multicalibration algorithms with better theoretical sample complexity guarantees. This is motivated by the fact that `HKRR` is known to be theoretically *sample inefficient* (Gopalan et al., 2022b), and easily overfits

---

[2] Multicalibration was introduced before smECE, and was designed to reduce a bucketed group-wise calibration error (similar to ECE). Therefore, our investigations concerning smECE are of a purely empirical character.

(Detommaso et al., 2024).[3] At a high level, each algorithm of Haghtalab et al. (2023) corresponds to a certain two-player game. Different algorithms in the family are a consequence of each player playing a different online learning algorithm. We detail the hyperparameters over which we search in Appendix F.2, but note here that we use the same code and predominantly the same parameters reported by the authors. We refer to any (post-processing) algorithm in this family as HJZ.

In addition to the multicalibration algorithms HJZ and HKRR, we test the usefulness of three standard *calibration* techniques in reducing multicalibration error: Platt scaling (Platt et al., 1999), isotonic regression (Zadrozny and Elkan, 2002), and temperature scaling Guo et al. (2017). The first two techniques are hyperparameter-free, and we use implementations given by Scikit-learn. We also use a parameter-free version of temperature scaling which we detail in Appendix F.3.

## 2.2   Subgroup Selection, Datasets, and Experimental Methodology

Multicalibration post-processing requires the selection of "groups" or subsets of the population of interest. As our investigation is primarily motivated by fairness desiderata, these subgroups determine what segments of the population the practitioner would like to "protect" or guarantee performance over. In most practical applications, these subgroups are constructed via features or conjunctions of features given as input for each data point. This way of constructing groups is standard: it is used by large production systems such as LinkedIn (Quiñonero Candela et al., 2023), in the measurement of bias in ML (Atwood et al., 2024; Tifrea et al., 2024) and NLP systems (Baldini et al., 2022; Li et al., 2023), and in the *auditing* of large, deployed ML systems (Ali et al., 2019; Imana et al., 2024).

We experiment across a variety of classification tasks: five tabular datasets (ACS Income, UCI Bank Marketing, UCI Credit Default, HMDA, MEPS), two language datasets (Civil Comments, Amazon Polarity), and two image datasets (CelebA, Camelyon17). For each dataset, we also define between 10 and 20 overlapping subgroups depending on available features or metadata. We detail and provide citations for each of our datasets and exact subgroup descriptions in Appendix E. In what follows, we give a high level overview of how we determined subgroups in our experiments.

For our tabular datasets, we determined groups by "sensitive" attributes—individual characteristics against which practitioners would not want to discriminate. In many cases, such attributes naturally include race, gender, and age, and vary with available information. For example, on ACS Income, we include groups such as "Multiracial" and "Seniors." We also include some groups which are conjunctions of two attributes, for example "Black Women" or "White Women."

On datasets where samples are not in correspondence with individuals—Camelyon17, Amazon Polarity, and Civil Comments—we define groups based on available information that can be viewed as "sensitive" with respect to the underlying task. In other words, we define groups such that an individual or institution using a predictor which is miscalibrated on this group may be seen as discriminating against the group. For example, a social media service should ideally not be *underconfident* when predicting the toxicity of posts mentioning a minority identity group; such predictions may allow hate speech to remain on the platform, or may provide differential engagement boosting based on the presence of racial identifiers in posts. Therefore, we include "Muslim," "Black," and various other phrases defining protected groups in the Civil Comments dataset. In Appendix B, we further discuss group selection methodology and speculate about other ways of achieving multicalibration via group design.

**Data Partitioning.**   For consistency, we partition all datasets into three subsets: training, validation, and test. Test sets remain fixed across all experiments. We report accuracy and multicalibration metrics on the test set averaged over five random splits of train and validation sets for tabular data, and three splits for more complex data. Whenever a (multi)calibration post-processing algorithm is used, we run it using a holdout set of variable size from our training set, which we term the *calibration set*. We define the fraction of the training set used in (multi)calibration post-processing to be the *calibration fraction*. The exact calibration fractions over which we search appear in Appendix G.1 for tabular datasets and Appendix G.2 for image and language datasets. Note that multicalibration post-processing methods are far less sample efficient than standard post-hoc calibration methods. Therefore, the calibration fractions we test are broadly distributed between 5% and 100% of the training data (rather than using, say, a standard 10% of data for post-hoc calibration).

---

[3]For example, the number of samples required for generalization guarantees of HKRR is typically $O(\frac{1}{\alpha^4 \lambda^{1.5}})$, where $\alpha$ determines the allowed multicalibration violation and $\lambda$ represents a suitable discretization width. For reasonably small values of $\alpha$ and $\lambda$, this can balloon the required number of samples to an unreasonable number.

The calibration set is used solely in multicalibration post-processing, and is *not* used in training a model prior to the post-process. This procedure is motivated by a need to measure the importance of *fresh* samples in multicalibration post-processing. If a model is already multicalibrated on its entire training set $S$, we *cannot* re-use $S$ in HKRR or HJZ to improve the model, since the algorithms cannot improve on a predictor which is already perfectly multicalibrated on a particular dataset. This applies to models such as neural networks, which usually fit their training set to very low calibration error and high accuracy (Carrell et al., 2022). For these models, we also anticipate that the *multicalibration* error on the training set will be low, and hence, the data from $S$ unusable for post-processing.[4] Therefore, the calibration fraction itself is an important hyperparameter we consider. Ideally, in order to maximize the resulting accuracy of the final model, we would utilize as much data as possible for model training, and minimize the amount of data required for multicalibration post-processing. However, due to the sample complexity of multicalibration algorithms, we will see that finding this specific point can be difficult (see Figure 3).

**Compute.** All experiments were performed on a collection four AWS G5 instances, each equipped with a NVIDIA 24GB A10 GPU. We used only the CPU for multicalibration and calibration post-processing, which was by far the most computationally intensive task. We estimate that all of our experiments cumulatively took 10 days of running time on these four instances.

## 3 Experiments on Tabular Datasets

We begin our investigation with tabular data. Although simpler than vision or language data, tabular data is an important and realistic setting which many algorithmic fairness practitioners encounter throughout the health, criminal justice, and finance sectors (Barda et al., 2021; Barenstein, 2019; Obermeyer et al., 2019). As our base predictors in this setting, we consider multilayer perceptron NNs (MLPs), decision trees and random forests, SVMs, naive Bayes, and logistic regression. We defer dataset and group details to Appendix E, and model details to Appendix G.1. We note here that our datasets span from 10K examples (MEPS) to 200K (ACS Income), and that we vary the size of the calibration set between 5% to 100% of the available training data. All of our results are computed with a mean and standard deviation over five train / validation splits. We instill the following insights from running multicalibration post-processing algorithms over 40K times on over 1K separately trained models.

**Observation 1:** On tabular data, ML models which tend to be calibrated out of the box also tend to be multicalibrated without additional effort.

In Figure 1, we show the performance of every choice of multicalibration algorithm (corresponding to each choice of aforementioned hyperparameters) for MLPs on three datasets: MEPS, Credit Default, and ACS Income. We find that ERM performs nearly as well — in terms of worst group calibration error — as the best set of hyperparameters for HJZ and HKRR across our wide parameter sweep. This is seen broadly across all of our tabular datasets for models which one may expect to be calibrated in practice, such as logistic regression or random forests.[5] We include the complete plots of all multicalibration runs versus ERM in Appendix H.1.

We provide further evidence for Observation 1 by inspection of Figure 2. This table corresponds to the best choice of hyperparameters (according to maximum group-wise smECE on a validation dataset) of each method tested on the MEPS dataset. We find that HKRR and HJZ show no statistically significant improvements to max smECE for MLPs, random forests, and logistic regression. The gains offered by HKRR and HJZ in terms of worst group calibration error are also marginal (0 to 0.01) on the Bank Marketing, ACS Income, and Credit Default datasets (see Appendix H.2). On HMDA, however, multicalibration does seem to provide a noticeable improvement on the order of 0.03-0.07 for MLPs, random forests, and logistic regression (Figure 27). We believe this is because ERM achieves worse *calibration* error on HMDA, possibly due to the increased difficulty of the dataset.

**Observation 2:** HKRR or HJZ post-processing can help un-calibrated models like SVMs or naive Bayes achieve low group-wise maximum calibration error. Oftentimes, however, similar results can be achieved with traditional calibration methods like isotonic regression (Zadrozny and Elkan, 2002).

---

[4]Indeed, we test this more rigorously in Appendix A, where we experiment with data reuse between model training and multicalibration post-processing.

[5]We use the `Scikit-learn` random forest implementation, which predicts a *probability* corresponding to the fraction of positive points at the leaf.

| Model | ECE ↓ | Max ECE ↓ | smECE ↓ | Max smECE ↓ | Acc ↑ |
|---|---|---|---|---|---|
| MLP ERM | 0.022 ± 0.006 | 0.106 ± 0.009 | 0.024 ± 0.002 | 0.086 ± 0.015 | 0.864 ± 0.001 |
| MLP HKRR | 0.019 ± 0.005 | 0.122 ± 0.008 | **0.019 ± 0.004** | 0.104 ± 0.002 | 0.835 ± 0.003 |
| MLP HJZ | 0.019 ± 0.003 | **0.088 ± 0.011** | 0.021 ± 0.002 | **0.076 ± 0.018** | 0.864 ± 0.003 |
| MLP Isotonic | 0.02 ± 0.006 | 0.108 ± 0.021 | 0.02 ± 0.004 | 0.089 ± 0.021 | 0.864 ± 0.003 |
| RandomForest ERM | 0.019 ± 0.001 | 0.094 ± 0.006 | 0.021 ± 0.001 | **0.083 ± 0.004** | 0.863 ± 0.003 |
| RandomForest HKRR | 0.019 ± 0.005 | 0.122 ± 0.008 | 0.019 ± 0.004 | 0.104 ± 0.002 | 0.835 ± 0.003 |
| RandomForest HJZ | 0.021 ± 0.004 | 0.106 ± 0.011 | 0.021 ± 0.003 | 0.101 ± 0.012 | 0.86 ± 0.003 |
| RandomForest Isotonic | **0.015 ± 0.002** | **0.089 ± 0.014** | **0.017 ± 0.001** | 0.084 ± 0.014 | 0.862 ± 0.002 |
| SVM ERM | 0.143 ± 0.002 | 0.376 ± 0.012 | 0.072 ± 0.001 | 0.186 ± 0.006 | 0.857 ± 0.002 |
| SVM HKRR | **0.019 ± 0.005** | **0.122 ± 0.008** | **0.019 ± 0.004** | **0.104 ± 0.002** | 0.835 ± 0.003 |
| SVM HJZ | 0.031 ± 0.003 | 0.156 ± 0.021 | 0.027 ± 0.004 | 0.155 ± 0.02 | 0.828 ± 0.002 |
| SVM Isotonic | 0.048 ± 0.023 | 0.231 ± 0.085 | 0.048 ± 0.023 | 0.218 ± 0.069 | 0.847 ± 0.017 |
| LogisticRegression ERM | 0.022 ± 0.002 | **0.106 ± 0.008** | 0.022 ± 0.001 | **0.083 ± 0.003** | **0.866 ± 0.002** |
| LogisticRegression HKRR | 0.019 ± 0.005 | 0.122 ± 0.008 | **0.019 ± 0.004** | 0.104 ± 0.002 | 0.835 ± 0.003 |
| LogisticRegression HJZ | 0.021 ± 0.003 | 0.114 ± 0.019 | 0.023 ± 0.001 | 0.09 ± 0.011 | 0.866 ± 0.003 |
| LogisticRegression Isotonic | **0.017 ± 0.003** | 0.109 ± 0.019 | 0.019 ± 0.003 | 0.097 ± 0.02 | 0.863 ± 0.002 |
| DecisionTree ERM | 0.067 ± 0.004 | 0.261 ± 0.028 | 0.047 ± 0.004 | 0.166 ± 0.012 | **0.85 ± 0.006** |
| DecisionTree HKRR | 0.019 ± 0.005 | **0.122 ± 0.008** | 0.019 ± 0.004 | **0.104 ± 0.002** | 0.835 ± 0.003 |
| DecisionTree HJZ | 0.031 ± 0.003 | 0.156 ± 0.021 | 0.027 ± 0.004 | 0.155 ± 0.02 | 0.828 ± 0.002 |
| DecisionTree Isotonic | **0.014 ± 0.003** | 0.196 ± 0.026 | **0.015 ± 0.003** | 0.186 ± 0.027 | 0.838 ± 0.01 |
| NaiveBayes ERM | 0.277 ± 0.019 | 0.544 ± 0.02 | 0.164 ± 0.013 | 0.287 ± 0.011 | 0.714 ± 0.018 |
| NaiveBayes HKRR | 0.019 ± 0.005 | **0.122 ± 0.008** | **0.019 ± 0.004** | **0.104 ± 0.002** | **0.835 ± 0.003** |
| NaiveBayes HJZ | 0.031 ± 0.003 | 0.156 ± 0.021 | 0.027 ± 0.004 | 0.155 ± 0.02 | 0.828 ± 0.002 |
| NaiveBayes Isotonic | **0.019 ± 0.005** | 0.128 ± 0.017 | 0.021 ± 0.005 | 0.122 ± 0.015 | 0.831 ± 0.006 |

Figure 2: Best performing HKRR and HJZ post-processing algorithm hyperparameters (selected based on validation max smECE) compared to ERM on the MEPS dataset. Calibrated models (MLP, random forest, logistic regression) need not be post-processed to achieve multicalibration. However, uncalibrated models (SVM, decision trees, naive Bayes) *do* benefit from multicalibration post-processing algorithms. Cells highlighted in blue show the importance of the choice of metric for selecting the best post-processing method for decision trees. Metric choice — worst group ECE vs. worst group smECE — can change which of ERM or HJZ is preferable.

Across our datasets, we find that SVMs, decision trees, and naive Bayes almost always have their max smECE error improve by 0.05 or more using multicalibration post-processing. We also point out the relatively strong performance of isotonic regression and other traditional calibration methods across datasets and models. For example, isotonic regression provides nearly all the improvements (up to 0.01 error) of the multicalibration algorithms when applied to naive Bayes in Figure 2. On the Credit Default dataset in Figure 28, isotonic regression is — when considering standard deviation — tied with the *optimal* multicalibration post-processing algorithms for SVM and naive Bayes. We have similar findings for the MEPS dataset and random forests trained on the HMDA dataset. Platt scaling and isotonic regression are desirable because they are *parameter-free* methods which work out of the box without tuning, are simple for practitioners to implement, and further do not require large parameter sweeps to find effective models.

**Observation 3:** A practitioner utilizing multicalibration post-processing can potentially face a trade-off between worst group calibration error and overall accuracy. This is most salient in high calibration fraction regimes (40-80%).

Due to the necessity of using hold-out data for running multicalibration post-processing, practitioners may have to choose between accuracy and worst group calibration error. For example, in Figure 3 we show that running multicalibration post-processing for MLPs on the HMDA dataset has a different optimal calibration fraction when considering accuracy and worst group calibration error as separate objectives. In fact, in this example, improving multicalibration error comes at a *cost* to accuracy of about 2%. Although small, this does indicate that the decision to use multicalibration post-processing here should be context-dependent. Additional examples of this tradeoff include decision trees or logistic regression on most datasets (Figures 31 and 33). We note, however, that these tradeoffs are more apparent in the higher calibration fraction regimes, where most of the training data is *held out* for multicalibration post-processing. This regime is potentially less relevant to practitioners, who

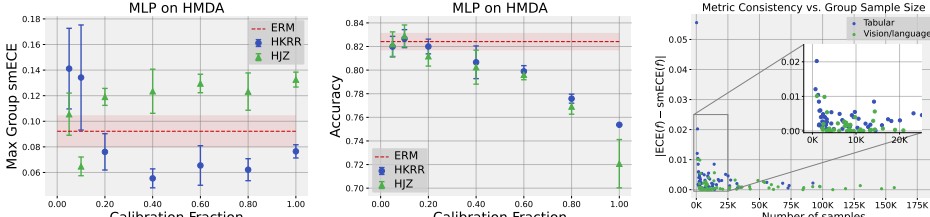

Figure 3: (**Left/Middle**): Hold-out calibration fraction vs. worst group calibration error (left) and accuracy (right) for MLPs on HMDA. Lowering worst group calibration error may come at a cost of model accuracy. The impact of calibration fraction for each dataset is available in Appendix H.3. (**Right**): Gap between measured smECE and ECE for every experiment. As sample size increases, the two metrics become very similar. However, some variability exists at lower sample sizes.

usually reserve most of the available data for base model training. Plots for each dataset and model are in Appendix H.3.

**Observation 4:** On small datasets, there can be variations between smECE and standard binned ECE.

To illustrate an example, if a practitioner were selecting the best post-processing method for decision trees from Figure 2 based on ECE (see table cells highlighted in blue), HJZ may seem like a reasonable choice since it has a worst group ECE calibration error of 0.156. However, when using worst group smECE to measure performance, HJZ does *not* significantly improve upon ERM. This has an important consequence: if selecting the best model based on only the worst subgroup calibration error, the choice of *calibration metric* used will impact the choice of model.

In the rightmost plot of Figure 3, we also show each group's sample size vs. the gap between measuring the group calibration error with smECE vs using ECE (over all datasets and groups). We find that as the group sample size increases, the gap between the metrics generally shrinks (and **Observation 4** becomes less relevant). We note, however, that even on the ACS Income dataset with 200K examples, we find a significant difference of 0.1 between measuring the *overall* calibration error of SVMs with ECE vs. smECE (Figure 25). More generally, to avoid issues stemming from ECE bin choice, we recommend that practitioners utilize the smECE calibration measurement tool[6] due to its theoretical guarantees and stability across our experiments.

**Observation 5:** When considering statistical significance, there is no clearly dominant algorithm between HKRR and HJZ on tabular data. However, HJZ is more robust towards the tested choice of hyperparameters. This may allow practitioners utilizing HJZ to find good solutions faster than using HKRR when post-processing simpler models such as naive Bayes or decision trees.

Over all tabular datasets and all base models (Appendix H.2), HJZ and HKRR had statistically distinguishable performance on 24 out of 30 cases. Among these 24 cases, HJZ performed better 7 times. Nonetheless, we observe that the HJZ family of algorithms is usually *less sensitive* to hyperparameter changes. In Figure 1 for example, most of the green points corresponding to hyperparameter choices for HJZ are tightly concentrated around ERM. We observe similar phenomena throughout additional model and dataset plots in Appendix H.1. Practitioners wishing to apply smaller hyperparameter searches over multicalibration algorithms may consider HJZ a suitable option, even if it gives slightly suboptimal worst group calibration error.

**Additional Experiments.** To understand how sensitive our observations are to our particular choice of group collection $G$, we also validate each of these observations with a *new* set of defined groups $G'$, whose definitions are found in Appendix E.5. The full tabular results and plots for these new groups for each dataset is in Appendix I.1. Overall, the takeaway for most models (including MLPs) largely remains the same: it is difficult to find instances where multicalibration helps in a statistically significant way over ERM (for calibrated models) or some form of simple calibration (**Observations 1** and **2**). Further, where multicalibration does help, this help may sometimes comes at a cost to accuracy (**Observation 3**).

---

[6] https://github.com/apple/ml-calibration

# 4 Experiments on Language and Vision Datasets

In this section, we evaluate the ability of multicalibration post-processing to improve upon the multicalibration of vision transformers, DistilBERT, ResNets, and DenseNets on a collection of image and language tasks. Our goal is to understand if multicalibration post-processing can help in more complicated, large-model regimes within both the train-from-scratch and pre-trained paradigms.

As we move from smaller, tabular datasets to larger image and language datasets, we find that multicalibration algorithms may provide empirical improvements. Note here that in cases where we use a ResNet on language data, we train from scratch but use pretrained GloVe embeddings in the fashion of Duchene et al. (2023). In cases where we use a ResNet or DenseNet on image data, we also train from scratch. Whenever using a transformer, we finetune from pretrained weights (Dosovitskiy et al., 2021; Sanh et al., 2019). We defer further dataset, model, and group information to Appendix E and Appendix G.2.

Multicalibrating large models has an increased computational cost: For a single base predictor on tabular data, a full parameter sweep—training the multicalibration algorithm with every choice of hyperparameter—required 1-2 hours on a typical calibration set of 100K examples. With more complex base predictors (e.g. ResNets or language models) and larger datasets, this process takes significantly longer. Additionally, due to the increased computational cost of *re-training* a model in the image and language regimes, we only search over calibration fractions in $\{0.0, 0.2, 0.4\}$ and report our results averaged over three random train / validation splits. After running multicalibration post-processing algorithms more than 1,700 times, we distill our findings into the following observations.

**Observation 6:** For image and language data, HKRR nearly always outperforms HJZ.

In all six of our experiments on image and language data (Appendix J.2), HKRR either matched or significantly beat the performance of HJZ. Note that we use the same parameter sweeps for HKRR and HJZ over image/language datasets that we used for the tabular datasets (see Appendix F), and leave open the possibility that HJZ may require a larger hyperparameter sweep to achieve good performance on these more complex tasks.

**Observation 7:** On language and vision data, multicalibration post-processing can improve worst group calibration error relative to neural network ERM baselines by 50% or more. This stands in contrast to multicalibration post-processing for MLPs on tabular data (**Observation 1**).

Over all language and vision datasets, HKRR improved worst group calibration error in 5 out of 6 cases. Among these 5, the least improvement we saw was HKRR decreasing the worst group smECE of ERM from 0.06 to 0.043 (DistilBERT on Civil Comments). The greatest improvement we saw was from 0.07 to 0.02 (ViT on Camelyon17) and 0.09 to 0.05 (ResNet-56 on Amazon Polarity). These examples all appear in Figure 4. A full collection of tables and plots can be found in Appendix J.2.

**Observation 8:** Binned ECE and smECE provide nearly identical estimates of calibration error.

Among nearly all of our experiments on vision and language datasets with more than 100k examples, we were not able to find any datasets where the metric used to measure worst group calibration error would change the outcome of chosen model. This suggests that larger sample sizes largely close observable gaps between calibration measures (c.f. **Observation 4**).

# 5 Takeaways for Practitioners and Discussion

In this section, we first provide reasonable recommendations to practitioners wishing to apply multicalibration algorithms in practice. In Appendix B, we also discuss additional details on the *subgroup selection* problem, which practitioners applying post-processing methods may find helpful.

First, we believe that the latent multicalibration of ERM has been generally underestimated for many models. In particular, on tabular datasets, multicalibration post-processing cannot improve upon ERM for MLPs (see **Observation 1**). Furthermore, the improvement offered for more complex image and language data is generally less than 0.05 smECE when considering standard deviation.

This directly motivates our next takeaway: Current multicalibration post-processing algorithms—when applied to calibrated models like neural networks—are extremely sensitive towards choice of hyperparameters, since the potential "scope" of improvement is on the scale of 0.02 to 0.03 smECE. The optimal hyperparameter choice for each algorithm largely varies by dataset and base model, and it takes quite a bit of granular searching to find the best performing algorithm, or indeed, an

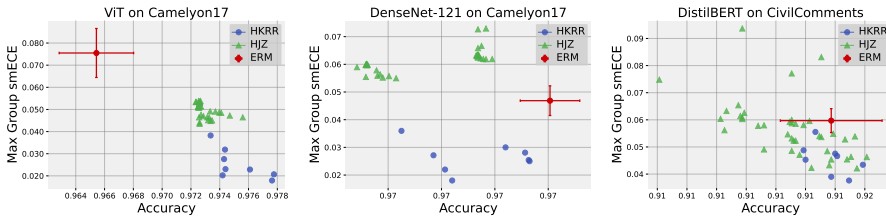

| Model | ECE ↓ | Max ECE ↓ | smECE ↓ | Max smECE ↓ | Acc ↑ |
|---|---|---|---|---|---|
| DistilBERT ERM | 0.021 ± 0.001 | 0.065 ± 0.005 | 0.021 ± 0.001 | 0.06 ± 0.004 | 0.915 ± 0.001 |
| DistilBERT HKRR | 0.013 ± 0.0 | 0.047 ± 0.005 | 0.013 ± 0.0 | 0.043 ± 0.004 | 0.915 ± 0.001 |
| DistilBERT HJZ | 0.004 ± 0.001 | 0.043 ± 0.008 | 0.007 ± 0.001 | 0.043 ± 0.007 | 0.915 ± 0.001 |
| DistilBERT Isotonic | **0.002 ± 0.0** | **0.032 ± 0.006** | **0.005 ± 0.0** | **0.032 ± 0.006** | **0.916 ± 0.0** |
| ResNet-56 ERM | 0.039 ± 0.013 | 0.094 ± 0.009 | 0.039 ± 0.013 | 0.094 ± 0.009 | **0.867 ± 0.001** |
| ResNet-56 HKRR | 0.015 ± 0.001 | **0.059 ± 0.01** | 0.015 ± 0.001 | **0.047 ± 0.005** | 0.848 ± 0.004 |
| ResNet-56 HJZ | 0.013 ± 0.005 | 0.081 ± 0.012 | 0.014 ± 0.005 | 0.081 ± 0.012 | 0.863 ± 0.002 |
| ResNet-56 Isotonic | **0.005 ± 0.001** | 0.079 ± 0.009 | **0.007 ± 0.0** | 0.078 ± 0.008 | 0.863 ± 0.002 |

Figure 4: (**Top**): Test accuracy vs. maximum group-wise calibration error (smECE) over three train/validation splits for ViT and DenseNet on Camelyon17, and DistilBERT on CivilComments. Multicalibration post-processing has scope for improvement in each setting, and does so with nearly no loss in accuracy. (**Bottom**): Impact of post-processing algorithms for Civil Comments (DistilBERT) and Amazon Polarity (ResNet-56). Multicalibration and isotonic regression both offer improvements to worst group calibration error. Full results are available in Appendix J.1.

algorithm which improves upon ERM at all. For example, the optimal HJZ algorithm used at least 15 different hyperparameter configurations across only our 30 tabular experiments (when considering calibration fraction as an additional parameter); HKRR has similar sensitivity issues. Further, many hyperparameter choices do not seem to improve upon the ERM base model—for example, see DenseNet-121 in Figure 4 or the full plots in Appendix J.1—making a significant portion of the hyperparameter sweeps not useful to perform. Since training HJZ or HKRR on a holdout of 100K examples can take 1-2 hours, it can be several hours before a suitable choice of hyperparameters is found. This computational cost is exacerbated in the larger regimes where multicalibration may be most useful, which poses a major obstacle for practical applications of either HKRR or HJZ.

As a direct stopgap measure, we recommend running and evaluating traditional calibration methods. As we point out in **Observation 2**, post-processing algorithms like isotonic regression can achieve nearly the performance of multicalibration algorithms on tabular data. Isotonic regression also directly improves worst group calibration error over ERM in 4/6 of our experiments on larger models (see, e.g., Figures 4, J.2). Due to the fact that it is efficient and parameter free, we do not see a downside to running Isotonic regression (or any other calibration method) and testing if the maximum group-wise calibration error is beneath a desired threshold.

## 6 Experimental Limitations and Conclusion

One limitation of our results is that they are restricted to binary classification problems. While multicalibration algorithms do extend to multiclass problems, this extension comes at a severe cost of sample efficiency usually *exponential* in the number of labels (Zhao et al., 2021). We show that — at least for tabular datasets — current multicalibration algorithms do not significantly improve upon a competitive and calibrated ERM baseline. If we were to further burden the multicalibration algorithm with the larger sample complexity of an additional label, we do not expect that their performance will improve. Nonetheless, we plan to investigate the multiclass setting in future work, and believe that those findings will be consistent with the results present in this paper. Another limitation is that we do not offer much explanation of why we see differing performance of the two algorithms HJZ and HKRR; we offer some discussion of this in Appendix C.3, but more is warranted in future work.

We believe that our work illuminates many avenues towards improving the viability of multicalibration algorithms in practice. For example, developing parameter free multicalibration methods (akin to what smECE accomplishes for calibration metrics) is an important direction with direct impacts on the practice of fair machine learning. Similarly, post-processing techniques with better empirical sample complexity could significantly help the practice of multicalibration.

**Acknowledgements.** SD was supported by the Department of Defense through the National Defense Science & Engineering Graduate (NDSEG) Fellowship Program. This work was also supported by NSF CAREER Award CCF-2239265 and an Amazon Research Award. Any opinions, findings, and conclusions or recommendations expressed in this material are those of the author(s) and do not reflect the views of sponsors such as Amazon or NSF. The authors would like to thank Bhavya Vasudeva for discussions that were helpful in the design of early experiments, and Eric Zhao for help in utilizing the HJZ algorithms. The authors also sincerely thank the anonymous Neurips reviewers for providing detailed feedback and discussion which greatly improved and clarified parts of this work.

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

# Contents

# A   Simplifying Data Partitioning with Data Reuse

As discussed in Section 2.2, throughout our experiments on tabular, vision, and language data in Sections 3 and 4, we held out a portion of data from training solely for running multicalibration post-processing. This was motivated by two facts: (1) multicalibration may require *fresh* samples for theoretical statistical guarantees; and (2) if a model already has low worst group calibration error on a holdout set $S$, that set $S$ cannot be used for post-processing since there are no "group calibration violations" that either HJZ or HKRR can correct. Fact (1) holds for any base model (neural networks, decision trees, random forests, etc.), while (2) only holds for models which we believe may always achieve perfect training loss or calibration error, like neural networks.

We investigate these two facts experimentally by asking whether a practitioner may *reuse* a portion of the base model training data for multicalibration post-processing. Such data reuse could be very convenient to practitioners already saddled with hyperparameter optimizations. To test for this, we run ERM on all available training data, and run each post-processing method on this *same* set of data. Again searching over all post-processing hyperparameters (now except for calibration fraction), we provide test results corresponding to hyperparameters that achieved the best validataion max smECE. We provide full results for all tabular datasets in Appendix H.4.

**Observation 9:** Reusing model training data for multicalibration post-processing can sometimes be competitive with holding out data for post-processing. However, it can also come at a steep cost to worst group calibration error.

In some cases, reusing training data can marginally improve max smECE over the setup in which we do not reuse data; this is true, for example, for MLPs post-processed with HKRR on the left of Figure 5. Here, reusing training data improves upon using a calibration holdout by 0.025. In the vast majority of cases, however, reusing training data either (1) does not improve upon utilizing a holdout calibration set (for most models on ACSIncome or HMDA); or (2) significantly hurts (for MEPS, CreditDefault, or BankMarketing). For example, many base models are significantly hurt by data reuse for multicalibration post-processing on the CreditDefault dataset (in the right of Figure 5), having their post-processed performance drastically drop by 0.05-0.1 worst group smECE. These results demonstrate that practitioners utilizing multicalibration algorithms in practice may be required to optimize over the calibration fraction holdout size in order to achieve competitive empirical performance.

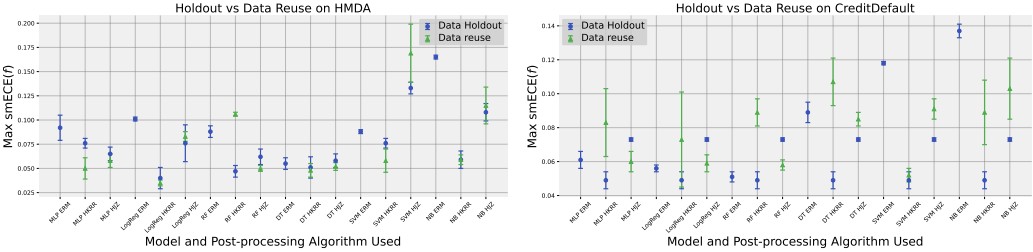

Figure 5: Impact of reusing all model training data for multicalibration post-processing on HMDA (**Left**) and CreditDefault (**Right**) as measured by worst group calibration error (max smECE). Results vary; for HMDA, post-processing with reused data essentially performs as well as post-processing by holding out data for all models except random forest postprocessed with HKRR. However, on CreditDefault, we find that data reuse can harm post-processing across the board. Plots for each dataset available in Appendix H.4.1.

# B  Additional Subgroup Design Considerations

For practitioners, there are (at least) two important properties of groups to consider during the group selection phase: minimum group *size* and the *richness* of the group collection. The minimum group size $\gamma$ is a parameter which has implications for the overall sample complexity of multicalibration. In particular, it introduces a $1/\gamma$ factor into known sample complexity upper and lower bounds (Hébert-Johnson et al., 2018; Shabat et al., 2020). Note that $\gamma \in [0, 1]$ is the size — as a fraction of the dataset under a distribution $\mathcal{D}$ — of the smallest group in the collection $G$. Therefore, there is a tradeoff between the *size* of the smallest group considered, and the number of samples needed for good multicalibration generalization.

In our experiments, we restricted to groups which were >0.5% of the entire dataset ($\gamma = 0.005$). This was a reasonable "sweet spot" for us: Without enough samples from a particular group, known multicalibration algorithms are prone to overfitting the training set and not providing desirable generalization performance. On the other hand, if groups are not small *enough*, then the guarantees provided by running multicalibration algorithm may not be much better than standard calibration methods. Note that we consider collections of groups with sizes spanning from 0.5% all the way to 70-80% of the data (see Appendix E for detailed information). We deem this range reasonably sufficient to capture the varying sizes of groups that a practitioner may desire to protect in practice.

Group *richness* is also an important factor in group design. As discussed in Section 2.2, our results are only relevant to the setting where (1) we have well-defined groups that we seek to protect which are defined by simple features or conjunctions of features; and (2) we run multicalibration post-processing with those same groups. These two points naturally give rise to a potentially promising direction: is it practically feasible to multicalibrate with respect to a *richer* class of subgroups in order to obtain better empirical performance with respect to the simpler groups which one may actually care about? For example, one could test whether multicalibrating with respect to the group of all halfspaces would provide improved worst group calibration error over the class of feature-defined groups than one may actually care about final performance for.[7] We leave such exploration to future work.

**Remark 1** *Using conjunctions of features or additional meta-data is not the only way to construct subgroups of the data. For any imperfect predictor $\hat{p}$, we can (nearly) always construct groups against which the predictor is not multicalibrated. For example, simply take the set of data points for which $\hat{p}$ predicts the incorrect label. Groups defined in this way may potentially be "as complex" as the underlying predictor $\hat{p}$. Nonetheless, it is not clear whether this is a meaningful set of groups to ask for multicalibration against (as it may require post-processing $\hat{p}$ to be a perfect predictor). To avoid such discussion, we intentionally determine groups by available features which we hypothesize practitioners may deem important or "sensitive" to the underlying prediction task.*

# C  Additional Related Work

There is reason to believe that empirical risk minimization (ERM) on neural networks and other machine learning models may result in multicalibrated predictors. In recent works, Błasiok et al. (2024, 2023) prove that loss minimization with neural networks may yield multicalibrated predictors. Their proofs, however, may not be directly applicable to practice as they rely on an idealized optimization procedure (we provide further discussion of the relation between our works in Appendix C.1). Nonetheless, both works echo a relationship between ERM and multicalibration also articulated in Liu et al. (2019), who show that group-wise calibration may be an inevitable consequence of well-performing models.

In recent work, Detommaso et al. (2024) utilize multicalibration as a tool to improve the overall uncertainty and confidence calibration of language models but, to our knowledge, do not focus on or report fairness towards protected subgroups. Like us, they point out various issues with the standard multicalibration algorithm, which they address with early stopping and adaptive binning. We instead perform a large hyperparameter sweep which effectively implements an early stopping mechanism. We discuss this further in Appendix C.2. Nonetheless, our results for large models are complementary to those of Detommaso et al. (2024): both works demonstrate that (1) standard multicalibration can

---

[7]Surprisingly, it is possible to multicalibrate with respect to the (rich) class of all halfspaces with access to an agnostic halfspace learner (Hébert-Johnson et al., 2018). In general, such agnostic learners are computationally hard to obtain in theory, but usually easy to construct in practice (via ERM).

at times be difficult to get working in practice; and (2) ideas from the theoretical multicalibration literature can have impact at the scale of large models.

**Limitations of Calibration.** A collection of works characterize the limitations of calibration as a property of predictors. Of particular note is Perez-Lebel et al. (2023), who remark that calibration is often misunderstood in the literature, as it does not guarantee that output probabilities are close to the ground truth probability distribution. We make no such claim about calibration, and only justify its use in order to ensure model predictions are meaningful. Yuksekgonul et al. (2024) draw connections between calibration and atypicality of certain examples, improving group wise-performance of NNs without subgroup annotations via what they term "atypicality-aware recalibration."

**Applications of Multicalibration.** Beyond classification, Globus-Harris et al. (2023) introduce algorithms that post-process multicalibrated regression functions to satisfy a variety of fairness constraints, and present experiments using such algorithms on logistic regression and gradient-boosted decision trees. Benz and Rodriguez (2023) study predictive confidence in the setting of AI-assisted decision making; they show the existence of distributions under which a "rational" decision maker is unlikely to find an optimal policy using calibrated confidence values, and prove that multicalibration with respect to the decision maker's preliminary confidence values is often sufficient for aversion of such issues. Zhang et al. (2024) also utilize a generalization of multicalibration to tackle interesting problems like de-biased text generation and false negative rate control.

**Calibration of NNs.** Literature on the calibration of neural networks (NNs) is very rich; see for example a treatment by Wang (2023). Most importantly, Minderer et al. (2021) have run comprehensive, large-scale experiments detailing the degree to which modern NNs are calibrated. Their results show that current state-of-the-art models appear to be nearly perfectly calibrated, and appear to remain so even in the presence of distribution shift. This is in somewhat striking contrast to the earlier results of Guo et al. (2017), which demonstrated the best models at time of their publication to be quite miscalibrated, and highlighted the need to further investigate calibration measures. Our focus in this work is instead on evaluating multicalibration of predictors on datasets over which we can naturally define a collection of protected subgroups.

Carrell et al. (2022) examine a connection between generalization and *calibration generalization*, the difference in calibration error on train and test sets. In particular, they claim DNNs to be well-calibrated on their training sets and the accuracy generalization gap to upper bound the calibration generalization gap. Such observations imply NNs that generalize well to be well-calibrated.

**Trainable Calibration Measures.** Laplace Kernel calibration measures have been shown to be effective in enforcing confidence calibration for neural networks when used in training. In particular, using MMCE (Maximum Mean Calibration Error) as a regularizer in tandem with cross-entropy loss yields high accuracy predictions while moderately improving calibration by taming overconfident predictions (Kumar et al., 2018). Additionally, this metric is efficiently computable in quadratic time.

**Subgroup Robustness.** Several works in subgroup robustness literature examine the performance of NNs by *worst-group-accuracy*, particularly in cases where NNs tend to rely on spurious correlations. Recent works by Kirichenko et al. (2023); LaBonte et al. (2023) propose last-layer fine-tuning as a simple and computationally inexpensive way to do exactly that. Indeed, Mao et al. (2023) extend the method to address fairness concerns, appending a fairness constraint to the training objective during fine-tuning. Such works examine only one sensitive attribute at a time, and often consider the disjoint groups produced by unique values of this attribute in conjunction with label. Rosenfeld and Garg (2023) show connections between robustness and distribution shift for neural networks via unlabeled test data.

In general, we view multi-group robustness as a notion of robustness which, like multicalibration, aims to "respect" multiple groups simultaneously. Perhaps the closest connection between multigroup robustness and multicalibration is: they intuitively may share similar mechanisms (as the reviewer noted). That is, the mechanistic question in both cases is to understand which groups can be easily computed from the predictor's "features" – ie, which groups are easy for the predictor to distinguish.

## C.1  Further Discussion of Błasiok et al. (2024)

The connection to our work is subtle. In Błasiok et al. (2024), the authors consider performing ERM over some $\mathcal{C}' \supseteq \mathcal{C}$ so that the resulting model is multicalibrated with respect to $\mathcal{C}$. In their case, $\mathcal{C}'$ is a family of neural networks (NNs), and $\mathcal{C}$ is some family of smaller NNs. In many of our experiments,

we indeed perform (approximate) ERM over some family of NNs, and one might expect this to result in multicalibration with respect to our finite collection of groups $G$, which are easily computable by some class of smaller NNs. This is because for NNs, it is possible to represent arbitrarily-complex groups by simply taking "large enough" networks, without making design choices specific to the group structure.

In our tabular experiments, we restrict our experiments to small multi-layer perceptron networks of 3-4 layers, whose smaller subnetworks may not be learning complex functions capturing groups of interest (see Appendix G for model descriptions). Nonetheless, we find that these models possess latent multicalibration properties, as discussed further in **Observation 1** in Section 3. Our vision and language experiments show that post-processing does have positive impact, suggesting that the models sub-networks are not sufficiently capturing the groups of interest (see Section 4).

## C.2   Equivalence of Early Stopping and Hyperparameter Sweeps

Detommaso et al. (2024) utilize two early-stopping criterion. The first is to halt further multicalibration iterations once the group conditioned on the bin set becomes "too small." We also utilize this technique, which is inherent in the algorithm of HKRR. Detommaso et al. (2024) also use a holdout validation set to early-stop a variant of HKRR when the mean-squared error (MSE) on the hold-out set fails to decrease. We instead vary the permitted violation parameter $\alpha$ of HKRR, and select the best parameter with a holdout validation set. $\alpha$ controls the permitted calibration error conditioned on a group and particular bin (of width $\lambda = 0.1$ in our work). We believe that early stopping for a validation metric should have similar performance to running the full HKRR with a variety of $\alpha$ levels, and selecting based on validation performance. Nonetheless, we note that in our experiments, we select based on validation smoothECE multicalibration error, while Detommaso et al. (2024) select on validation MSE. This could potentially lead to performance differences.

## C.3   Discussion on Dynamics and Performance of HJZ and HKRR

The dynamics of both of the algorithms HJZ and HKRR are complex. In particular, the performance of the algorithms depend on at least the following parameters:

1. Distribution of initial predictions output by the models (i.e. input to post-processing algorithm);

2. Choice of hyperparameters for HKRR and HJZ;

3. "Complexity" or "expressiveness" of the groups; and

4. Number of available samples, and whether the samples are re-used from training or not.

In our work, we focus mainly on (2) and (4). We discuss (3) to an extent in Appendix B, but teasing apart exactly how (1) and (3) contribute to the performance of multicalibration post-processign algorithms is certainly an interesting avenue for future work. We believe that part of the reason for the superiority of HKRR in language/vision data may be explained within the lens of (2) and (4). That is, we may have found better hyperparameters for HKRR with a wider search, and the sample complexity may be better in practice than the game-theoretic approach offered by HJZ.

Due to computational constraints and the added dimension of choosing how much data to save for calibration, we search a large — but not all-encompassing — collection of hyperparameters for each of the multicalibration algorithms tested. With regards to dataset size, 3 of the 4 vision/language datasets are noticeably larger than the tabular datasets (by at least 100k samples). It is possible that HKRR generally performs better on such dataset sizes, or that the optimal hyperparameters for HJZ change significantly in this larger-sample regime.

Understanding why HJZ may be more *stable* to hyperparameter choices (see **Observation 5** in Section 3) is a more challenging question to answer, since it likely has to do with internal game dynamics in the learning algorithm. In particular, by choosing various online learning algorithms, HJZ implements a family of multicalibration post-processing methods. We test all algorithms from this family with varying parameters. It is possible that the family of algorithms itself somehow has a shift in stability as we scale to a large data regime (4). However, as analyzing even a singular algorithm (e.g. HKRR) is challenging, we are not sure that speculating about the stability of a family of algorithms is currently possible, and hence, leave this to future work.

## D   Broader Impacts

Our work performs a comprehensive empirical evaluation of multicalibration post-processing, which could help practitioners apply these notions more effectively in practice. We note however, that fairness can be subtle, and multicalibration by itself may not be enough to ensure fairness. By now it is well understood that there are tradeoffs between different notions of fairness, and the right definitions to be deployed are context-dependent and depend on societal norms. Therefore, our results on the latent multicalibration of neural networks should not be construed as implying that these models are already fair, and care should be taken before deploying any ML model in applications with consequential societal outcomes.

## E   Dataset and Subgroup Descriptions

Here, we detail the datasets and group information used in all experiments.

### E.1   Tabular Datasets

The ACS Income dataset, introduced by Ding et al. (2021), is a superset of the UCI Adult[8] dataset (Becker and Kohavi, 1996) derived from additional US Census data. We use the `folktables` package introduced alongside the work. In particular, we consider the task of predicting whether an American adult living in California receives income greater than $50,000 in the year 2018. Features include race, gender, age, and occupation. For this task, the dataset furnishes just under 200,000 samples.

The UCI Bank Marketing dataset documents 45,000 phone calls made by a Portuguese banking institution over the course of several marketing campaigns (Moro et al., 2012). We consider the task of predicting whether, on a given call, the client will subscribe a term deposit, given features characterizing the housing, occupation, education, and age of the client.

The UCI Default of Credit Card Clients dataset (termed "Credit Default" in our experiments) documents the partial credit histories of 30,000 Taiwanese individuals (Yeh, 2016). We consider the task of predicting whether an individual will default on credit card debt, given payment history and additional identity attributes.

The HMDA (Home Mortgage Disclosure Act) dataset documents the US mortgage applications, identity attributes of associated applicants, and the outcome of these applications (Federal Financial Institutions Examination Council, 2017). We use a 114,000-sample variant of this dataset given by Cooper et al. (2023), and consider the task of predicting whether a 2017 application in the state of Texas was accepted.

The MEPS (Medical Expenditure Panel Survey) dataset comes from the US Department of Health and Human Services and documents healthcare utilization of US households. We use a 11,000-sample variant of the dataset, originally studied in Sharma et al. (2021), and consider the task of predicting whether a household makes at least 10 medical visits, given socioeconomic and geographic information of household applicants.

### E.2   Image Datasets

The CelebA dataset, introduced by Liu et al. (2015), consists of 200,000 cropped and aligned images of celebrity faces. We consider the task of predicting hair color, a task known to be difficult for certain label-dependent subgroups due to the existence of spurious correlations (Sagawa et al., 2019). Metadata documents certain characteristics of the individuals in the images such as gender, face shape, hair style, and the presence of fashion accessories.

The Camelyon17 dataset, introduced by Bándi et al. (2019), consists of histopathological images of human lymph node tissue. We use a patch-based variant of this dataset, introduced by Koh et al. (2021), which consists of 450,000 96x96 images. Unlike Koh et al. (2021), we shuffle the predetermined training and test splits. We consider the task of predicting whether a given image contains tumorous tissue. Metadata documents the hospital from which a given patch originates, and the original slide from which the patch is drawn.

### E.3   Language Datasets

The CivilComments dataset, introduced by Borkan et al. (2019), contains 450,000 online comments annotated for toxicity and identity mentions by crowdsourcing and majority vote. We use the WILDS

---

[8]For comparison with prior work, we include the original UCI Adult dataset in our benchmark repository.

variant of this dataset, provided by Koh et al. (2021), though we shuffle the predetermined training and test splits, and consider the task of prediction whether a given comment is labeled toxic.

The Amazon Polarity dataset, also introduced by Zhang et al. (2015) and a subset of the Amazon Reviews dataset, provides the text content of 4,000,000 Amazon reviews. A review receives the label 1 when associated with a rating greater than or equal to 4 stars, and a label of 0 when associated with a rating of less than or equal to 2 stars. As with Yelp Polarity, this dataset comes with no metadata, so we define groups based on the presence of meaningful words. We use a randomly-drawn, 400,000 sample subset across all experiments.

### E.4 Groups for Tabular Datasets

Here we present the subgroups considered for each dataset in our experiments. In all cases, we only consider subgroups composing at least a 0.005-fraction of the underlying dataset.

Note that the 'Dataset' row in each table does not correspond to a group used in multicalibration post-processing, nor are aggregate metrics used to compute worst-group metrics such as max smECE. We include this row for convenience.

| group name | n samples | fraction | y mean |
|---|---|---|---|
| Black Adults | 8508 | 0.0435 | 0.3461 |
| Black Females | 4353 | 0.0222 | 0.3193 |
| Women | 92354 | 0.4720 | 0.3491 |
| Never Married | 68408 | 0.3496 | 0.2344 |
| American Indian | 1294 | 0.0066 | 0.2836 |
| Seniors | 14476 | 0.0740 | 0.5410 |
| White Women | 55856 | 0.2855 | 0.3729 |
| Multiracial | 8206 | 0.0419 | 0.3572 |
| Asian | 32709 | 0.1672 | 0.4805 |
| Dataset | 195665 | 1.0000 | 0.4106 |

Figure 6: ACS Income groups.

| group name | n samples | fraction | y mean |
|---|---|---|---|
| Job = Management | 9458 | 0.2092 | 0.1376 |
| Job = Technician | 7597 | 0.1680 | 0.1106 |
| Job = Entrepreneur | 1487 | 0.0329 | 0.0827 |
| Job = Blue-Collar | 9732 | 0.2153 | 0.0727 |
| Job = Retired | 2264 | 0.0501 | 0.2279 |
| Marital = Married | 27214 | 0.6019 | 0.1012 |
| Marital = Single | 12790 | 0.2829 | 0.1495 |
| Education = Primary | 6851 | 0.1515 | 0.0863 |
| Education = Secondary | 23202 | 0.5132 | 0.1056 |
| Education = Tertiary | 13301 | 0.2942 | 0.1501 |
| Housing = Yes | 25130 | 0.5558 | 0.0770 |
| Housing = No | 20081 | 0.4442 | 0.1670 |
| Age < 30 | 3050 | 0.0675 | 0.1951 |
| 30 ≤ Age < 40 | 17359 | 0.3840 | 0.1129 |
| Age ≥ 50 | 12185 | 0.2695 | 0.1287 |
| Dataset | 45211 | 1.0000 | 0.1170 |

Figure 7: Bank Marketing groups.

| group name | n samples | fraction | y mean |
|---|---|---|---|
| Male, Age $< 30$ | 3281 | 0.1094 | 0.2405 |
| Single | 15964 | 0.5321 | 0.2093 |
| Single, Age $> 30$ | 6888 | 0.2296 | 0.1992 |
| Female | 18112 | 0.6037 | 0.2078 |
| Married, Age $< 30$ | 1482 | 0.0494 | 0.2611 |
| Married, Age $> 60$ | 225 | 0.0075 | 0.2667 |
| Education $=$ High School | 4917 | 0.1639 | 0.2516 |
| Education $=$ High School, Married | 2861 | 0.0954 | 0.2635 |
| Education $=$ High School, Age $> 40$ | 2456 | 0.0819 | 0.2577 |
| Education $=$ University, Age $< 25$ | 1610 | 0.0537 | 0.2795 |
| Female, Education $=$ University | 8656 | 0.2885 | 0.2220 |
| Education $=$ Graduate School | 10585 | 0.3528 | 0.1923 |
| Female, Education $=$ Graduate School | 6231 | 0.2077 | 0.1814 |
| Dataset | 30000 | 1.0000 | 0.2212 |

Figure 8: Credit Default groups.

| group name | n samples | fraction | y mean |
|---|---|---|---|
| Applicant Ethnicity: Hispanic or Latino | 26416 | 0.2313 | 0.6806 |
| Applicant Ethnicity: Not Hispanic or Latino | 73527 | 0.6439 | 0.7940 |
| Applicant Ethnicity: Not provided | 14128 | 0.1237 | 0.6704 |
| Applicant Sex: Female | 32143 | 0.2815 | 0.7319 |
| Applicant Sex: Male | 72635 | 0.6361 | 0.7713 |
| Co-Applicant Sex: Female | 35164 | 0.3080 | 0.8029 |
| Co-Applicant Sex: Male | 10336 | 0.0905 | 0.7767 |
| Applicant Race: Black | 9044 | 0.0792 | 0.6703 |
| Applicant Race: Asian | 8086 | 0.0708 | 0.8097 |
| Applicant Race: Native American or Alaskan | 1019 | 0.0089 | 0.5927 |
| Co-Applicant Race: Black | 2760 | 0.0242 | 0.7120 |
| Co-Applicant Race: Asian | 3339 | 0.0292 | 0.8194 |
| Dataset | 114185 | 1.0000 | 0.7524 |

Figure 9: HMDA groups.

| group name | n samples | fraction | y mean |
|---|---|---|---|
| Age 0-18 | 3308 | 0.2986 | 0.0605 |
| Age 19-34 | 2468 | 0.2228 | 0.1021 |
| Age 35-50 | 2186 | 0.1973 | 0.1404 |
| Age 51-64 | 1813 | 0.1636 | 0.2670 |
| Age 65-79 | 977 | 0.0882 | 0.4637 |
| Not White | 7121 | 0.6427 | 0.1227 |
| Northeast | 1553 | 0.1402 | 0.2260 |
| Midwest | 2020 | 0.1823 | 0.2040 |
| South | 4325 | 0.3904 | 0.1487 |
| West | 3181 | 0.2871 | 0.1481 |
| Poverty Category 1 | 2435 | 0.2198 | 0.1577 |
| Poverty Category 2 | 704 | 0.0635 | 0.1378 |
| Poverty Category 3 | 1941 | 0.1752 | 0.1484 |
| Poverty Category 4 | 3100 | 0.2798 | 0.1519 |
| Dataset | 11079 | 1.0000 | 0.1694 |

Figure 10: MEPS groups.

## E.5 Alternate Groups for Tabular Datasets

To validate our observations on tabular data, we repeated all experiments on each tabular dataset with an alternate collection of groups. Like the original groups, these groups are defined by a feature or conjunction of two features. We provide the alternate group definitions here, and present results on these groups in Appendix I.

| group name | n samples | fraction | y mean |
|---|---|---|---|
| Associates Degree Male | 7331 | 0.0375 | 0.4957 |
| Associates Degree Female | 8372 | 0.0428 | 0.3186 |
| Divorced Female | 10652 | 0.0544 | 0.4415 |
| Under Part Time | 16525 | 0.0845 | 0.1025 |
| Part Time | 55269 | 0.2825 | 0.1408 |
| Full Time | 135989 | 0.6950 | 0.5214 |
| Over Full Time | 11471 | 0.0586 | 0.6441 |
| Not White | 74659 | 0.3816 | 0.3574 |
| Government Employee | 29121 | 0.1488 | 0.5337 |
| Private Employee | 166544 | 0.8512 | 0.3890 |
| Under 21 | 10166 | 0.0520 | 0.0106 |
| Middle Aged | 81582 | 0.4169 | 0.5064 |
| Dataset | 195665 | 1.0000 | 0.4106 |

Figure 11: ACS Income alternate groups.

| group name | n samples | fraction | y mean |
|---|---|---|---|
| Job = Management, Age < 50 | 7091 | 0.1568 | 0.1393 |
| Job = Technician, Age < 30 | 436 | 0.0096 | 0.1858 |
| Job = Blue-Collar, Age > 50 | 1075 | 0.0238 | 0.0679 |
| Married, Education = Primary | 5246 | 0.1160 | 0.0755 |
| Single, Education = Tertiary | 4792 | 0.1060 | 0.1836 |
| Housing = Yes, Age < 30 | 1621 | 0.0359 | 0.0993 |
| Housing = No, Age < 30 | 1429 | 0.0316 | 0.3037 |
| Under 21 | 305 | 0.0067 | 0.3115 |
| Middle Age | 23841 | 0.5273 | 0.0977 |
| Senior Age | 961 | 0.0213 | 0.4225 |
| Dataset | 45211 | 1.0000 | 0.1170 |

Figure 12: Bank Marketing alternate groups.

| group name | n samples | fraction | y mean |
|---|---|---|---|
| Single, Male | 6553 | 0.2184 | 0.2266 |
| Single, Female | 15964 | 0.5321 | 0.2093 |
| Young Adult | 9618 | 0.3206 | 0.2284 |
| Middle Aged | 8872 | 0.2957 | 0.2356 |
| Education = High School, Female | 2927 | 0.0976 | 0.2364 |
| Education = University, Female | 8656 | 0.2885 | 0.2220 |
| Education = High School, Single | 1909 | 0.0636 | 0.2368 |
| Education = High School, Married | 2861 | 0.0954 | 0.2635 |
| Education = University, Single | 7020 | 0.2340 | 0.2306 |
| Education = University, Married | 6842 | 0.2281 | 0.2435 |
| Education = Graduate, Single | 6809 | 0.2270 | 0.1842 |
| Dataset | 30000 | 1.0000 | 0.2212 |

Figure 13: Credit Default alternate groups.

| group name | n samples | fraction | y mean |
|---|---|---|---|
| Loan Type 1 | 87857 | 0.7694 | 0.7327 |
| Loan Type 2 | 17587 | 0.1540 | 0.8125 |
| Loan Type 3 | 8047 | 0.0705 | 0.8311 |
| HUD Median Family Income > 50k | 107450 | 0.9410 | 0.7583 |
| HUD Median Family Income ≤ 50k | 6735 | 0.0590 | 0.6578 |
| Has Co-Applicant | 42535 | 0.3725 | 0.8079 |
| Agency = OCC | 5966 | 0.0522 | 0.8235 |
| Agency = FRS | 3879 | 0.0340 | 0.8729 |
| Agency = FDIC | 6951 | 0.0609 | 0.8708 |
| Agency = NCUA | 10626 | 0.0931 | 0.6882 |
| Agency = HUD | 64915 | 0.5685 | 0.7562 |
| Agency = CFPB | 21848 | 0.1913 | 0.6939 |
| Loan Type = 1 to 4 Family | 87857 | 0.7694 | 0.7327 |
| Loan Type = Manufactured Housing | 17587 | 0.1540 | 0.8125 |
| Loan Type = Multi-Family | 8047 | 0.0705 | 0.8311 |
| Dataset | 114185 | 1.0000 | 0.7524 |

Figure 14: HMDA alternate groups.

| group name | n samples | fraction | y mean |
|---|---|---|---|
| Under 21 | 3772 | 0.3405 | 0.0607 |
| Middle Age | 2874 | 0.2594 | 0.1990 |
| Senior Age | 1304 | 0.1177 | 0.4862 |
| Sex = 1 | 5281 | 0.4767 | 0.1274 |
| Sex = 2 | 5798 | 0.5233 | 0.2077 |
| White | 3958 | 0.3573 | 0.2534 |
| Active Duty Group 2 | 6454 | 0.5825 | 0.1432 |
| Marriage Group 1 | 3645 | 0.3290 | 0.2222 |
| Marriage Group 2 | 450 | 0.0406 | 0.4867 |
| Pregnancy Group 1 | 124 | 0.0112 | 0.4274 |
| Pregnancy Group 2 | 2167 | 0.1956 | 0.1398 |
| Insurance Group 1 | 5926 | 0.5349 | 0.1790 |
| Insurance Group 2 | 3890 | 0.3511 | 0.1979 |
| Dataset | 11079 | 1.0000 | 0.1694 |

Figure 15: MEPS alternate groups.

## E.6  Groups for Image Datasets

| group name | n samples | fraction | y mean |
|---|---|---|---|
| Male | 84434 | 0.4168 | 0.0207 |
| Female | 118165 | 0.5832 | 0.2389 |
| Arched Eyebrows | 54090 | 0.2670 | 0.2227 |
| Bangs | 30709 | 0.1516 | 0.2310 |
| Big Lips | 48785 | 0.2408 | 0.1629 |
| Chubby | 11663 | 0.0576 | 0.0189 |
| Double Chin | 9459 | 0.0467 | 0.0248 |
| Eyeglasses | 13193 | 0.0651 | 0.0392 |
| High Cheekbones | 92189 | 0.4550 | 0.1949 |
| Mouth Slightly Open | 97942 | 0.4834 | 0.1737 |
| Oval Face | 57567 | 0.2841 | 0.1761 |
| Pale Skin | 8701 | 0.0429 | 0.2455 |
| Receding Hairline | 16163 | 0.0798 | 0.0630 |
| Smiling | 97669 | 0.4821 | 0.1812 |
| Straight Hair | 42222 | 0.2084 | 0.1518 |
| Wavy Hair | 64744 | 0.3196 | 0.2145 |
| Wearing Hat | 9818 | 0.0485 | 0.0168 |
| Young | 156734 | 0.7736 | 0.1581 |
| Dataset | 202599 | 1.0000 | 0.1480 |

Figure 16: CelebA groups.

| group name | n samples | fraction | y mean |
|---|---|---|---|
| Hospital = 0 | 59436 | 0.1304 | 0.5000 |
| Hospital = 1 | 34904 | 0.0766 | 0.5000 |
| Hospital = 2 | 85054 | 0.1865 | 0.5000 |
| Hospital = 3 | 129838 | 0.2848 | 0.5000 |
| Hospital = 4 | 146722 | 0.3218 | 0.5000 |
| Slide = 0 | 4316 | 0.0095 | 0.0083 |
| Slide = 4 | 7294 | 0.0160 | 0.6697 |
| Slide = 8 | 13455 | 0.0295 | 0.6469 |
| Slide = 16 | 4971 | 0.0109 | 0.8236 |
| Slide = 20 | 3810 | 0.0084 | 0.0071 |
| Slide = 24 | 7727 | 0.0169 | 0.0238 |
| Slide = 28 | 31878 | 0.0699 | 0.8469 |
| Slide = 32 | 8831 | 0.0194 | 0.2466 |
| Slide = 36 | 10661 | 0.0234 | 0.0015 |
| Slide = 40 | 7395 | 0.0162 | 0.0170 |
| Slide = 44 | 7958 | 0.0175 | 0.0030 |
| Slide = 48 | 61110 | 0.1340 | 0.9273 |
| Dataset | 455954 | 1.0000 | 0.5000 |

Figure 17: Camelyon17 groups.

## E.7    Groups for Language Datasets

| group name | n samples | fraction | y mean |
|---|---|---|---|
| Male | 20880 | 0.0466 | 0.1488 |
| Female | 33113 | 0.0739 | 0.1399 |
| LGBTQ | 14303 | 0.0319 | 0.2684 |
| Not LGBTQ | 433695 | 0.9681 | 0.1083 |
| Christian | 18961 | 0.0423 | 0.1103 |
| Not Christian | 380222 | 0.8487 | 0.1177 |
| Muslim | 13939 | 0.0311 | 0.2429 |
| Not Muslim | 418737 | 0.9347 | 0.1065 |
| Other Religions | 11030 | 0.0246 | 0.1528 |
| Black | 8448 | 0.0189 | 0.3638 |
| Not Black | 426444 | 0.9519 | 0.1049 |
| White | 14339 | 0.0320 | 0.3068 |
| Not White | 415090 | 0.9265 | 0.1016 |
| Dataset | 447998 | 1.0000 | 0.1134 |

Figure 18: Civil Comments groups.

| group name | n samples | fraction | y mean |
|---|---|---|---|
| expensive | 5834 | 0.0146 | 0.4434 |
| cheap | 10928 | 0.0273 | 0.2753 |
| food | 3868 | 0.0097 | 0.5476 |
| health | 2381 | 0.0060 | 0.6237 |
| music | 26463 | 0.0662 | 0.6192 |
| book | 108100 | 0.2703 | 0.5289 |
| movie | 36191 | 0.0905 | 0.4731 |
| tech | 8515 | 0.0213 | 0.4547 |
| exercise | 2262 | 0.0057 | 0.5535 |
| garbage | 3248 | 0.0081 | 0.0702 |
| terrible | 6138 | 0.0153 | 0.0893 |
| incredible | 2532 | 0.0063 | 0.7986 |
| love | 53005 | 0.1325 | 0.7212 |
| again | 26106 | 0.0653 | 0.4454 |
| star | 42632 | 0.1066 | 0.4495 |
| Dataset | 399980 | 1.0000 | 0.5009 |

Figure 19: Amazon Polarity groups.

## E.8    Dataset Usage and Licensing

- **ACSIncome**: While Folktables provides API for downloading ACS data, usage of this data is governed by the terms of use provided by the Census Bureau. For more information, see https://www.census.gov/data/developers/about/terms-of-service.html.

- **BankMarketing**: Creative Commons Attribution 4.0 International (CC BY 4.0)

- **CreditDefault**: Creative Commons Attribution 4.0 International (CC BY 4.0)

- **HMDA**: The variant we use is available for download on https://github.com/pasta41/hmda?tab=readme-ov-file under an MIT license.

- **MEPS**: The variant we use is available for download on https://github.com/alangee/FaiR-N/tree/master under an Apache 2.0 license.

- **Civil Comments**: This dataset is in the public domain and distributed under CC0.

- **Amazon Polarity**: We were not able to find a license for this dataset. It is a downstream variant of a dataset generated with content from Internet Archive[9].

---

[9] http://archive.org/details/asin_listing/

- **CelebA**: The creators of this dataset do not provide a license, though they encourage its use for non-commercial research purposes only.

- **Camelyon17**: This dataset is in the public domain and distributed under CC0.

# F  Hyperparameters for Multicalibration and Calibration Algorithms

Here, we detail the hyperparameters with which equip algorithms from Haghtalab et al. (2023) and Hébert-Johnson et al. (2018), as well as the hyperparameters used in standard calibration methods.

### F.1  Hébert-Johnson et al. (2018) Algorithm

These authors do not report empirical performance of their algorithm. For consistency with calibration measures, we fix a bin width of $\lambda = 0.1$ in all experiments. The algorithm depends on one other hyperparameter $\alpha$, which is some constant factor of the acceptable difference between mean prediction and mean label within each "category," a subgroup $g \subseteq \mathcal{X}$ restricted to the preimage of a particular bin $b \subseteq [0, 1]$. Modulo the randomness induced by a statistical query oracle, this algorithm converges when violations within each category are sufficiently small. As originally proposed, we skip categories $g \cap f^{-1}(b)$ in iterations where $|g \cap f^{-1}(b)| \leq \lambda\alpha|g|$. For each dataset and each base predictor, we sweep over $\alpha \in \{0.1, 0.05, 0.025, 0.0125\}$.

### F.2  Haghtalab et al. (2023) Algorithms

These authors present an empirical examination in conjunction with theoretical results, though our use of the algorithms differs significantly. Namely, instead of initializing predictions uniformly, we initialize with the predictions of some base predictor. The authors also train their multicalibration algorithms with substantially larger collections of subgroups, in some cases defining group by all unique values of individual features. Instead, we only consider a collection of at most 20 "protected" groups.

The authors evaluate six algorithms: four based on "no-regret best-response dynamics," using an empirical risk minimizer as the adversary and Hedge, Prod, Optimistic Hedge, or Gradient Descent as the learner; two based on "no-regret no-regret dynamics," using either Hedge or Optimistic Hedge as both the adversary and learner. On each task, and with each algorithm, the authors train for 50-100 iterations and sweep over learning rate decay rates of $\eta \in \{0.8, 0.85, 0.9, .95\}$ for the learner and, when applicable, rates of $\eta \in \{0.9, 0.95, 0.98, 0.99\}$ for the adversary. On two of the three datasets they examine, the authors fix a bin-width of $\lambda = 0.1$.

In all experiments, we consider the same multicalibration algorithms but restrict to 30 iterations and a smaller collection of decay rates. In particular, for each dataset and base predictor, and for each of the six algorithms, we sweep over decay rates $\eta \in \{0.9, 0.95\}$ for the learner and $\eta \in \{0.9, 0.95, 0.98\}$ for the adversary. We justify these restrictions by noting that (1) our base predictors already achieve nontrivial accuracy on each task and (2) that this sweep covers a large portion of the optimal hyperparameters found in Haghtalab et al. (2023). For consistency with our hyperparameters for HKRR and chosen calibration measures, we fix $\lambda = 0.1$ on all datasets.

### F.3  Temperature Implementation

Temperature scaling can be made hyperparameter-free by choosing a divisor $T^*$ which minimizes the cross-entropy loss on a held-out calibration split. One can also fix $T$ to some positive real number. For each dataset and base predictor, we examine both methods, scaling logits by $1/T$ for all $T \in \{0.2 \cdot k : k \in [20]\}$, as well as by $1/T^*$ with $T^*$ obtained via the Pytorch implementation of L-BFGS. We report only the best temperature scaling method on a hold-out validation set.

# G  Models and Training

Here, we describe the models used as base predictors and their hyperparameters (or the procedure for obtaining these hyperparameters) in all experiments. Across all datasets, we use a train-validation-test split of $(0.6 : 0.2 : 0.2)$, fixing the test set and determining train/validation sets via random seed. In all cases of a base-predictor hyperparameter search, we use validation accuracy to select hyperparameters.

## G.1 Models Used on Tabular Data

On all tabular datasets, we examine five standard prediction models from supervised learning: Decision Tree, Random Forest, Logistic Regression, SVM, and Naive Bayes, using a `Scikit-learn` implementation in all cases. We also examine MLPs of varying architecture. For each dataset and prediciton model, we examine all calibration fractions $\text{CF} \in \{0, 0.01, 0.05, 0.1, 0.2, 0.4, 0.6, 0.8, 1.0\}$. When $\text{CF} = 1.0$, we take the base predictor to output $1/2$ for all samples. During both hyperparameter search and test-set evaluation we average all metrics over five random splits of the training and validation data.

### G.1.1 Standard Supervised Learning Models

For Decision Tree, we vary maximum depth over $\{\texttt{None}, 10, 20, 50\}$ and the minimum number of samples required to split an internal node over $\{2, 5, 10\}$. For Random Forest, we vary these same hyperparameters and fix the number of estimators at 100. For Logistic Regression and SVM, we vary regularization strength; for Logistic Regression we let the inverse regularization strength $C \in \{0.4, 1, 2, 4\}$ and for SVM we let the regularization strength $\alpha \in \{0.00001, 0.0001, 0.001, 0.01\}$. Naive Bayes is hyperparameter-free.

While Decision Tree, Random Forest, Logistic Regression, and Naive Bayes are naturally probabilistic, SVM is not. While `Scikit-learn` provides a probabilistic prediction method with the `predict_proba()` function, this is implemented via Platt scaling of the SVM scores. For this reason, we treat the SVM's standard output labels as probabilities. For efficient on larger datasets, we also use the `SGDClassifier` implementation of SVM.

### G.1.2 MLPs

To reduce computation during MLP hyperparameter search, for each dataset we constructed a smaller set of hyperparameters over which to sweep, based on what yielded the best performance in preliminary experiments. In what follows, we let a sequence $(\ell_i)_{i=1}^{N}$ denote the ordered layer widths for a particular MLP with $N$ hidden layers. When we substitute some $\ell_i$ with BN, this indicates the presence of a batch-normalization layer. In all experiments with MLPs on tabular datasets, we use the Adam optimizer.

On ACS Income, we train for 50 epochs. We search over hidden-layer widths: $(128, \text{BN}, 128)$, $(128, 256, 128)$, and $(128, \text{BN}, 256, \text{BN}, 128)$. We vary batch size over $\{32, 64, 128\}$ and learning rate over $\{0.01, 0.001, 0.0001, 0.00001\}$.

On Bank Marketing, we train for 50 epochs. We search over hidden-layer widths of $(100)$, $(128, \text{BN}, 128)$, $(128, 256, 128)$, and $(128, \text{BN}, 256, \text{BN}, 128)$. We vary batch size over $\{64, 128, 256, 512\}$ and learning rate over $\{0.001, 0.0001, 0.00001\}$. We also include a learning rate schedule under which our learning rate is $0.00005$ for the first five epochs and $0.00001$ for the remaining.

On Credit Default, we train for 5 epochs. We search over hidden-layer widths of $(100)$, $(128, 256, 128)$, and $(128, \text{BN}, 256, \text{BN}, 128)$. We vary batch size over $\{16, 32, 64, 128\}$ and learning rate over $\{0.01, 0.001, 0.0001, 0.00001\}$.

On HMDA, we train for 30 epochs. We search over hidden-layer widths of $(100)$, $(128, \text{BN}, 128)$, $(128, 256, 128)$, and $(128, \text{BN}, 256, \text{BN}, 128)$. We vary batch size over $\{128, 256, 512\}$ and learning rate over $\{0.001, 0.0001, 0.00001\}$. We also include a learning rate schedule under which our learning rate is $0.00005$ for the first five epochs and $0.00001$ for the remaining. In addition, we search over weight decays in $\{0, 0.0001, 0.00001\}$.

On MEPS, we train for 50 epochs. We search over hidden-layer widths of $(100)$, $(128, \text{BN}, 128)$, $(128, 256, 128)$, and $(128, \text{BN}, 256, \text{BN}, 128)$. We vary batch size over $\{16, 32, 64\}$ and learning rate over $\{0.1, 0.01, 0.01, 0.001, 0.0001, 0.00001\}$. In addition, we search over weight decays in $\{0, 0.0001, 0.00001\}$.

To ensure all NN outputs are probabilistic, we apply the softmax function before evaluating predictions or passing into any post-processing algorithm. The only exception to this rule is when we apply temperature scaling, which scales raw logits *before* passing into softmax.

## G.2 Models Used on Vision and Language Tasks

On our vision and language datasets, we use much larger models, in some cases with as many as 85 million trainable parameters. We opt for hyperparameters already present in the literature, or alter-

ations of such hyperparameters which give nontrivial accuracy, and we use the same hyperparameters for each calibration fraction. We examine all calibration fractions $CF \in \{0, 0.2, 0.4\}$ and average all runs over three random splits of the training and validation data.

Our experiments with language datasets involved two models: (1) DistilBERT, a pretrained transformer introduced by Sanh et al. (2019), and (2) a ResNet-56 using unfrozen, pretrained GloVe embeddings (Pennington et al., 2014), the implementation for which comes from Duchene et al. (2023).

On the CivilComments dataset, we train a DistilBERT for 10 epochs with a batch size of 16, learning rate of 0.00001, and weight decay of 0.01, using the Adam optimizer. We fix a maximum token length of 300.

On the Amazon Polarity dataset, we train a ResNet-56 with three input channels, which accept a stacked embedding of 512 dimensions. We use the `basic_english` tokenizer provided by `torchtext`, fixing a maximum token length of 70 and minimum frequency of 5. We train for 10 epochs with a batch size of 32 and learning rate of 0.0001 using Adam. Implementation-specific details are provided in our code.

Our experiments with vision datasets involve three models: (1) `vit-large-patch32-224-in21k`, a pretrained vision transformer introduced by Dosovitskiy et al. (2021), (2) ResNet-50 (He et al., 2016), and (3) DenseNet-121 (Huang et al., 2017).

On CelebA, we train the ViT for 10 epochs with a batch size fo 64, learning rate of 0.0001, and weight decay of 0.01, using Adam. We also train a ResNet-50 for 50 epochs with a batch size of 64 and learning rate of 0.001, using SGD with a momentum of 0.9.

On Camelyon17, we train the ViT for 5 epochs with a batch size of 32, learning rate of 0.001, and weight decay of 0.01. We optimize with SGD, using a momentum of 0.9. We also train a DenseNet-121 for 10 epochs with a batch size of 32, learning rate of 0.001, and weight decay of 0.01, using SGD with a momentum of 0.9.

# H Results on Tabular Datasets

## H.1 Plots for All Multicalibration Algorithms

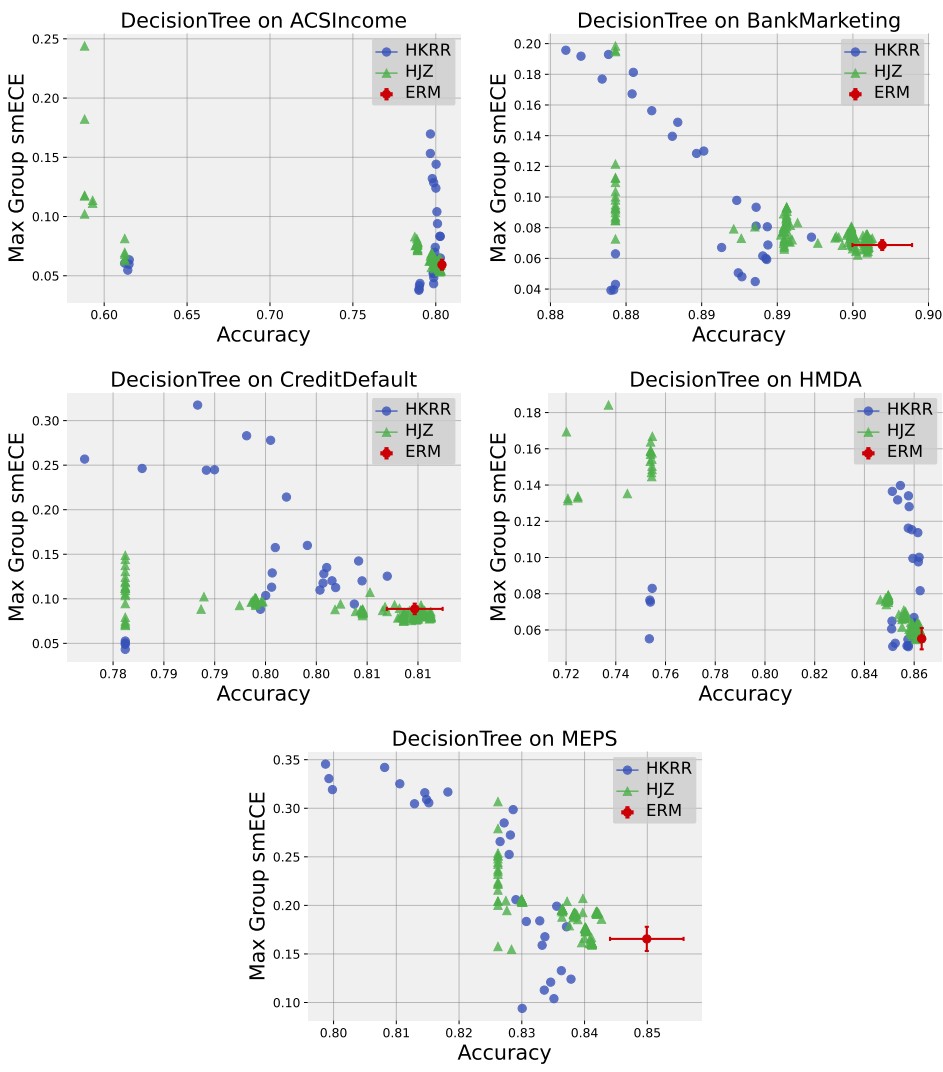

Figure 20: All multicalibration algorithms on Decision Trees.

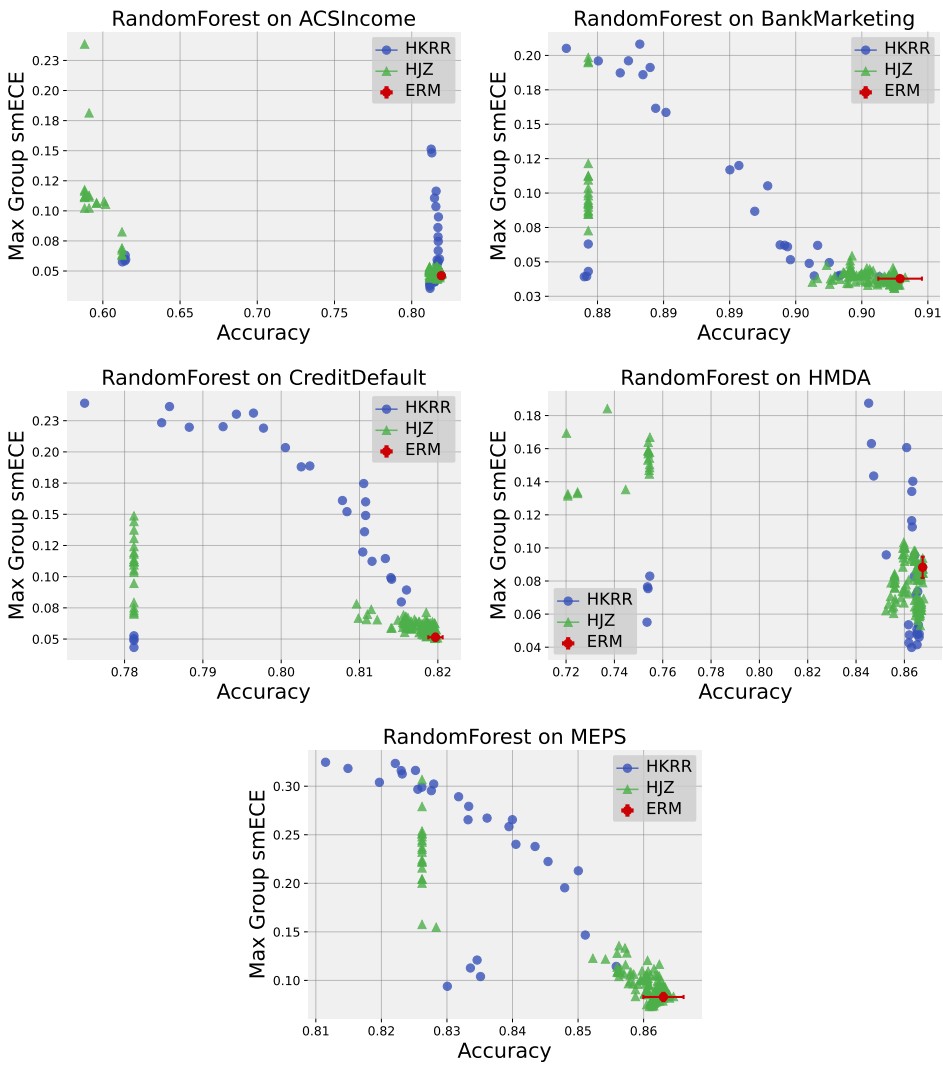

Figure 21: All multicalibration algorithms on Random Forest.

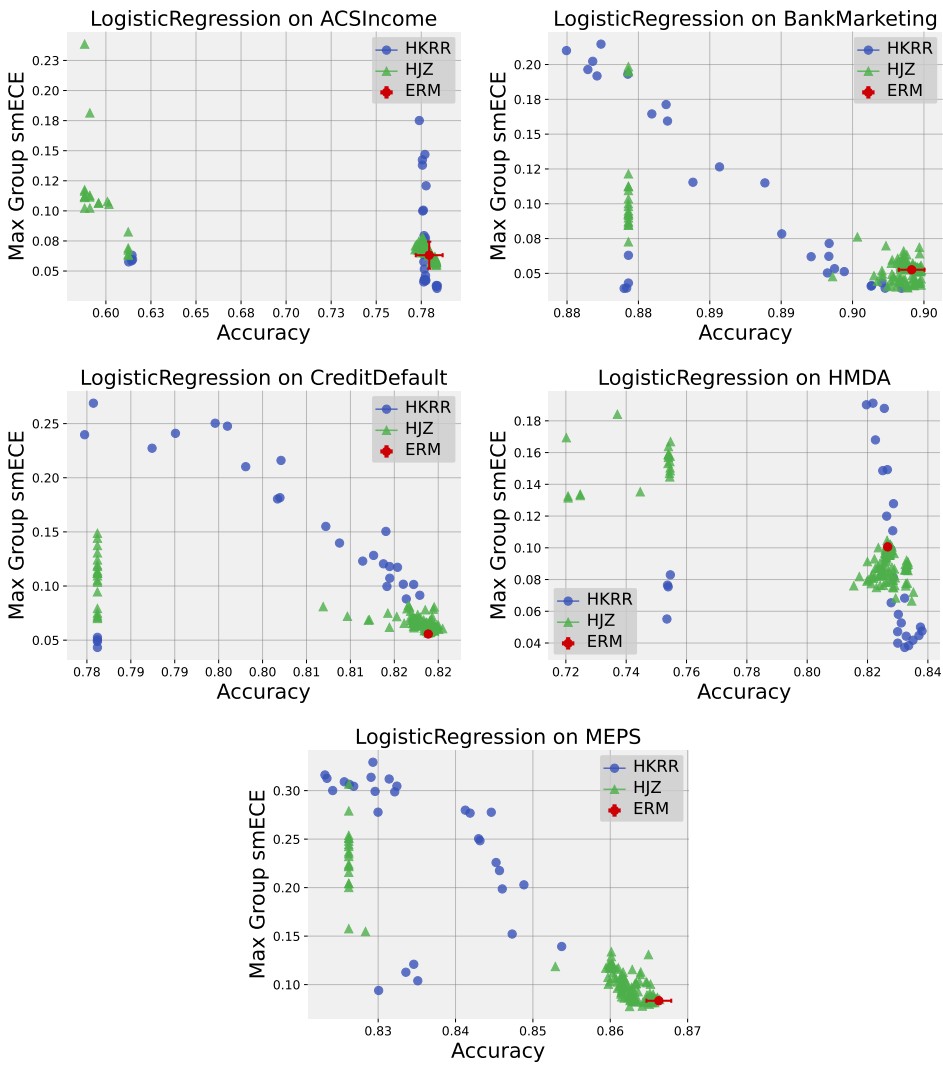

Figure 22: All multicalibration algorithms on Logistic Regression.

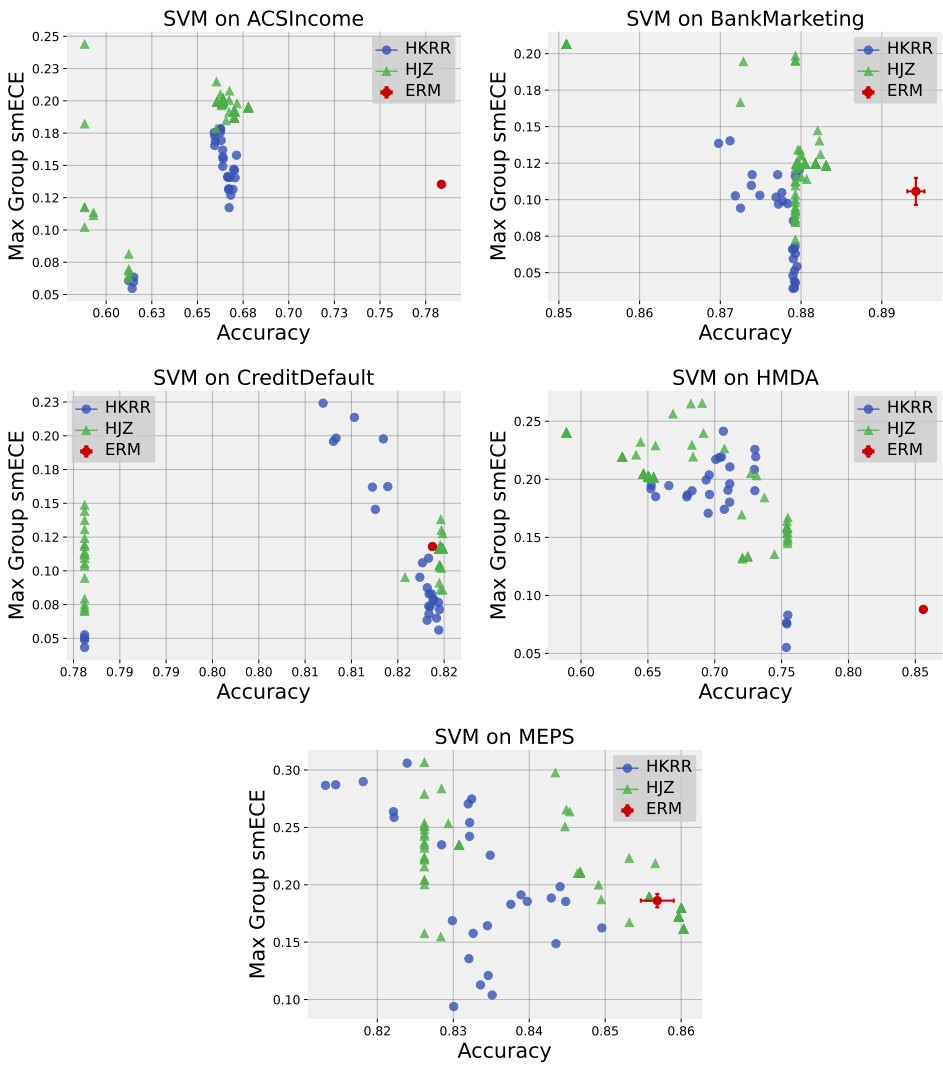

Figure 23: All multicalibration algorithms on SVMs.

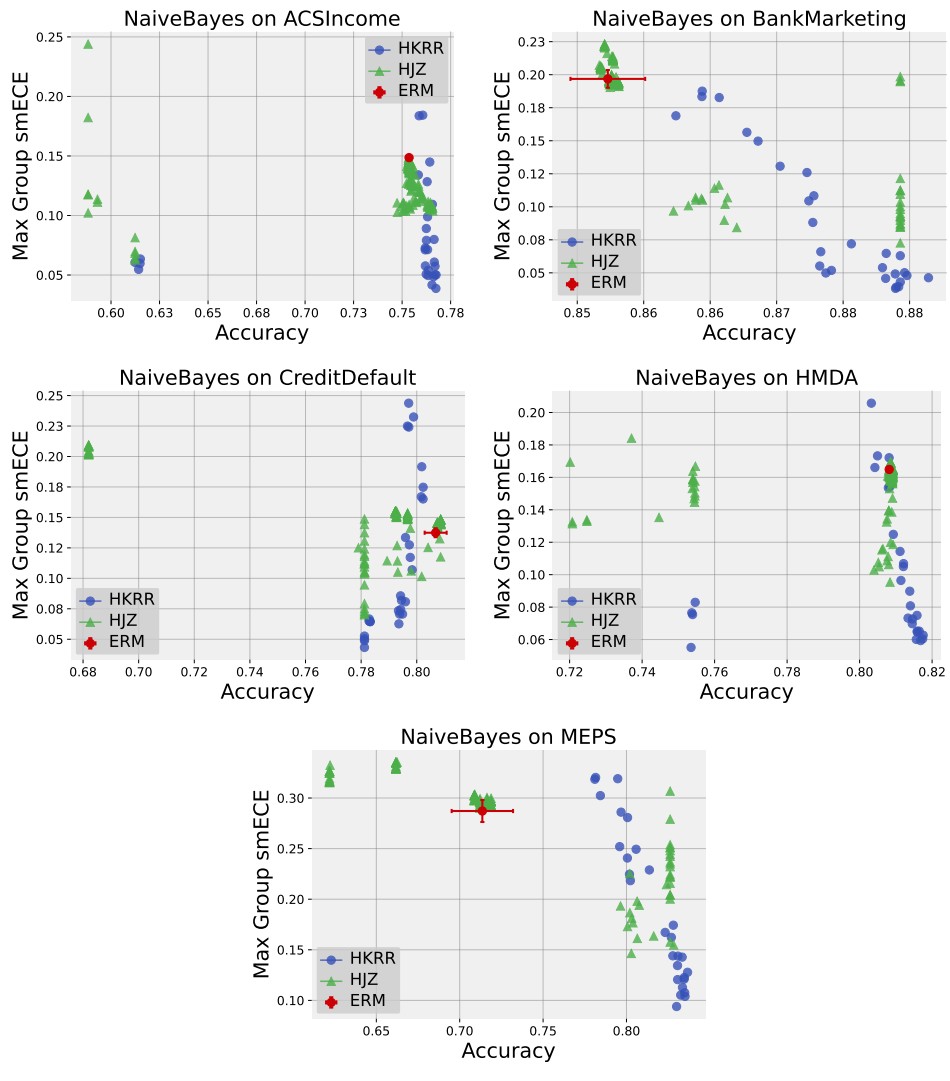

Figure 24: All multicalibration algorithms on Naive Bayes.

## H.2 Tables Comparing Best-Performing Multicalibration Algorithms with ERM

| Model | ECE ↓ | Max ECE ↓ | smECE ↓ | Max smECE ↓ | Acc ↑ |
|---|---|---|---|---|---|
| MLP ERM | 0.01 ± 0.003 | 0.069 ± 0.011 | 0.012 ± 0.003 | 0.058 ± 0.005 | **0.812 ± 0.001** |
| MLP HKRR | 0.023 ± 0.001 | **0.065 ± 0.004** | 0.023 ± 0.001 | 0.063 ± 0.002 | 0.615 ± 0.0 |
| MLP HJZ | 0.01 ± 0.002 | 0.069 ± 0.008 | 0.013 ± 0.001 | **0.055 ± 0.004** | 0.81 ± 0.002 |
| MLP Platt | 0.017 ± 0.009 | 0.076 ± 0.008 | 0.018 ± 0.008 | 0.064 ± 0.008 | 0.809 ± 0.003 |
| MLP Temp | 0.011 ± 0.005 | 0.068 ± 0.01 | 0.013 ± 0.004 | 0.059 ± 0.007 | 0.811 ± 0.001 |
| MLP Isotonic | **0.01 ± 0.001** | 0.067 ± 0.008 | **0.011 ± 0.001** | 0.057 ± 0.002 | 0.811 ± 0.001 |
| RandomForest ERM | 0.01 ± 0.001 | **0.051 ± 0.01** | 0.011 ± 0.0 | **0.046 ± 0.002** | **0.819 ± 0.001** |
| RandomForest HKRR | 0.023 ± 0.001 | 0.066 ± 0.004 | 0.023 ± 0.001 | 0.063 ± 0.002 | 0.614 ± 0.001 |
| RandomForest HJZ | 0.007 ± 0.002 | 0.052 ± 0.003 | 0.01 ± 0.001 | 0.047 ± 0.003 | 0.818 ± 0.001 |
| RandomForest Platt | **0.006 ± 0.001** | 0.054 ± 0.001 | **0.01 ± 0.001** | 0.047 ± 0.002 | 0.818 ± 0.001 |
| RandomForest Temp | 0.027 ± 0.001 | 0.074 ± 0.008 | 0.027 ± 0.0 | 0.061 ± 0.004 | 0.819 ± 0.001 |
| RandomForest Isotonic | 0.008 ± 0.001 | 0.059 ± 0.011 | 0.011 ± 0.0 | 0.048 ± 0.004 | 0.818 ± 0.001 |
| SVM ERM | 0.216 ± 0.001 | 0.268 ± 0.002 | 0.109 ± 0.0 | 0.135 ± 0.001 | **0.784 ± 0.001** |
| SVM HKRR | 0.023 ± 0.001 | **0.065 ± 0.004** | 0.023 ± 0.001 | **0.063 ± 0.002** | 0.615 ± 0.0 |
| SVM HJZ | 0.03 ± 0.002 | 0.074 ± 0.002 | 0.026 ± 0.002 | 0.068 ± 0.006 | 0.612 ± 0.0 |
| SVM Platt | 0.336 ± 0.007 | 0.403 ± 0.003 | 0.168 ± 0.004 | 0.2 ± 0.001 | 0.664 ± 0.007 |
| SVM Temp | **0.022 ± 0.005** | 0.117 ± 0.005 | **0.022 ± 0.005** | 0.117 ± 0.005 | 0.678 ± 0.006 |
| SVM Isotonic | 0.081 ± 0.012 | 0.155 ± 0.007 | 0.081 ± 0.012 | 0.146 ± 0.006 | 0.664 ± 0.007 |
| LogisticRegression ERM | 0.012 ± 0.002 | 0.065 ± 0.011 | 0.015 ± 0.002 | 0.063 ± 0.011 | 0.779 ± 0.007 |
| LogisticRegression HKRR | 0.01 ± 0.001 | **0.042 ± 0.006** | 0.01 ± 0.001 | **0.037 ± 0.002** | **0.783 ± 0.0** |
| LogisticRegression HJZ | 0.011 ± 0.001 | 0.065 ± 0.005 | 0.014 ± 0.001 | 0.057 ± 0.002 | 0.783 ± 0.001 |
| LogisticRegression Platt | 0.023 ± 0.006 | 0.08 ± 0.019 | 0.024 ± 0.006 | 0.076 ± 0.02 | 0.772 ± 0.011 |
| LogisticRegression Temp | 0.02 ± 0.001 | 0.078 ± 0.005 | 0.021 ± 0.0 | 0.072 ± 0.003 | 0.783 ± 0.0 |
| LogisticRegression Isotonic | **0.005 ± 0.001** | 0.068 ± 0.008 | **0.009 ± 0.001** | 0.066 ± 0.009 | 0.775 ± 0.009 |
| DecisionTree ERM | 0.017 ± 0.001 | 0.066 ± 0.01 | 0.016 ± 0.001 | 0.059 ± 0.004 | **0.804 ± 0.0** |
| DecisionTree HKRR | 0.023 ± 0.001 | 0.065 ± 0.004 | 0.023 ± 0.001 | 0.063 ± 0.002 | 0.615 ± 0.0 |
| DecisionTree HJZ | 0.013 ± 0.002 | 0.064 ± 0.005 | 0.013 ± 0.001 | **0.054 ± 0.005** | 0.803 ± 0.002 |
| DecisionTree Platt | 0.015 ± 0.002 | **0.058 ± 0.004** | 0.014 ± 0.002 | 0.055 ± 0.004 | 0.803 ± 0.002 |
| DecisionTree Temp | 0.029 ± 0.002 | 0.088 ± 0.009 | 0.028 ± 0.002 | 0.072 ± 0.006 | 0.803 ± 0.001 |
| DecisionTree Isotonic | **0.007 ± 0.002** | 0.072 ± 0.01 | **0.01 ± 0.001** | 0.057 ± 0.003 | 0.803 ± 0.001 |
| NaiveBayes ERM | 0.117 ± 0.0 | 0.165 ± 0.0 | 0.109 ± 0.0 | 0.149 ± 0.001 | 0.754 ± 0.0 |
| NaiveBayes HKRR | 0.023 ± 0.001 | **0.065 ± 0.004** | 0.023 ± 0.001 | **0.063 ± 0.002** | 0.615 ± 0.0 |
| NaiveBayes HJZ | 0.03 ± 0.002 | 0.074 ± 0.002 | 0.026 ± 0.002 | 0.068 ± 0.006 | 0.612 ± 0.0 |
| NaiveBayes Platt | 0.091 ± 0.004 | 0.13 ± 0.004 | 0.086 ± 0.004 | 0.12 ± 0.003 | 0.759 ± 0.001 |
| NaiveBayes Temp | 0.089 ± 0.003 | 0.154 ± 0.004 | 0.087 ± 0.002 | 0.153 ± 0.004 | 0.754 ± 0.001 |
| NaiveBayes Isotonic | **0.004 ± 0.001** | 0.094 ± 0.003 | **0.007 ± 0.0** | 0.085 ± 0.002 | **0.769 ± 0.001** |

Figure 25: ACS Income.

| Model | ECE ↓ | Max ECE ↓ | smECE ↓ | Max smECE ↓ | Acc ↑ |
|---|---|---|---|---|---|
| MLP ERM | 0.009 ± 0.004 | 0.048 ± 0.012 | 0.012 ± 0.002 | 0.046 ± 0.01 | **0.901 ± 0.002** |
| MLP HKRR | **0.007 ± 0.001** | 0.044 ± 0.006 | **0.007 ± 0.002** | **0.039 ± 0.003** | 0.879 ± 0.0 |
| MLP HJZ | 0.01 ± 0.002 | **0.043 ± 0.011** | 0.013 ± 0.002 | 0.039 ± 0.007 | 0.9 ± 0.003 |
| MLP Platt | 0.01 ± 0.002 | 0.048 ± 0.012 | 0.012 ± 0.001 | 0.044 ± 0.01 | 0.899 ± 0.001 |
| MLP Temp | 0.021 ± 0.006 | 0.047 ± 0.005 | 0.022 ± 0.005 | 0.041 ± 0.003 | 0.9 ± 0.002 |
| MLP Isotonic | 0.014 ± 0.003 | 0.044 ± 0.009 | 0.015 ± 0.002 | 0.04 ± 0.007 | 0.9 ± 0.0 |
| RandomForest ERM | 0.014 ± 0.001 | 0.045 ± 0.003 | 0.015 ± 0.0 | 0.038 ± 0.002 | **0.903 ± 0.002** |
| RandomForest HKRR | **0.007 ± 0.001** | 0.044 ± 0.006 | **0.007 ± 0.002** | 0.039 ± 0.003 | 0.879 ± 0.0 |
| RandomForest HJZ | 0.008 ± 0.001 | **0.035 ± 0.005** | 0.011 ± 0.001 | **0.031 ± 0.003** | 0.902 ± 0.001 |
| RandomForest Platt | 0.01 ± 0.002 | 0.039 ± 0.002 | 0.013 ± 0.001 | 0.033 ± 0.002 | 0.903 ± 0.001 |
| RandomForest Temp | 0.06 ± 0.002 | 0.084 ± 0.005 | 0.056 ± 0.001 | 0.07 ± 0.003 | 0.899 ± 0.001 |
| RandomForest Isotonic | 0.013 ± 0.005 | 0.037 ± 0.009 | 0.015 ± 0.004 | 0.034 ± 0.004 | 0.902 ± 0.001 |
| SVM ERM | 0.106 ± 0.001 | 0.211 ± 0.019 | 0.053 ± 0.001 | 0.106 ± 0.009 | **0.894 ± 0.001** |
| SVM HKRR | 0.007 ± 0.001 | **0.044 ± 0.006** | **0.007 ± 0.002** | **0.039 ± 0.003** | 0.879 ± 0.0 |
| SVM HJZ | **0.003 ± 0.001** | 0.073 ± 0.005 | 0.009 ± 0.001 | 0.073 ± 0.005 | 0.879 ± 0.0 |
| SVM Platt | 0.117 ± 0.001 | 0.246 ± 0.005 | 0.059 ± 0.001 | 0.123 ± 0.003 | 0.883 ± 0.001 |
| SVM Temp | 0.041 ± 0.004 | 0.091 ± 0.001 | 0.041 ± 0.004 | 0.091 ± 0.001 | 0.879 ± 0.001 |
| SVM Isotonic | 0.023 ± 0.009 | 0.129 ± 0.024 | 0.023 ± 0.009 | 0.129 ± 0.023 | 0.88 ± 0.001 |
| LogisticRegression ERM | 0.032 ± 0.001 | 0.062 ± 0.01 | 0.03 ± 0.001 | 0.053 ± 0.002 | 0.899 ± 0.001 |
| LogisticRegression HKRR | **0.007 ± 0.001** | 0.044 ± 0.006 | **0.007 ± 0.002** | 0.039 ± 0.003 | 0.879 ± 0.0 |
| LogisticRegression HJZ | 0.011 ± 0.001 | 0.045 ± 0.004 | 0.015 ± 0.001 | 0.042 ± 0.001 | **0.9 ± 0.001** |
| LogisticRegression Platt | 0.012 ± 0.001 | 0.049 ± 0.008 | 0.016 ± 0.001 | 0.043 ± 0.005 | 0.899 ± 0.002 |
| LogisticRegression Temp | 0.062 ± 0.001 | 0.088 ± 0.007 | 0.055 ± 0.001 | 0.066 ± 0.003 | 0.899 ± 0.001 |
| LogisticRegression Isotonic | 0.009 ± 0.002 | **0.04 ± 0.006** | 0.013 ± 0.001 | **0.036 ± 0.006** | 0.899 ± 0.002 |
| DecisionTree ERM | 0.028 ± 0.002 | 0.096 ± 0.014 | 0.022 ± 0.001 | 0.069 ± 0.003 | **0.897 ± 0.002** |
| DecisionTree HKRR | 0.007 ± 0.001 | **0.044 ± 0.006** | **0.007 ± 0.002** | **0.039 ± 0.003** | 0.879 ± 0.0 |
| DecisionTree HJZ | **0.003 ± 0.001** | 0.073 ± 0.005 | 0.009 ± 0.001 | 0.073 ± 0.005 | 0.879 ± 0.0 |
| DecisionTree Platt | 0.028 ± 0.004 | 0.086 ± 0.01 | 0.023 ± 0.003 | 0.067 ± 0.008 | 0.897 ± 0.003 |
| DecisionTree Temp | 0.056 ± 0.003 | 0.092 ± 0.01 | 0.05 ± 0.002 | 0.073 ± 0.002 | 0.896 ± 0.002 |
| DecisionTree Isotonic | 0.01 ± 0.002 | 0.06 ± 0.004 | 0.011 ± 0.002 | 0.055 ± 0.005 | 0.896 ± 0.002 |
| NaiveBayes ERM | 0.122 ± 0.003 | 0.271 ± 0.002 | 0.093 ± 0.002 | 0.197 ± 0.007 | 0.857 ± 0.003 |
| NaiveBayes HKRR | 0.007 ± 0.001 | **0.044 ± 0.006** | **0.007 ± 0.002** | **0.039 ± 0.003** | 0.879 ± 0.0 |
| NaiveBayes HJZ | **0.003 ± 0.001** | 0.073 ± 0.005 | 0.009 ± 0.001 | 0.073 ± 0.005 | 0.879 ± 0.0 |
| NaiveBayes Platt | 0.121 ± 0.003 | 0.263 ± 0.006 | 0.094 ± 0.002 | 0.195 ± 0.008 | 0.857 ± 0.002 |
| NaiveBayes Temp | 0.083 ± 0.003 | 0.264 ± 0.008 | 0.08 ± 0.003 | 0.24 ± 0.01 | 0.858 ± 0.002 |
| NaiveBayes Isotonic | 0.011 ± 0.003 | 0.055 ± 0.009 | 0.014 ± 0.003 | 0.047 ± 0.006 | **0.885 ± 0.002** |

Figure 26: Bank Marketing.

| Model | ECE ↓ | Max ECE ↓ | smECE ↓ | Max smECE ↓ | Acc ↑ |
|---|---|---|---|---|---|
| MLP ERM | 0.05 ± 0.007 | 0.104 ± 0.013 | 0.049 ± 0.006 | 0.092 ± 0.013 | 0.824 ± 0.007 |
| MLP HKRR | **0.005 ± 0.001** | 0.08 ± 0.006 | **0.005 ± 0.001** | 0.076 ± 0.005 | 0.754 ± 0.0 |
| MLP HJZ | 0.014 ± 0.003 | **0.076 ± 0.013** | 0.016 ± 0.002 | **0.065 ± 0.007** | 0.829 ± 0.009 |
| MLP Platt | 0.132 ± 0.01 | 0.176 ± 0.013 | 0.12 ± 0.007 | 0.149 ± 0.011 | 0.816 ± 0.008 |
| MLP Temp | 0.022 ± 0.007 | 0.083 ± 0.014 | 0.022 ± 0.006 | 0.078 ± 0.013 | 0.83 ± 0.007 |
| MLP Isotonic | 0.009 ± 0.001 | 0.076 ± 0.011 | 0.011 ± 0.001 | 0.071 ± 0.009 | **0.831 ± 0.007** |
| RandomForest ERM | 0.038 ± 0.002 | 0.099 ± 0.008 | 0.038 ± 0.002 | 0.088 ± 0.006 | 0.868 ± 0.001 |
| RandomForest HKRR | 0.013 ± 0.001 | 0.061 ± 0.019 | 0.013 ± 0.001 | **0.047 ± 0.006** | 0.862 ± 0.002 |
| RandomForest HJZ | 0.024 ± 0.003 | 0.076 ± 0.01 | 0.024 ± 0.002 | 0.062 ± 0.008 | 0.852 ± 0.022 |
| RandomForest Platt | 0.017 ± 0.002 | 0.078 ± 0.009 | 0.017 ± 0.002 | 0.069 ± 0.006 | 0.868 ± 0.001 |
| RandomForest Temp | 0.04 ± 0.002 | 0.061 ± 0.003 | 0.04 ± 0.001 | 0.055 ± 0.004 | 0.867 ± 0.001 |
| RandomForest Isotonic | **0.009 ± 0.002** | **0.058 ± 0.008** | **0.01 ± 0.002** | 0.048 ± 0.004 | **0.869 ± 0.001** |
| SVM ERM | 0.144 ± 0.001 | 0.175 ± 0.004 | 0.072 ± 0.0 | 0.088 ± 0.002 | **0.856 ± 0.001** |
| SVM HKRR | **0.005 ± 0.001** | **0.08 ± 0.006** | **0.005 ± 0.001** | **0.076 ± 0.005** | 0.754 ± 0.0 |
| SVM HJZ | 0.051 ± 0.006 | 0.133 ± 0.006 | 0.047 ± 0.009 | 0.133 ± 0.006 | 0.721 ± 0.02 |
| SVM Platt | 0.353 ± 0.003 | 0.417 ± 0.006 | 0.175 ± 0.001 | 0.205 ± 0.002 | 0.647 ± 0.003 |
| SVM Temp | 0.268 ± 0.001 | 0.288 ± 0.003 | 0.254 ± 0.001 | 0.269 ± 0.002 | 0.631 ± 0.005 |
| SVM Isotonic | 0.06 ± 0.049 | 0.187 ± 0.033 | 0.044 ± 0.036 | 0.152 ± 0.028 | 0.754 ± 0.0 |
| LogisticRegression ERM | 0.016 ± 0.001 | 0.103 ± 0.002 | 0.016 ± 0.001 | 0.101 ± 0.002 | 0.827 ± 0.001 |
| LogisticRegression HKRR | 0.012 ± 0.001 | **0.043 ± 0.01** | 0.012 ± 0.0 | **0.04 ± 0.011** | 0.83 ± 0.002 |
| LogisticRegression HJZ | 0.023 ± 0.006 | 0.084 ± 0.014 | 0.024 ± 0.006 | 0.076 ± 0.019 | **0.833 ± 0.01** |
| LogisticRegression Platt | 0.019 ± 0.007 | 0.079 ± 0.016 | 0.02 ± 0.006 | 0.077 ± 0.016 | 0.831 ± 0.011 |
| LogisticRegression Temp | 0.062 ± 0.012 | 0.1 ± 0.017 | 0.062 ± 0.01 | 0.095 ± 0.015 | 0.832 ± 0.012 |
| LogisticRegression Isotonic | **0.004 ± 0.001** | 0.088 ± 0.018 | **0.007 ± 0.001** | 0.087 ± 0.019 | 0.832 ± 0.012 |
| DecisionTree ERM | 0.019 ± 0.001 | 0.064 ± 0.007 | 0.018 ± 0.002 | 0.055 ± 0.006 | **0.863 ± 0.001** |
| DecisionTree HKRR | 0.013 ± 0.002 | **0.056 ± 0.014** | 0.013 ± 0.002 | **0.051 ± 0.011** | 0.858 ± 0.001 |
| DecisionTree HJZ | 0.018 ± 0.002 | 0.07 ± 0.013 | 0.017 ± 0.002 | 0.058 ± 0.007 | 0.862 ± 0.001 |
| DecisionTree Platt | 0.017 ± 0.002 | 0.073 ± 0.007 | 0.016 ± 0.001 | 0.057 ± 0.009 | 0.863 ± 0.002 |
| DecisionTree Temp | 0.064 ± 0.002 | 0.09 ± 0.007 | 0.055 ± 0.001 | 0.07 ± 0.004 | 0.859 ± 0.001 |
| DecisionTree Isotonic | **0.011 ± 0.004** | 0.066 ± 0.007 | **0.013 ± 0.003** | 0.057 ± 0.008 | 0.86 ± 0.003 |
| NaiveBayes ERM | 0.134 ± 0.001 | 0.199 ± 0.003 | 0.126 ± 0.0 | 0.165 ± 0.002 | 0.808 ± 0.001 |
| NaiveBayes HKRR | 0.009 ± 0.002 | **0.062 ± 0.008** | 0.009 ± 0.002 | **0.059 ± 0.009** | 0.817 ± 0.003 |
| NaiveBayes HJZ | 0.052 ± 0.012 | 0.117 ± 0.013 | 0.052 ± 0.01 | 0.108 ± 0.009 | 0.805 ± 0.003 |
| NaiveBayes Platt | 0.124 ± 0.003 | 0.2 ± 0.009 | 0.118 ± 0.003 | 0.168 ± 0.006 | 0.809 ± 0.002 |
| NaiveBayes Temp | 0.174 ± 0.002 | 0.185 ± 0.001 | 0.173 ± 0.002 | 0.177 ± 0.002 | 0.809 ± 0.0 |
| NaiveBayes Isotonic | **0.006 ± 0.001** | 0.116 ± 0.002 | **0.009 ± 0.001** | 0.116 ± 0.002 | **0.817 ± 0.001** |

Figure 27: HMDA.

| Model | ECE ↓ | Max ECE ↓ | smECE ↓ | Max smECE ↓ | Acc ↑ |
|---|---|---|---|---|---|
| MLP ERM | 0.018 ± 0.006 | 0.116 ± 0.035 | 0.02 ± 0.005 | 0.061 ± 0.005 | 0.819 ± 0.001 |
| MLP HKRR | 0.029 ± 0.002 | **0.057 ± 0.005** | 0.026 ± 0.001 | **0.049 ± 0.005** | 0.781 ± 0.0 |
| MLP HJZ | 0.029 ± 0.003 | 0.086 ± 0.003 | 0.028 ± 0.002 | 0.073 ± 0.001 | 0.781 ± 0.0 |
| MLP Platt | 0.028 ± 0.002 | 0.086 ± 0.001 | 0.027 ± 0.001 | 0.075 ± 0.002 | 0.781 ± 0.0 |
| MLP Temp | 0.035 ± 0.008 | 0.098 ± 0.015 | 0.035 ± 0.008 | 0.064 ± 0.004 | **0.821 ± 0.001** |
| MLP Isotonic | **0.003 ± 0.001** | 0.059 ± 0.001 | **0.003 ± 0.001** | 0.059 ± 0.001 | 0.781 ± 0.0 |
| RandomForest ERM | **0.019 ± 0.001** | 0.112 ± 0.017 | **0.02 ± 0.001** | 0.051 ± 0.003 | 0.82 ± 0.001 |
| RandomForest HKRR | 0.029 ± 0.002 | **0.057 ± 0.005** | 0.026 ± 0.001 | 0.049 ± 0.005 | 0.781 ± 0.0 |
| RandomForest HJZ | 0.029 ± 0.003 | 0.086 ± 0.003 | 0.028 ± 0.002 | 0.073 ± 0.001 | 0.781 ± 0.0 |
| RandomForest Platt | 0.027 ± 0.006 | 0.125 ± 0.028 | 0.029 ± 0.005 | 0.07 ± 0.008 | 0.812 ± 0.01 |
| RandomForest Temp | 0.027 ± 0.002 | 0.071 ± 0.006 | 0.026 ± 0.001 | **0.048 ± 0.001** | **0.82 ± 0.002** |
| RandomForest Isotonic | 0.027 ± 0.008 | 0.114 ± 0.015 | 0.025 ± 0.006 | 0.059 ± 0.004 | 0.818 ± 0.001 |
| SVM ERM | 0.181 ± 0.0 | 0.236 ± 0.002 | 0.091 ± 0.0 | 0.118 ± 0.001 | 0.819 ± 0.0 |
| SVM HKRR | 0.029 ± 0.002 | 0.057 ± 0.005 | 0.026 ± 0.001 | 0.049 ± 0.005 | 0.781 ± 0.0 |
| SVM HJZ | 0.029 ± 0.003 | 0.086 ± 0.003 | 0.028 ± 0.002 | 0.073 ± 0.001 | 0.781 ± 0.0 |
| SVM Platt | 0.18 ± 0.0 | 0.232 ± 0.003 | 0.09 ± 0.0 | 0.116 ± 0.001 | **0.82 ± 0.0** |
| SVM Temp | 0.06 ± 0.001 | 0.088 ± 0.0 | 0.06 ± 0.001 | 0.088 ± 0.0 | 0.819 ± 0.0 |
| SVM Isotonic | **0.013 ± 0.005** | **0.04 ± 0.005** | **0.013 ± 0.005** | **0.04 ± 0.004** | **0.82 ± 0.0** |
| LogisticRegression ERM | 0.01 ± 0.001 | 0.102 ± 0.026 | 0.015 ± 0.001 | 0.056 ± 0.002 | 0.819 ± 0.0 |
| LogisticRegression HKRR | 0.029 ± 0.002 | **0.057 ± 0.005** | 0.026 ± 0.001 | **0.049 ± 0.005** | 0.781 ± 0.0 |
| LogisticRegression HJZ | 0.029 ± 0.003 | 0.086 ± 0.003 | 0.028 ± 0.002 | 0.073 ± 0.001 | 0.781 ± 0.0 |
| LogisticRegression Platt | 0.023 ± 0.005 | 0.114 ± 0.018 | 0.023 ± 0.004 | 0.068 ± 0.005 | 0.817 ± 0.002 |
| LogisticRegression Temp | 0.022 ± 0.003 | 0.101 ± 0.015 | 0.022 ± 0.002 | 0.056 ± 0.003 | **0.819 ± 0.001** |
| LogisticRegression Isotonic | **0.009 ± 0.002** | 0.115 ± 0.026 | **0.014 ± 0.002** | 0.06 ± 0.004 | 0.818 ± 0.001 |
| DecisionTree ERM | 0.04 ± 0.003 | 0.181 ± 0.01 | 0.031 ± 0.001 | 0.089 ± 0.006 | 0.81 ± 0.003 |
| DecisionTree HKRR | 0.029 ± 0.002 | **0.057 ± 0.005** | 0.026 ± 0.001 | **0.049 ± 0.005** | 0.781 ± 0.0 |
| DecisionTree HJZ | 0.029 ± 0.003 | 0.086 ± 0.003 | 0.028 ± 0.002 | 0.073 ± 0.001 | 0.781 ± 0.0 |
| DecisionTree Platt | 0.038 ± 0.003 | 0.138 ± 0.027 | 0.029 ± 0.003 | 0.08 ± 0.007 | **0.811 ± 0.003** |
| DecisionTree Temp | 0.077 ± 0.001 | 0.154 ± 0.015 | 0.075 ± 0.001 | 0.103 ± 0.004 | 0.81 ± 0.003 |
| DecisionTree Isotonic | **0.021 ± 0.006** | 0.084 ± 0.018 | **0.022 ± 0.005** | 0.07 ± 0.007 | 0.811 ± 0.005 |
| NaiveBayes ERM | 0.187 ± 0.006 | 0.248 ± 0.009 | 0.108 ± 0.005 | 0.137 ± 0.004 | 0.807 ± 0.004 |
| NaiveBayes HKRR | 0.029 ± 0.002 | **0.057 ± 0.005** | **0.026 ± 0.001** | **0.049 ± 0.005** | 0.781 ± 0.0 |
| NaiveBayes HJZ | 0.029 ± 0.003 | 0.086 ± 0.003 | 0.028 ± 0.002 | 0.073 ± 0.001 | 0.781 ± 0.0 |
| NaiveBayes Platt | 0.197 ± 0.011 | 0.257 ± 0.012 | 0.119 ± 0.016 | 0.154 ± 0.019 | 0.792 ± 0.014 |
| NaiveBayes Temp | 0.07 ± 0.019 | 0.1 ± 0.023 | 0.069 ± 0.019 | 0.097 ± 0.025 | **0.807 ± 0.003** |
| NaiveBayes Isotonic | **0.028 ± 0.006** | 0.09 ± 0.022 | 0.027 ± 0.005 | 0.057 ± 0.008 | 0.806 ± 0.005 |

Figure 28: Credit Default.

| Model | ECE ↓ | Max ECE ↓ | smECE ↓ | Max smECE ↓ | Acc ↑ |
|---|---|---|---|---|---|
| MLP ERM | 0.022 ± 0.006 | 0.106 ± 0.009 | 0.024 ± 0.002 | 0.086 ± 0.015 | 0.864 ± 0.001 |
| MLP HKRR | 0.019 ± 0.005 | 0.122 ± 0.008 | **0.019 ± 0.004** | 0.104 ± 0.002 | 0.835 ± 0.003 |
| MLP HJZ | 0.019 ± 0.003 | **0.088 ± 0.011** | 0.021 ± 0.002 | **0.076 ± 0.018** | 0.864 ± 0.003 |
| MLP Platt | **0.017 ± 0.005** | 0.1 ± 0.019 | 0.019 ± 0.003 | 0.088 ± 0.02 | 0.865 ± 0.003 |
| MLP Temp | 0.019 ± 0.007 | 0.091 ± 0.016 | 0.02 ± 0.004 | 0.081 ± 0.02 | **0.866 ± 0.001** |
| MLP Isotonic | 0.02 ± 0.006 | 0.108 ± 0.021 | 0.02 ± 0.004 | 0.089 ± 0.021 | 0.864 ± 0.003 |
| RandomForest ERM | 0.019 ± 0.001 | 0.094 ± 0.006 | 0.021 ± 0.001 | **0.083 ± 0.004** | 0.863 ± 0.003 |
| RandomForest HKRR | 0.019 ± 0.005 | 0.122 ± 0.008 | 0.019 ± 0.004 | 0.104 ± 0.002 | 0.835 ± 0.003 |
| RandomForest HJZ | 0.021 ± 0.004 | 0.106 ± 0.011 | 0.021 ± 0.003 | 0.101 ± 0.012 | 0.86 ± 0.003 |
| RandomForest Platt | 0.017 ± 0.003 | 0.093 ± 0.003 | 0.02 ± 0.001 | 0.085 ± 0.005 | 0.861 ± 0.006 |
| RandomForest Temp | 0.045 ± 0.003 | 0.096 ± 0.007 | 0.045 ± 0.002 | 0.092 ± 0.009 | **0.863 ± 0.002** |
| RandomForest Isotonic | **0.015 ± 0.002** | **0.089 ± 0.014** | **0.017 ± 0.001** | 0.084 ± 0.014 | 0.862 ± 0.002 |
| SVM ERM | 0.143 ± 0.002 | 0.376 ± 0.012 | 0.072 ± 0.001 | 0.186 ± 0.006 | 0.857 ± 0.002 |
| SVM HKRR | **0.019 ± 0.005** | **0.122 ± 0.008** | **0.019 ± 0.004** | **0.104 ± 0.002** | 0.835 ± 0.003 |
| SVM HJZ | 0.031 ± 0.003 | 0.156 ± 0.021 | 0.027 ± 0.004 | 0.155 ± 0.02 | 0.828 ± 0.002 |
| SVM Platt | 0.14 ± 0.001 | 0.322 ± 0.019 | 0.07 ± 0.001 | 0.161 ± 0.009 | **0.86 ± 0.001** |
| SVM Temp | 0.073 ± 0.008 | 0.163 ± 0.019 | 0.073 ± 0.009 | 0.158 ± 0.015 | **0.86 ± 0.001** |
| SVM Isotonic | 0.048 ± 0.023 | 0.231 ± 0.085 | 0.048 ± 0.023 | 0.218 ± 0.069 | 0.847 ± 0.017 |
| LogisticRegression ERM | 0.022 ± 0.002 | **0.106 ± 0.008** | 0.022 ± 0.001 | **0.083 ± 0.003** | **0.866 ± 0.002** |
| LogisticRegression HKRR | 0.019 ± 0.005 | 0.122 ± 0.008 | **0.019 ± 0.004** | 0.104 ± 0.002 | 0.835 ± 0.003 |
| LogisticRegression HJZ | 0.021 ± 0.003 | 0.114 ± 0.019 | 0.023 ± 0.001 | 0.09 ± 0.011 | 0.866 ± 0.003 |
| LogisticRegression Platt | 0.018 ± 0.003 | 0.109 ± 0.009 | 0.021 ± 0.002 | 0.093 ± 0.017 | 0.864 ± 0.003 |
| LogisticRegression Temp | 0.047 ± 0.002 | 0.119 ± 0.007 | 0.044 ± 0.001 | 0.087 ± 0.003 | **0.866 ± 0.002** |
| LogisticRegression Isotonic | **0.017 ± 0.003** | 0.109 ± 0.019 | 0.019 ± 0.003 | 0.097 ± 0.02 | 0.863 ± 0.002 |
| DecisionTree ERM | 0.067 ± 0.004 | 0.261 ± 0.028 | 0.047 ± 0.004 | 0.166 ± 0.012 | **0.85 ± 0.006** |
| DecisionTree HKRR | 0.019 ± 0.005 | **0.122 ± 0.008** | 0.019 ± 0.004 | **0.104 ± 0.002** | 0.835 ± 0.003 |
| DecisionTree HJZ | 0.031 ± 0.003 | 0.156 ± 0.021 | 0.027 ± 0.004 | 0.155 ± 0.02 | 0.828 ± 0.002 |
| DecisionTree Platt | 0.08 ± 0.005 | 0.316 ± 0.029 | 0.054 ± 0.004 | 0.192 ± 0.009 | 0.838 ± 0.004 |
| DecisionTree Temp | 0.098 ± 0.007 | 0.214 ± 0.025 | 0.092 ± 0.005 | 0.172 ± 0.015 | 0.838 ± 0.004 |
| DecisionTree Isotonic | **0.014 ± 0.003** | 0.196 ± 0.026 | **0.015 ± 0.003** | 0.186 ± 0.027 | 0.838 ± 0.01 |
| NaiveBayes ERM | 0.277 ± 0.019 | 0.544 ± 0.02 | 0.164 ± 0.013 | 0.287 ± 0.011 | 0.714 ± 0.018 |
| NaiveBayes HKRR | 0.019 ± 0.005 | **0.122 ± 0.008** | **0.019 ± 0.004** | **0.104 ± 0.002** | **0.835 ± 0.003** |
| NaiveBayes HJZ | 0.031 ± 0.003 | 0.156 ± 0.021 | 0.027 ± 0.004 | 0.155 ± 0.02 | 0.828 ± 0.002 |
| NaiveBayes Platt | 0.269 ± 0.008 | 0.535 ± 0.009 | 0.165 ± 0.004 | 0.292 ± 0.003 | 0.719 ± 0.007 |
| NaiveBayes Temp | 0.294 ± 0.003 | 0.368 ± 0.008 | 0.274 ± 0.002 | 0.323 ± 0.005 | 0.719 ± 0.007 |
| NaiveBayes Isotonic | **0.019 ± 0.005** | 0.128 ± 0.017 | 0.021 ± 0.005 | 0.122 ± 0.015 | 0.831 ± 0.006 |

Figure 29: MEPS.

## H.3 Influence of Calibration Fraction on Multicalibration Error and Accuracy

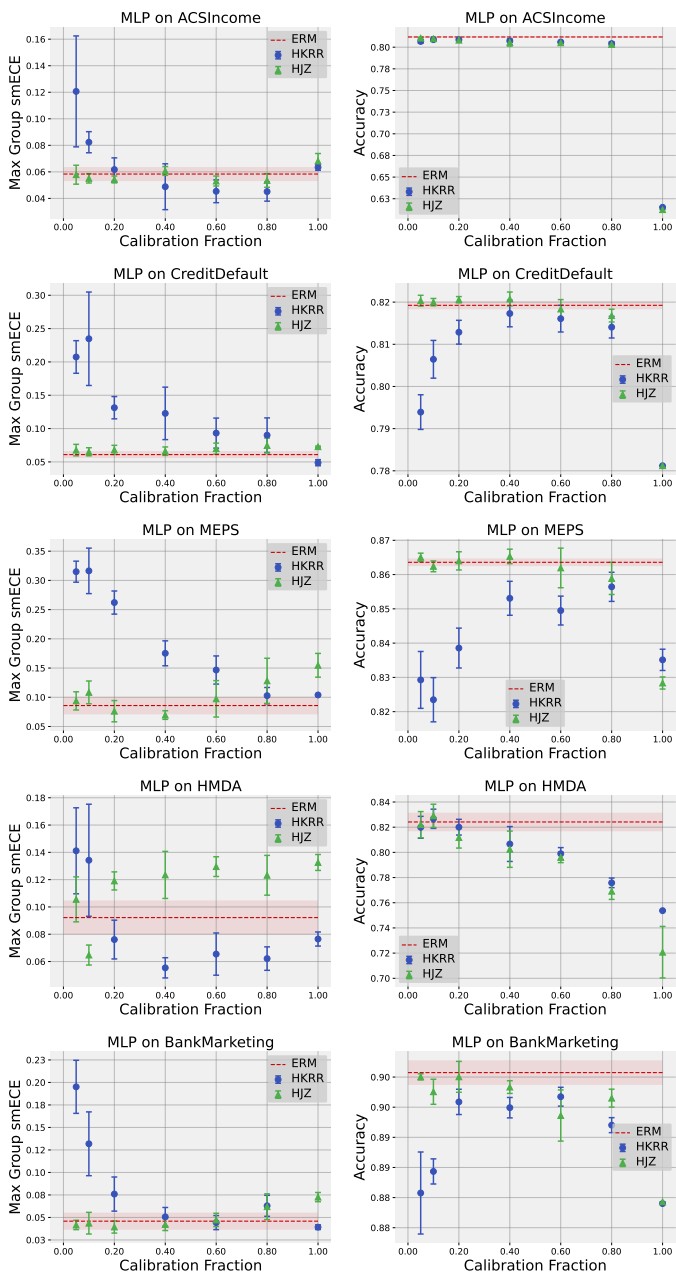

Figure 30: Influence of calibration fraction on MLP multicalibration and accuracy.

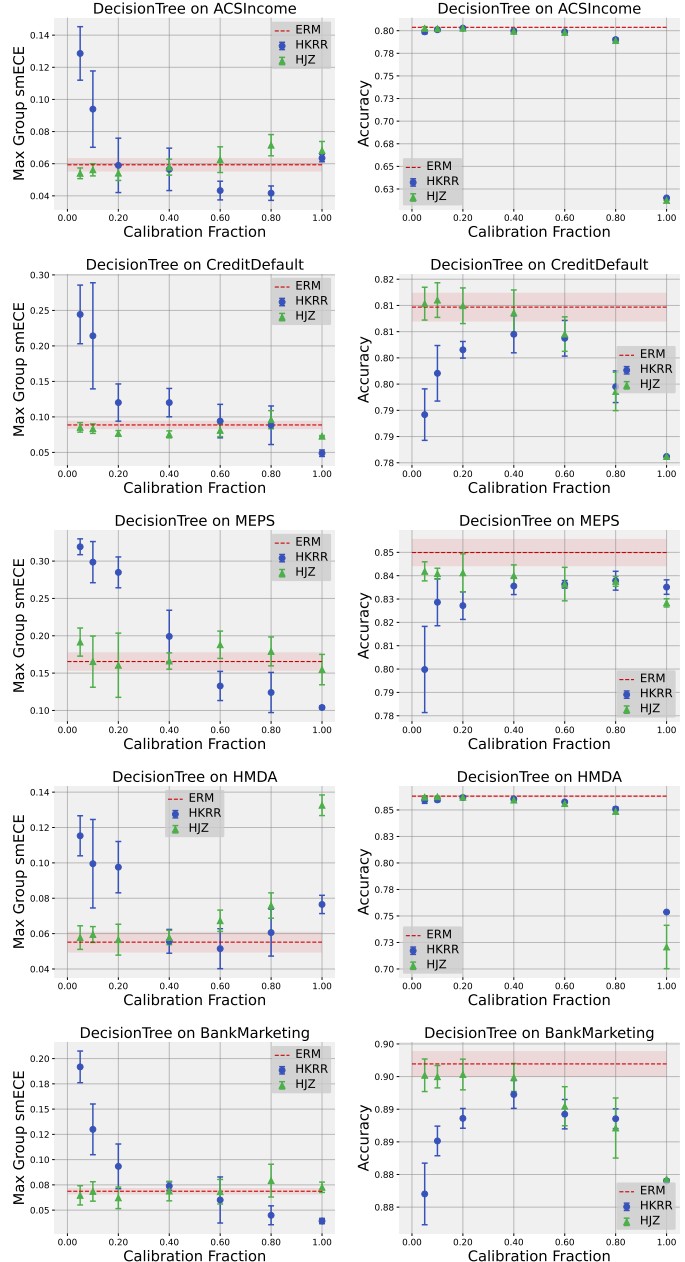

Figure 31: Influence of calibration fraction on decision tree multicalibration.

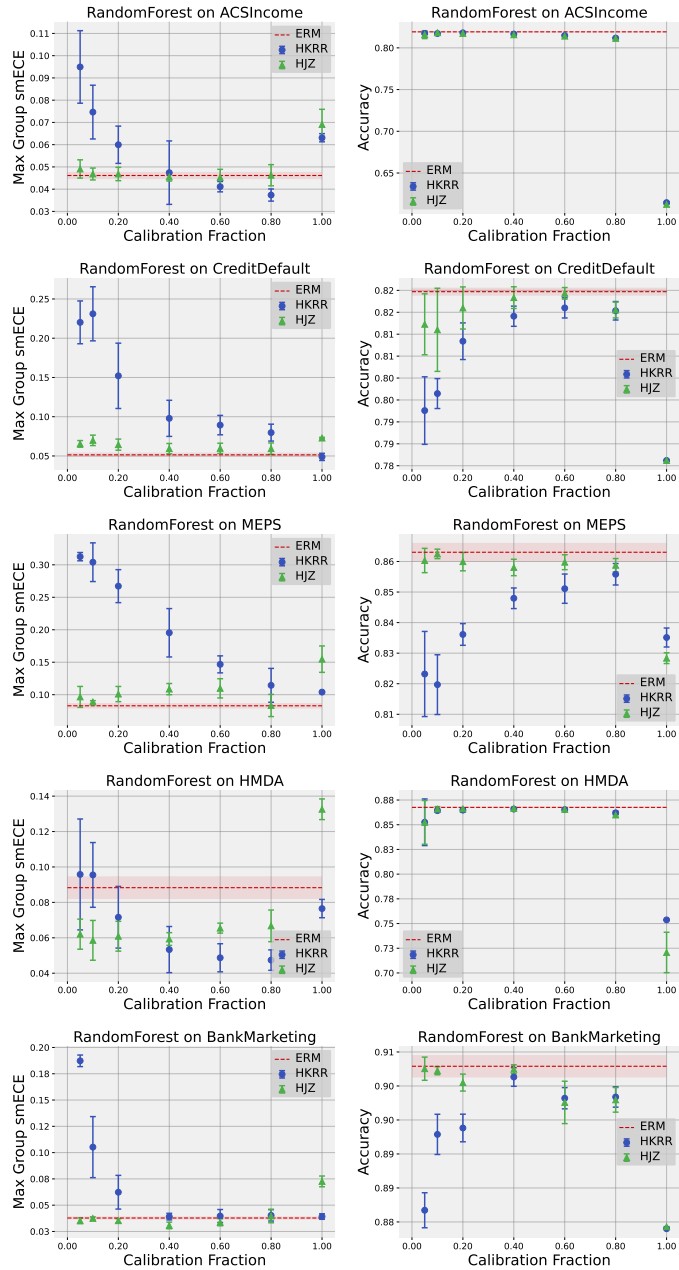

Figure 32: Influence of calibration fraction on RandomForest multicalibration.

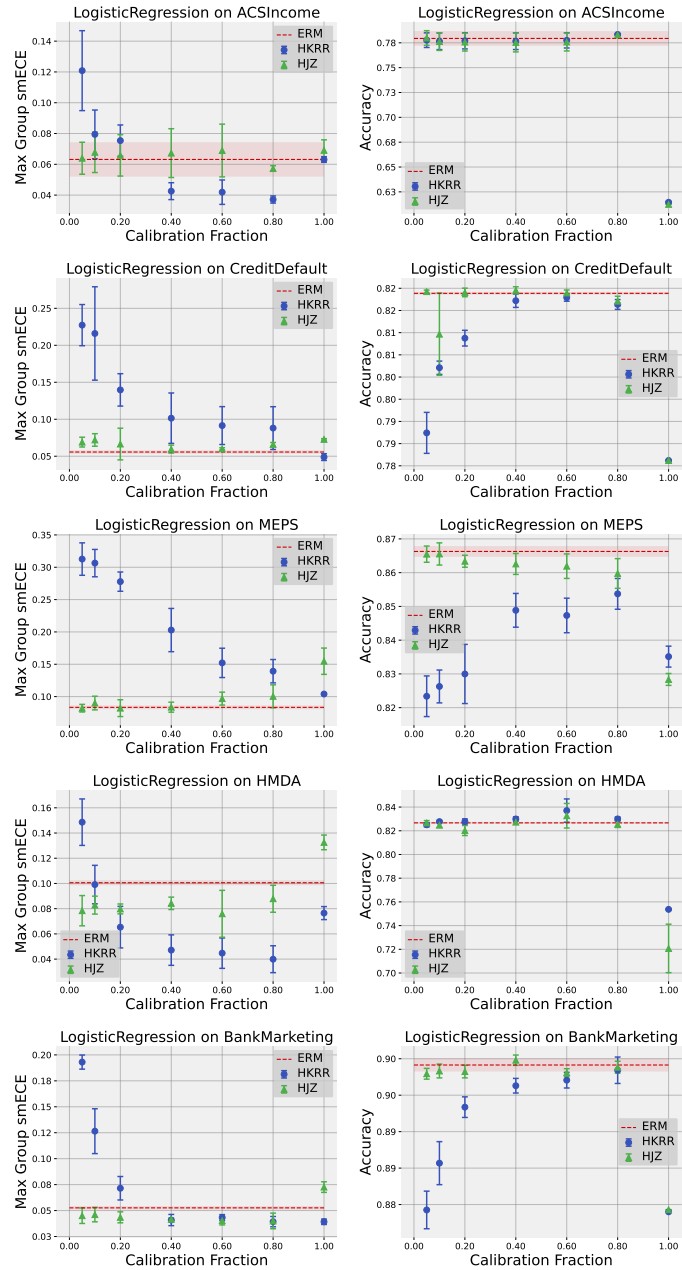

Figure 33: Influence of calibration fraction on LogisticRegression multicalibration.

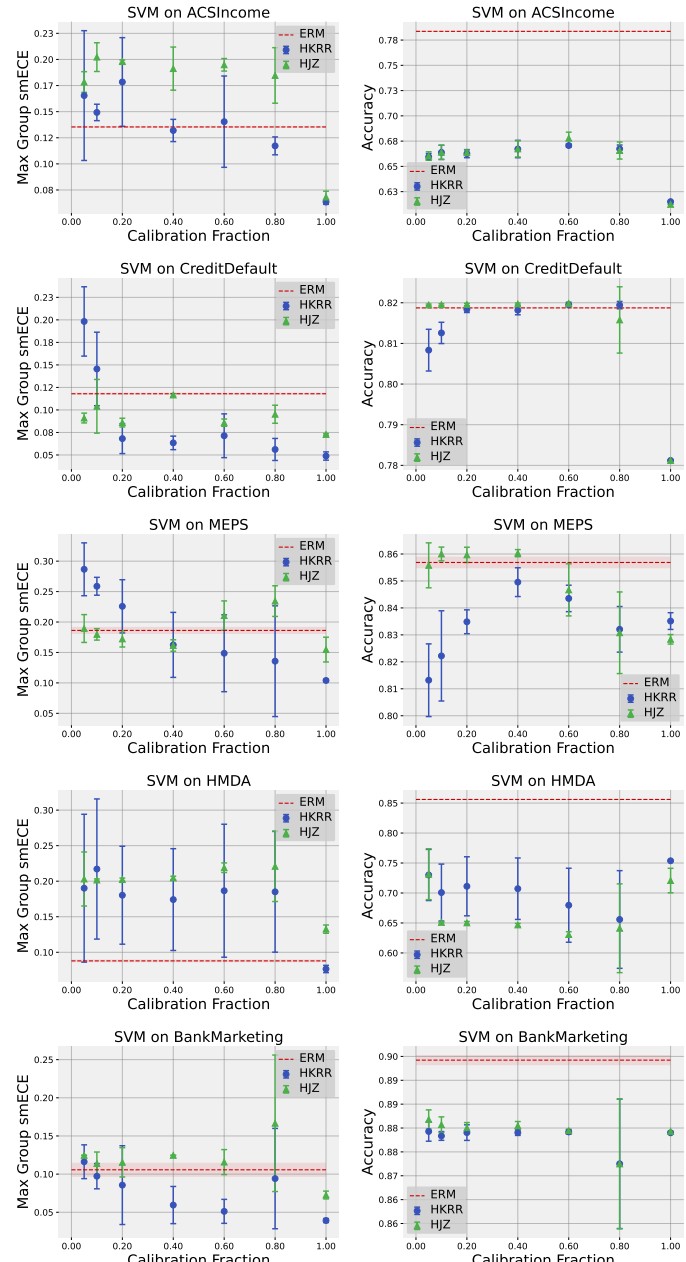

Figure 34: Influence of calibration fraction on SVM multicalibration.

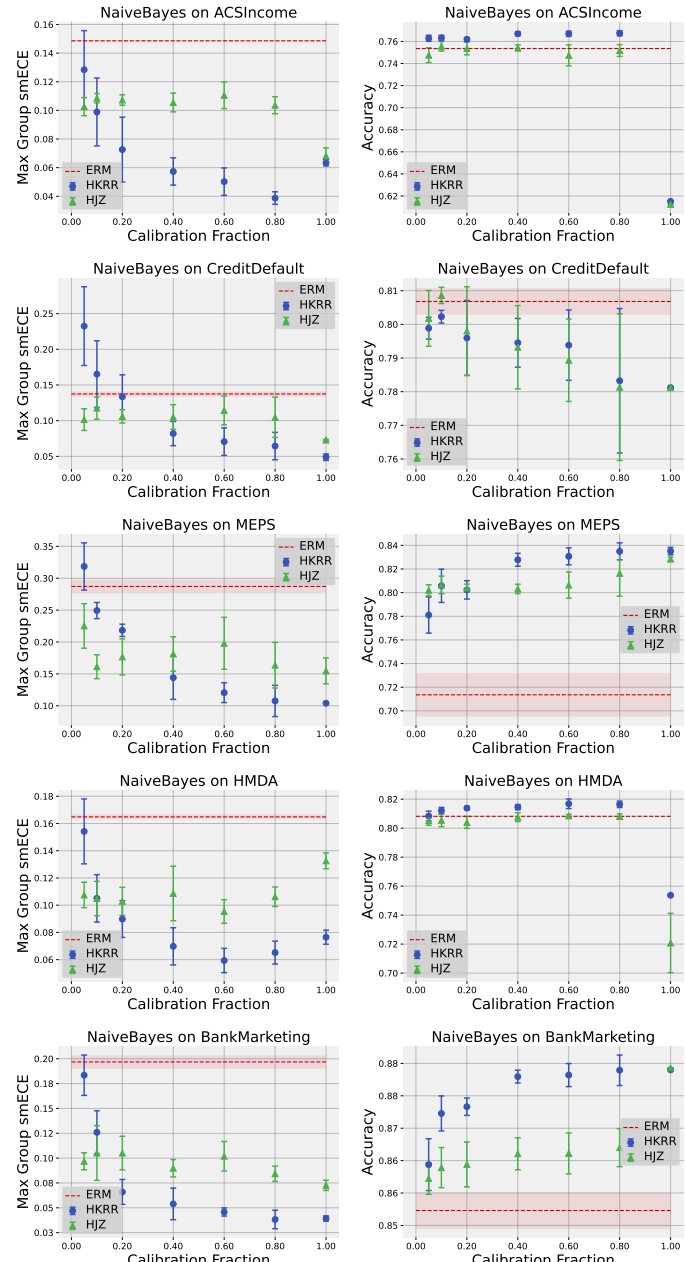

Figure 35: Influence of calibration fraction on NaiveBayes multicalibration.

## H.4   Tables Comparing Multicalibration Algorithms on Reused Data with ERM

| Model | ECE ↓ | Max ECE ↓ | smECE ↓ | Max smECE ↓ | Acc ↑ |
|---|---|---|---|---|---|
| MLP ERM | 0.017 ± 0.003 | 0.071 ± 0.009 | 0.017 ± 0.003 | 0.058 ± 0.009 | 0.81 ± 0.001 |
| MLP HKRR | 0.009 ± 0.002 | **0.048 ± 0.009** | **0.009 ± 0.002** | **0.04 ± 0.006** | **0.811 ± 0.001** |
| MLP HJZ | 0.013 ± 0.003 | 0.072 ± 0.009 | 0.015 ± 0.003 | 0.059 ± 0.007 | 0.808 ± 0.004 |
| MLP Platt | 0.01 ± 0.002 | 0.074 ± 0.008 | 0.012 ± 0.001 | 0.059 ± 0.004 | 0.811 ± 0.002 |
| MLP Temp | 0.017 ± 0.003 | 0.071 ± 0.009 | 0.017 ± 0.003 | 0.058 ± 0.009 | 0.81 ± 0.001 |
| MLP Isotonic | **0.008 ± 0.001** | 0.07 ± 0.002 | 0.01 ± 0.001 | 0.061 ± 0.002 | **0.811 ± 0.002** |
| RandomForest ERM | 0.009 ± 0.001 | 0.062 ± 0.009 | 0.011 ± 0.001 | 0.05 ± 0.003 | 0.819 ± 0.001 |
| RandomForest HKRR | 0.049 ± 0.001 | 0.123 ± 0.003 | 0.049 ± 0.001 | 0.12 ± 0.002 | 0.819 ± 0.001 |
| RandomForest HJZ | **0.007 ± 0.001** | 0.059 ± 0.01 | **0.009 ± 0.001** | 0.048 ± 0.003 | 0.819 ± 0.001 |
| RandomForest Platt | 0.017 ± 0.001 | **0.058 ± 0.003** | 0.017 ± 0.001 | **0.048 ± 0.002** | 0.819 ± 0.001 |
| RandomForest Temp | 0.027 ± 0.001 | 0.078 ± 0.007 | 0.027 ± 0.0 | 0.06 ± 0.001 | 0.819 ± 0.001 |
| RandomForest Isotonic | 0.075 ± 0.0 | 0.099 ± 0.004 | 0.071 ± 0.0 | 0.09 ± 0.003 | **0.819 ± 0.001** |
| SVM ERM | 0.299 ± 0.028 | 0.38 ± 0.054 | 0.15 ± 0.013 | 0.188 ± 0.024 | 0.701 ± 0.028 |
| SVM HKRR | 0.088 ± 0.012 | **0.061 ± 0.019** | 0.046 ± 0.006 | **0.052 ± 0.011** | **0.704 ± 0.025** |
| SVM HJZ | 0.135 ± 0.037 | 0.197 ± 0.04 | 0.084 ± 0.032 | 0.134 ± 0.039 | 0.703 ± 0.027 |
| SVM Platt | 0.299 ± 0.028 | 0.38 ± 0.054 | 0.15 ± 0.013 | 0.188 ± 0.024 | 0.701 ± 0.028 |
| SVM Temp | 0.064 ± 0.034 | 0.171 ± 0.029 | 0.062 ± 0.033 | 0.164 ± 0.031 | 0.701 ± 0.028 |
| SVM Isotonic | **0.002 ± 0.001** | 0.118 ± 0.014 | **0.002 ± 0.001** | 0.118 ± 0.014 | 0.701 ± 0.028 |
| LogisticRegression ERM | 0.012 ± 0.002 | 0.065 ± 0.011 | 0.015 ± 0.002 | 0.063 ± 0.011 | 0.779 ± 0.007 |
| LogisticRegression HKRR | 0.011 ± 0.005 | **0.045 ± 0.011** | 0.011 ± 0.005 | **0.038 ± 0.007** | **0.781 ± 0.006** |
| LogisticRegression HJZ | 0.011 ± 0.004 | 0.066 ± 0.014 | 0.013 ± 0.003 | 0.062 ± 0.015 | 0.78 ± 0.007 |
| LogisticRegression Platt | 0.012 ± 0.004 | 0.069 ± 0.018 | 0.013 ± 0.003 | 0.064 ± 0.016 | 0.78 ± 0.006 |
| LogisticRegression Temp | 0.02 ± 0.001 | 0.08 ± 0.01 | 0.02 ± 0.0 | 0.076 ± 0.009 | 0.779 ± 0.007 |
| LogisticRegression Isotonic | **0.005 ± 0.001** | 0.064 ± 0.007 | **0.008 ± 0.001** | 0.062 ± 0.008 | 0.779 ± 0.007 |
| DecisionTree ERM | 0.017 ± 0.001 | 0.066 ± 0.01 | **0.016 ± 0.001** | 0.059 ± 0.004 | 0.804 ± 0.0 |
| DecisionTree HKRR | **0.016 ± 0.001** | **0.05 ± 0.004** | 0.016 ± 0.001 | **0.047 ± 0.004** | **0.804 ± 0.0** |
| DecisionTree HJZ | 0.017 ± 0.001 | 0.06 ± 0.007 | 0.016 ± 0.001 | 0.055 ± 0.002 | 0.803 ± 0.001 |
| DecisionTree Platt | 0.017 ± 0.001 | 0.06 ± 0.007 | 0.016 ± 0.001 | 0.055 ± 0.002 | 0.803 ± 0.001 |
| DecisionTree Temp | 0.027 ± 0.001 | 0.078 ± 0.008 | 0.027 ± 0.001 | 0.07 ± 0.006 | 0.804 ± 0.0 |
| DecisionTree Isotonic | 0.017 ± 0.001 | 0.066 ± 0.01 | **0.016 ± 0.001** | 0.059 ± 0.004 | 0.804 ± 0.0 |
| NaiveBayes ERM | 0.117 ± 0.0 | 0.165 ± 0.0 | 0.109 ± 0.0 | 0.149 ± 0.001 | 0.754 ± 0.0 |
| NaiveBayes HKRR | 0.039 ± 0.002 | **0.059 ± 0.014** | 0.038 ± 0.002 | **0.047 ± 0.005** | 0.764 ± 0.002 |
| NaiveBayes HJZ | 0.07 ± 0.013 | 0.111 ± 0.011 | 0.064 ± 0.01 | 0.103 ± 0.009 | 0.751 ± 0.004 |
| NaiveBayes Platt | 0.09 ± 0.001 | 0.127 ± 0.001 | 0.085 ± 0.0 | 0.12 ± 0.001 | 0.76 ± 0.0 |
| NaiveBayes Temp | 0.089 ± 0.001 | 0.155 ± 0.003 | 0.087 ± 0.0 | 0.154 ± 0.001 | 0.754 ± 0.0 |
| NaiveBayes Isotonic | **0.003 ± 0.001** | 0.094 ± 0.002 | **0.007 ± 0.0** | 0.086 ± 0.0 | **0.768 ± 0.0** |

Figure 36: ACS Income. Training data reused for post-processing.

| Model | ECE ↓ | Max ECE ↓ | smECE ↓ | Max smECE ↓ | Acc ↑ |
|---|---|---|---|---|---|
| MLP ERM | 0.013 ± 0.005 | 0.046 ± 0.005 | 0.014 ± 0.004 | 0.042 ± 0.005 | 0.901 ± 0.001 |
| MLP HKRR | 0.009 ± 0.002 | 0.046 ± 0.002 | **0.009 ± 0.002** | 0.041 ± 0.002 | 0.9 ± 0.001 |
| MLP HJZ | **0.007 ± 0.002** | 0.047 ± 0.006 | 0.011 ± 0.0 | **0.039 ± 0.005** | 0.9 ± 0.002 |
| MLP Platt | 0.008 ± 0.002 | **0.046 ± 0.006** | 0.011 ± 0.001 | 0.04 ± 0.005 | **0.901 ± 0.001** |
| MLP Temp | 0.013 ± 0.005 | 0.046 ± 0.005 | 0.014 ± 0.004 | 0.042 ± 0.005 | 0.901 ± 0.001 |
| MLP Isotonic | 0.008 ± 0.002 | 0.047 ± 0.007 | 0.01 ± 0.001 | 0.042 ± 0.006 | 0.9 ± 0.001 |
| RandomForest ERM | 0.014 ± 0.002 | 0.04 ± 0.003 | 0.015 ± 0.001 | 0.037 ± 0.002 | 0.902 ± 0.001 |
| RandomForest HKRR | 0.047 ± 0.003 | 0.135 ± 0.006 | 0.046 ± 0.002 | 0.132 ± 0.006 | 0.902 ± 0.001 |
| RandomForest HJZ | **0.01 ± 0.001** | **0.038 ± 0.004** | **0.012 ± 0.001** | **0.035 ± 0.002** | **0.903 ± 0.001** |
| RandomForest Platt | 0.015 ± 0.001 | 0.054 ± 0.006 | 0.017 ± 0.001 | 0.045 ± 0.002 | 0.903 ± 0.001 |
| RandomForest Temp | 0.058 ± 0.001 | 0.086 ± 0.003 | 0.056 ± 0.001 | 0.076 ± 0.001 | 0.902 ± 0.001 |
| RandomForest Isotonic | 0.056 ± 0.001 | 0.117 ± 0.005 | 0.045 ± 0.001 | 0.093 ± 0.006 | 0.902 ± 0.001 |
| SVM ERM | 0.205 ± 0.11 | 0.309 ± 0.087 | 0.102 ± 0.055 | 0.154 ± 0.041 | 0.795 ± 0.11 |
| SVM HKRR | 0.007 ± 0.001 | **0.042 ± 0.005** | 0.007 ± 0.001 | **0.037 ± 0.002** | **0.88 ± 0.001** |
| SVM HJZ | 0.021 ± 0.005 | 0.121 ± 0.012 | 0.024 ± 0.002 | 0.119 ± 0.014 | 0.878 ± 0.002 |
| SVM Platt | 0.205 ± 0.11 | 0.309 ± 0.087 | 0.102 ± 0.055 | 0.154 ± 0.041 | 0.795 ± 0.11 |
| SVM Temp | 0.165 ± 0.113 | 0.218 ± 0.105 | 0.155 ± 0.094 | 0.205 ± 0.081 | 0.795 ± 0.11 |
| SVM Isotonic | **0.004 ± 0.001** | 0.144 ± 0.006 | **0.004 ± 0.001** | 0.143 ± 0.006 | 0.879 ± 0.0 |
| LogisticRegression ERM | 0.032 ± 0.001 | 0.062 ± 0.01 | 0.03 ± 0.001 | 0.053 ± 0.002 | 0.899 ± 0.001 |
| LogisticRegression HKRR | 0.014 ± 0.003 | **0.039 ± 0.007** | 0.014 ± 0.003 | 0.036 ± 0.006 | 0.897 ± 0.002 |
| LogisticRegression HJZ | 0.01 ± 0.001 | 0.048 ± 0.005 | 0.014 ± 0.001 | 0.04 ± 0.003 | 0.899 ± 0.001 |
| LogisticRegression Platt | 0.012 ± 0.001 | 0.048 ± 0.006 | 0.015 ± 0.001 | 0.04 ± 0.003 | 0.899 ± 0.001 |
| LogisticRegression Temp | 0.062 ± 0.001 | 0.084 ± 0.004 | 0.056 ± 0.0 | 0.064 ± 0.001 | 0.899 ± 0.001 |
| LogisticRegression Isotonic | **0.006 ± 0.002** | 0.04 ± 0.005 | **0.01 ± 0.002** | **0.034 ± 0.003** | **0.9 ± 0.001** |
| DecisionTree ERM | 0.029 ± 0.002 | 0.099 ± 0.017 | 0.022 ± 0.001 | **0.069 ± 0.006** | 0.897 ± 0.002 |
| DecisionTree HKRR | 0.029 ± 0.003 | 0.101 ± 0.018 | 0.029 ± 0.003 | 0.09 ± 0.013 | **0.897 ± 0.002** |
| DecisionTree HJZ | 0.029 ± 0.003 | 0.1 ± 0.018 | 0.022 ± 0.002 | 0.07 ± 0.01 | 0.897 ± 0.002 |
| DecisionTree Platt | **0.028 ± 0.002** | 0.103 ± 0.017 | **0.022 ± 0.001** | 0.071 ± 0.007 | 0.897 ± 0.002 |
| DecisionTree Temp | 0.053 ± 0.004 | **0.085 ± 0.008** | 0.048 ± 0.001 | 0.072 ± 0.003 | 0.897 ± 0.002 |
| DecisionTree Isotonic | 0.029 ± 0.002 | 0.099 ± 0.017 | 0.022 ± 0.001 | **0.069 ± 0.006** | 0.897 ± 0.002 |
| NaiveBayes ERM | 0.122 ± 0.003 | 0.271 ± 0.002 | 0.093 ± 0.002 | 0.197 ± 0.007 | 0.857 ± 0.003 |
| NaiveBayes HKRR | 0.011 ± 0.005 | **0.044 ± 0.01** | 0.011 ± 0.005 | **0.038 ± 0.007** | 0.88 ± 0.002 |
| NaiveBayes HJZ | 0.044 ± 0.009 | 0.143 ± 0.023 | 0.042 ± 0.009 | 0.105 ± 0.011 | 0.864 ± 0.002 |
| NaiveBayes Platt | 0.12 ± 0.003 | 0.263 ± 0.002 | 0.094 ± 0.002 | 0.194 ± 0.007 | 0.857 ± 0.003 |
| NaiveBayes Temp | 0.083 ± 0.002 | 0.267 ± 0.004 | 0.081 ± 0.002 | 0.241 ± 0.005 | 0.857 ± 0.003 |
| NaiveBayes Isotonic | **0.006 ± 0.001** | 0.047 ± 0.003 | **0.009 ± 0.001** | 0.043 ± 0.001 | **0.886 ± 0.001** |

Figure 37: Bank Marketing. Training data reused for post-processing.

| Model | ECE ↓ | Max ECE ↓ | smECE ↓ | Max smECE ↓ | Acc ↑ |
|---|---|---|---|---|---|
| MLP ERM | 0.034 ± 0.003 | 0.093 ± 0.014 | 0.034 ± 0.003 | 0.08 ± 0.005 | 0.83 ± 0.005 |
| MLP HKRR | 0.011 ± 0.002 | **0.056 ± 0.015** | 0.011 ± 0.002 | **0.05 ± 0.011** | 0.835 ± 0.004 |
| MLP HJZ | 0.009 ± 0.001 | 0.068 ± 0.01 | 0.012 ± 0.0 | 0.058 ± 0.007 | 0.833 ± 0.004 |
| MLP Platt | 0.011 ± 0.002 | 0.076 ± 0.007 | 0.013 ± 0.002 | 0.063 ± 0.003 | **0.835 ± 0.005** |
| MLP Temp | 0.024 ± 0.008 | 0.086 ± 0.01 | 0.024 ± 0.008 | 0.075 ± 0.011 | 0.83 ± 0.005 |
| MLP Isotonic | **0.005 ± 0.001** | 0.07 ± 0.017 | **0.008 ± 0.002** | 0.057 ± 0.005 | 0.833 ± 0.006 |
| RandomForest ERM | 0.038 ± 0.001 | 0.097 ± 0.007 | 0.038 ± 0.001 | 0.089 ± 0.005 | 0.868 ± 0.001 |
| RandomForest HKRR | 0.073 ± 0.002 | 0.11 ± 0.002 | 0.073 ± 0.002 | 0.106 ± 0.002 | 0.865 ± 0.001 |
| RandomForest HJZ | **0.023 ± 0.001** | 0.058 ± 0.004 | **0.023 ± 0.001** | **0.05 ± 0.003** | 0.868 ± 0.001 |
| RandomForest Platt | 0.024 ± 0.001 | 0.064 ± 0.005 | 0.024 ± 0.001 | 0.059 ± 0.004 | 0.868 ± 0.001 |
| RandomForest Temp | 0.038 ± 0.001 | **0.057 ± 0.003** | 0.039 ± 0.001 | 0.051 ± 0.002 | 0.868 ± 0.001 |
| RandomForest Isotonic | 0.074 ± 0.002 | 0.09 ± 0.005 | 0.068 ± 0.002 | 0.079 ± 0.003 | **0.868 ± 0.001** |
| SVM ERM | 0.345 ± 0.2 | 0.49 ± 0.158 | 0.166 ± 0.08 | 0.233 ± 0.055 | 0.655 ± 0.2 |
| SVM HKRR | 0.008 ± 0.002 | **0.071 ± 0.012** | 0.008 ± 0.002 | **0.058 ± 0.012** | 0.754 ± 0.0 |
| SVM HJZ | 0.051 ± 0.017 | 0.177 ± 0.033 | 0.048 ± 0.016 | 0.169 ± 0.03 | 0.723 ± 0.029 |
| SVM Platt | 0.345 ± 0.2 | 0.49 ± 0.158 | 0.166 ± 0.08 | 0.233 ± 0.055 | 0.655 ± 0.2 |
| SVM Temp | 0.135 ± 0.155 | 0.197 ± 0.155 | 0.119 ± 0.124 | 0.176 ± 0.112 | 0.655 ± 0.2 |
| SVM Isotonic | **0.002 ± 0.001** | 0.163 ± 0.002 | **0.002 ± 0.001** | 0.164 ± 0.002 | **0.754 ± 0.0** |
| LogisticRegression ERM | 0.016 ± 0.001 | 0.103 ± 0.002 | 0.016 ± 0.001 | 0.101 ± 0.002 | 0.827 ± 0.001 |
| LogisticRegression HKRR | 0.008 ± 0.001 | **0.036 ± 0.004** | 0.008 ± 0.001 | **0.035 ± 0.003** | **0.832 ± 0.001** |
| LogisticRegression HJZ | 0.017 ± 0.002 | 0.085 ± 0.006 | 0.018 ± 0.002 | 0.083 ± 0.005 | 0.828 ± 0.001 |
| LogisticRegression Platt | 0.008 ± 0.001 | 0.082 ± 0.004 | 0.01 ± 0.0 | 0.082 ± 0.004 | 0.829 ± 0.001 |
| LogisticRegression Temp | 0.069 ± 0.001 | 0.109 ± 0.002 | 0.067 ± 0.001 | 0.102 ± 0.003 | 0.827 ± 0.001 |
| LogisticRegression Isotonic | **0.003 ± 0.0** | 0.097 ± 0.002 | **0.006 ± 0.001** | 0.095 ± 0.001 | 0.826 ± 0.001 |
| DecisionTree ERM | 0.019 ± 0.002 | 0.064 ± 0.008 | 0.018 ± 0.002 | 0.054 ± 0.004 | 0.863 ± 0.001 |
| DecisionTree HKRR | 0.019 ± 0.002 | **0.052 ± 0.007** | 0.019 ± 0.002 | **0.048 ± 0.007** | **0.864 ± 0.001** |
| DecisionTree HJZ | 0.02 ± 0.001 | 0.058 ± 0.005 | 0.019 ± 0.001 | 0.052 ± 0.004 | 0.863 ± 0.001 |
| DecisionTree Platt | **0.019 ± 0.002** | 0.066 ± 0.006 | **0.017 ± 0.001** | 0.055 ± 0.006 | 0.863 ± 0.001 |
| DecisionTree Temp | 0.062 ± 0.002 | 0.089 ± 0.005 | 0.054 ± 0.002 | 0.073 ± 0.003 | 0.863 ± 0.001 |
| DecisionTree Isotonic | 0.019 ± 0.002 | 0.064 ± 0.008 | 0.018 ± 0.002 | 0.054 ± 0.004 | 0.863 ± 0.001 |
| NaiveBayes ERM | 0.134 ± 0.001 | 0.199 ± 0.003 | 0.126 ± 0.0 | 0.165 ± 0.002 | 0.808 ± 0.001 |
| NaiveBayes HKRR | 0.009 ± 0.001 | **0.069 ± 0.008** | 0.009 ± 0.001 | **0.059 ± 0.005** | 0.814 ± 0.002 |
| NaiveBayes HJZ | 0.054 ± 0.009 | 0.12 ± 0.018 | 0.052 ± 0.008 | 0.115 ± 0.019 | 0.807 ± 0.001 |
| NaiveBayes Platt | 0.122 ± 0.0 | 0.193 ± 0.003 | 0.116 ± 0.0 | 0.165 ± 0.002 | 0.808 ± 0.001 |
| NaiveBayes Temp | 0.175 ± 0.001 | 0.185 ± 0.001 | 0.174 ± 0.001 | 0.178 ± 0.001 | 0.808 ± 0.001 |
| NaiveBayes Isotonic | **0.006 ± 0.0** | 0.117 ± 0.001 | **0.008 ± 0.0** | 0.117 ± 0.001 | **0.817 ± 0.001** |

Figure 38: HMDA. Training data reused for post-processing.

| Model | ECE ↓ | Max ECE ↓ | smECE ↓ | Max smECE ↓ | Acc ↑ |
|---|---|---|---|---|---|
| MLP ERM | 0.018 ± 0.005 | 0.115 ± 0.043 | 0.02 ± 0.004 | 0.064 ± 0.007 | 0.819 ± 0.001 |
| MLP HKRR | **0.016 ± 0.003** | 0.121 ± 0.052 | **0.016 ± 0.003** | 0.083 ± 0.02 | **0.819 ± 0.001** |
| MLP HJZ | 0.017 ± 0.003 | **0.097 ± 0.011** | 0.019 ± 0.002 | 0.06 ± 0.006 | 0.819 ± 0.002 |
| MLP Platt | 0.016 ± 0.002 | 0.116 ± 0.036 | 0.018 ± 0.001 | **0.058 ± 0.007** | 0.819 ± 0.002 |
| MLP Temp | 0.018 ± 0.005 | 0.115 ± 0.043 | 0.02 ± 0.004 | 0.064 ± 0.007 | 0.819 ± 0.001 |
| MLP Isotonic | 0.019 ± 0.002 | 0.156 ± 0.026 | 0.018 ± 0.001 | 0.071 ± 0.005 | 0.819 ± 0.001 |
| RandomForest ERM | 0.019 ± 0.002 | 0.112 ± 0.021 | 0.02 ± 0.001 | 0.052 ± 0.004 | 0.82 ± 0.001 |
| RandomForest HKRR | 0.022 ± 0.003 | 0.144 ± 0.023 | 0.022 ± 0.002 | 0.089 ± 0.008 | 0.818 ± 0.001 |
| RandomForest HJZ | **0.018 ± 0.003** | 0.123 ± 0.027 | **0.019 ± 0.002** | 0.058 ± 0.003 | 0.818 ± 0.002 |
| RandomForest Platt | 0.027 ± 0.003 | 0.141 ± 0.014 | 0.026 ± 0.002 | 0.063 ± 0.002 | 0.82 ± 0.001 |
| RandomForest Temp | 0.026 ± 0.003 | **0.073 ± 0.01** | 0.026 ± 0.001 | **0.048 ± 0.002** | 0.82 ± 0.001 |
| RandomForest Isotonic | 0.037 ± 0.003 | 0.124 ± 0.02 | 0.035 ± 0.002 | 0.073 ± 0.003 | **0.82 ± 0.001** |
| SVM ERM | 0.18 ± 0.0 | 0.233 ± 0.0 | 0.09 ± 0.0 | 0.117 ± 0.0 | **0.82 ± 0.0** |
| SVM HKRR | 0.03 ± 0.001 | 0.06 ± 0.002 | 0.021 ± 0.002 | 0.052 ± 0.004 | **0.82 ± 0.0** |
| SVM HJZ | 0.051 ± 0.006 | 0.166 ± 0.015 | 0.038 ± 0.004 | 0.091 ± 0.006 | 0.813 ± 0.008 |
| SVM Platt | 0.18 ± 0.0 | 0.233 ± 0.0 | 0.09 ± 0.0 | 0.117 ± 0.0 | **0.82 ± 0.0** |
| SVM Temp | 0.057 ± 0.0 | 0.088 ± 0.0 | 0.057 ± 0.0 | 0.088 ± 0.0 | **0.82 ± 0.0** |
| SVM Isotonic | **0.002 ± 0.001** | **0.044 ± 0.001** | **0.002 ± 0.001** | **0.043 ± 0.001** | **0.82 ± 0.0** |
| LogisticRegression ERM | 0.01 ± 0.001 | 0.102 ± 0.026 | 0.015 ± 0.001 | 0.056 ± 0.002 | 0.819 ± 0.0 |
| LogisticRegression HKRR | **0.008 ± 0.001** | 0.091 ± 0.036 | **0.008 ± 0.001** | 0.073 ± 0.028 | 0.819 ± 0.001 |
| LogisticRegression HJZ | 0.013 ± 0.004 | 0.102 ± 0.032 | 0.017 ± 0.002 | 0.059 ± 0.005 | 0.817 ± 0.002 |
| LogisticRegression Platt | 0.013 ± 0.001 | 0.097 ± 0.035 | 0.016 ± 0.001 | 0.057 ± 0.004 | **0.819 ± 0.0** |
| LogisticRegression Temp | 0.023 ± 0.001 | **0.075 ± 0.011** | 0.024 ± 0.0 | **0.053 ± 0.002** | 0.819 ± 0.0 |
| LogisticRegression Isotonic | 0.013 ± 0.003 | 0.13 ± 0.021 | 0.017 ± 0.002 | 0.061 ± 0.005 | 0.819 ± 0.001 |
| DecisionTree ERM | **0.041 ± 0.004** | 0.186 ± 0.014 | 0.031 ± 0.002 | 0.088 ± 0.007 | **0.81 ± 0.003** |
| DecisionTree HKRR | 0.041 ± 0.004 | 0.183 ± 0.017 | 0.039 ± 0.003 | 0.107 ± 0.014 | 0.81 ± 0.002 |
| DecisionTree HJZ | 0.042 ± 0.003 | 0.168 ± 0.021 | **0.031 ± 0.001** | **0.085 ± 0.004** | **0.81 ± 0.003** |
| DecisionTree Platt | 0.043 ± 0.003 | 0.188 ± 0.02 | 0.033 ± 0.001 | 0.089 ± 0.006 | **0.81 ± 0.003** |
| DecisionTree Temp | 0.08 ± 0.001 | **0.167 ± 0.022** | 0.076 ± 0.001 | 0.111 ± 0.014 | **0.81 ± 0.003** |
| DecisionTree Isotonic | **0.041 ± 0.004** | 0.186 ± 0.014 | 0.031 ± 0.002 | 0.088 ± 0.007 | **0.81 ± 0.003** |
| NaiveBayes ERM | 0.187 ± 0.006 | 0.248 ± 0.009 | 0.108 ± 0.005 | 0.137 ± 0.004 | 0.807 ± 0.004 |
| NaiveBayes HKRR | 0.028 ± 0.004 | 0.091 ± 0.016 | 0.028 ± 0.004 | 0.089 ± 0.019 | 0.807 ± 0.004 |
| NaiveBayes HJZ | 0.069 ± 0.018 | 0.139 ± 0.027 | 0.057 ± 0.016 | 0.103 ± 0.018 | 0.807 ± 0.004 |
| NaiveBayes Platt | 0.184 ± 0.007 | 0.245 ± 0.011 | 0.11 ± 0.005 | 0.142 ± 0.005 | 0.807 ± 0.004 |
| NaiveBayes Temp | 0.07 ± 0.02 | 0.101 ± 0.024 | 0.068 ± 0.02 | 0.098 ± 0.026 | 0.807 ± 0.004 |
| NaiveBayes Isotonic | **0.008 ± 0.002** | **0.076 ± 0.017** | **0.012 ± 0.001** | **0.049 ± 0.002** | **0.81 ± 0.001** |

Figure 39: Credit Default. Training data reused for post-processing.

| Model | ECE ↓ | Max ECE ↓ | smECE ↓ | Max smECE ↓ | Acc ↑ |
|---|---|---|---|---|---|
| MLP ERM | 0.024 ± 0.006 | 0.107 ± 0.018 | 0.026 ± 0.004 | 0.1 ± 0.025 | **0.865 ± 0.002** |
| MLP HKRR | 0.024 ± 0.006 | 0.109 ± 0.022 | 0.024 ± 0.006 | 0.096 ± 0.016 | 0.862 ± 0.003 |
| MLP HJZ | 0.018 ± 0.001 | 0.105 ± 0.011 | 0.021 ± 0.002 | 0.093 ± 0.021 | 0.864 ± 0.003 |
| MLP Platt | 0.019 ± 0.003 | 0.096 ± 0.016 | 0.02 ± 0.002 | 0.084 ± 0.017 | **0.865 ± 0.002** |
| MLP Temp | 0.024 ± 0.006 | 0.107 ± 0.018 | 0.026 ± 0.004 | 0.1 ± 0.025 | **0.865 ± 0.002** |
| MLP Isotonic | **0.017 ± 0.004** | **0.081 ± 0.008** | **0.019 ± 0.002** | **0.07 ± 0.009** | 0.863 ± 0.002 |
| RandomForest ERM | **0.017 ± 0.001** | 0.091 ± 0.005 | **0.02 ± 0.001** | 0.082 ± 0.005 | 0.862 ± 0.002 |
| RandomForest HKRR | 0.089 ± 0.004 | 0.25 ± 0.026 | 0.088 ± 0.004 | 0.221 ± 0.025 | 0.848 ± 0.002 |
| RandomForest HJZ | 0.022 ± 0.003 | **0.088 ± 0.007** | 0.024 ± 0.001 | 0.083 ± 0.004 | 0.862 ± 0.002 |
| RandomForest Platt | 0.027 ± 0.002 | 0.09 ± 0.003 | 0.029 ± 0.001 | **0.08 ± 0.006** | **0.863 ± 0.001** |
| RandomForest Temp | 0.044 ± 0.001 | 0.102 ± 0.01 | 0.043 ± 0.001 | 0.089 ± 0.003 | 0.862 ± 0.002 |
| RandomForest Isotonic | 0.075 ± 0.002 | 0.167 ± 0.005 | 0.058 ± 0.002 | 0.121 ± 0.007 | 0.86 ± 0.003 |
| SVM ERM | 0.149 ± 0.015 | 0.359 ± 0.04 | 0.075 ± 0.008 | 0.179 ± 0.019 | 0.851 ± 0.015 |
| SVM HKRR | 0.018 ± 0.005 | **0.111 ± 0.013** | 0.018 ± 0.005 | **0.101 ± 0.006** | 0.853 ± 0.01 |
| SVM HJZ | 0.06 ± 0.007 | 0.211 ± 0.069 | 0.048 ± 0.009 | 0.183 ± 0.055 | **0.857 ± 0.006** |
| SVM Platt | 0.149 ± 0.015 | 0.359 ± 0.04 | 0.075 ± 0.008 | 0.179 ± 0.019 | 0.851 ± 0.015 |
| SVM Temp | 0.111 ± 0.039 | 0.198 ± 0.033 | 0.11 ± 0.038 | 0.194 ± 0.03 | 0.851 ± 0.015 |
| SVM Isotonic | **0.007 ± 0.004** | 0.202 ± 0.057 | **0.007 ± 0.004** | 0.195 ± 0.051 | 0.852 ± 0.014 |
| LogisticRegression ERM | 0.022 ± 0.002 | 0.106 ± 0.008 | 0.022 ± 0.001 | 0.083 ± 0.003 | **0.866 ± 0.002** |
| LogisticRegression HKRR | 0.024 ± 0.001 | 0.12 ± 0.011 | 0.023 ± 0.001 | 0.105 ± 0.009 | 0.861 ± 0.003 |
| LogisticRegression HJZ | 0.022 ± 0.003 | 0.106 ± 0.011 | 0.022 ± 0.002 | 0.083 ± 0.005 | 0.866 ± 0.001 |
| LogisticRegression Platt | 0.016 ± 0.004 | 0.092 ± 0.009 | 0.021 ± 0.002 | 0.078 ± 0.003 | 0.863 ± 0.001 |
| LogisticRegression Temp | 0.049 ± 0.002 | 0.112 ± 0.008 | 0.046 ± 0.001 | 0.087 ± 0.003 | **0.866 ± 0.002** |
| LogisticRegression Isotonic | **0.014 ± 0.002** | **0.087 ± 0.01** | **0.018 ± 0.002** | **0.073 ± 0.002** | 0.866 ± 0.001 |
| DecisionTree ERM | **0.067 ± 0.006** | 0.266 ± 0.029 | **0.048 ± 0.004** | 0.17 ± 0.013 | 0.849 ± 0.007 |
| DecisionTree HKRR | 0.074 ± 0.007 | 0.282 ± 0.034 | 0.073 ± 0.007 | 0.237 ± 0.022 | 0.845 ± 0.007 |
| DecisionTree HJZ | 0.068 ± 0.007 | 0.264 ± 0.035 | 0.049 ± 0.005 | 0.167 ± 0.018 | **0.849 ± 0.007** |
| DecisionTree Platt | 0.069 ± 0.008 | 0.267 ± 0.03 | 0.049 ± 0.005 | 0.171 ± 0.013 | 0.849 ± 0.007 |
| DecisionTree Temp | 0.095 ± 0.004 | **0.175 ± 0.019** | 0.091 ± 0.001 | **0.155 ± 0.01** | 0.849 ± 0.007 |
| DecisionTree Isotonic | **0.067 ± 0.006** | 0.266 ± 0.029 | **0.048 ± 0.004** | 0.17 ± 0.013 | 0.849 ± 0.007 |
| NaiveBayes ERM | 0.277 ± 0.019 | 0.544 ± 0.02 | 0.164 ± 0.013 | 0.287 ± 0.011 | 0.714 ± 0.018 |
| NaiveBayes HKRR | 0.023 ± 0.003 | **0.119 ± 0.026** | 0.023 ± 0.003 | **0.102 ± 0.019** | 0.833 ± 0.005 |
| NaiveBayes HJZ | 0.05 ± 0.017 | 0.205 ± 0.038 | 0.044 ± 0.014 | 0.183 ± 0.038 | 0.803 ± 0.004 |
| NaiveBayes Platt | 0.275 ± 0.018 | 0.54 ± 0.019 | 0.169 ± 0.011 | 0.295 ± 0.009 | 0.714 ± 0.018 |
| NaiveBayes Temp | 0.3 ± 0.007 | 0.373 ± 0.016 | 0.278 ± 0.005 | 0.326 ± 0.01 | 0.714 ± 0.018 |
| NaiveBayes Isotonic | **0.014 ± 0.002** | 0.121 ± 0.009 | **0.017 ± 0.001** | 0.111 ± 0.005 | **0.834 ± 0.006** |

Figure 40: MEPS. Training data reused for post-processing.

### H.4.1 Comparing Multicalibration Post-Processing Performance with Data Reuse

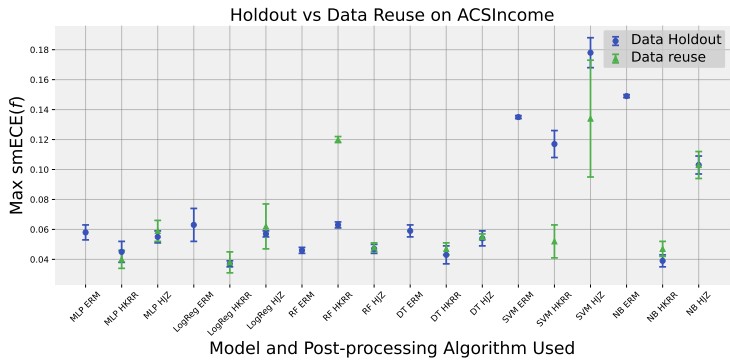

Figure 41: Data reuse comparison for ACSIncome.

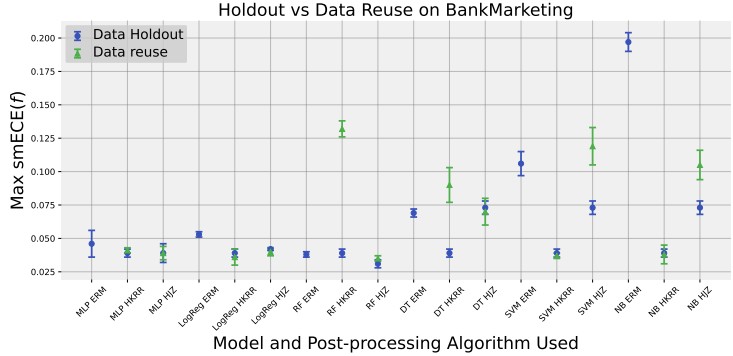

Figure 42: Data reuse comparison for BankMarketing.

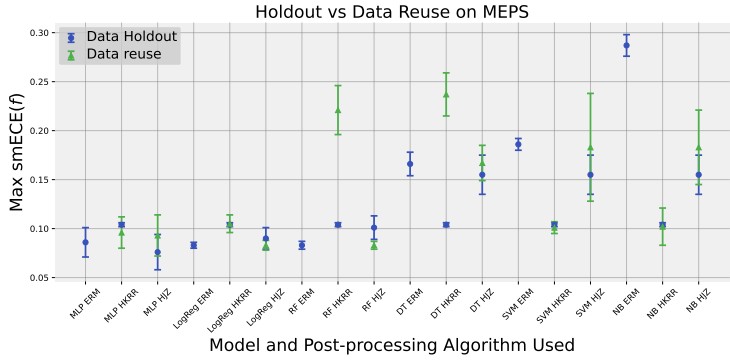

Figure 43: Data reuse comparison for MEPS.

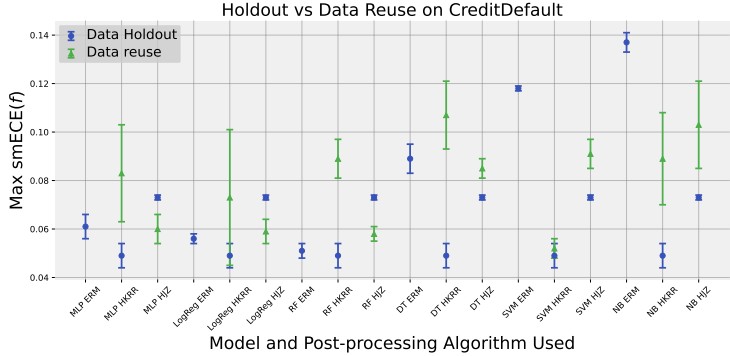

Figure 44: Data reuse comparison for CreditDefault.

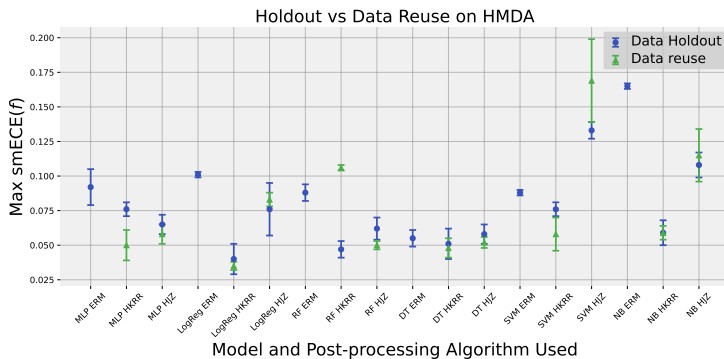

Figure 45: Data reuse comparison for HMDA.

# I Results on Tabular Datasets with Alternate Groups

## I.1 Plots for All Multicalibration Algorithms

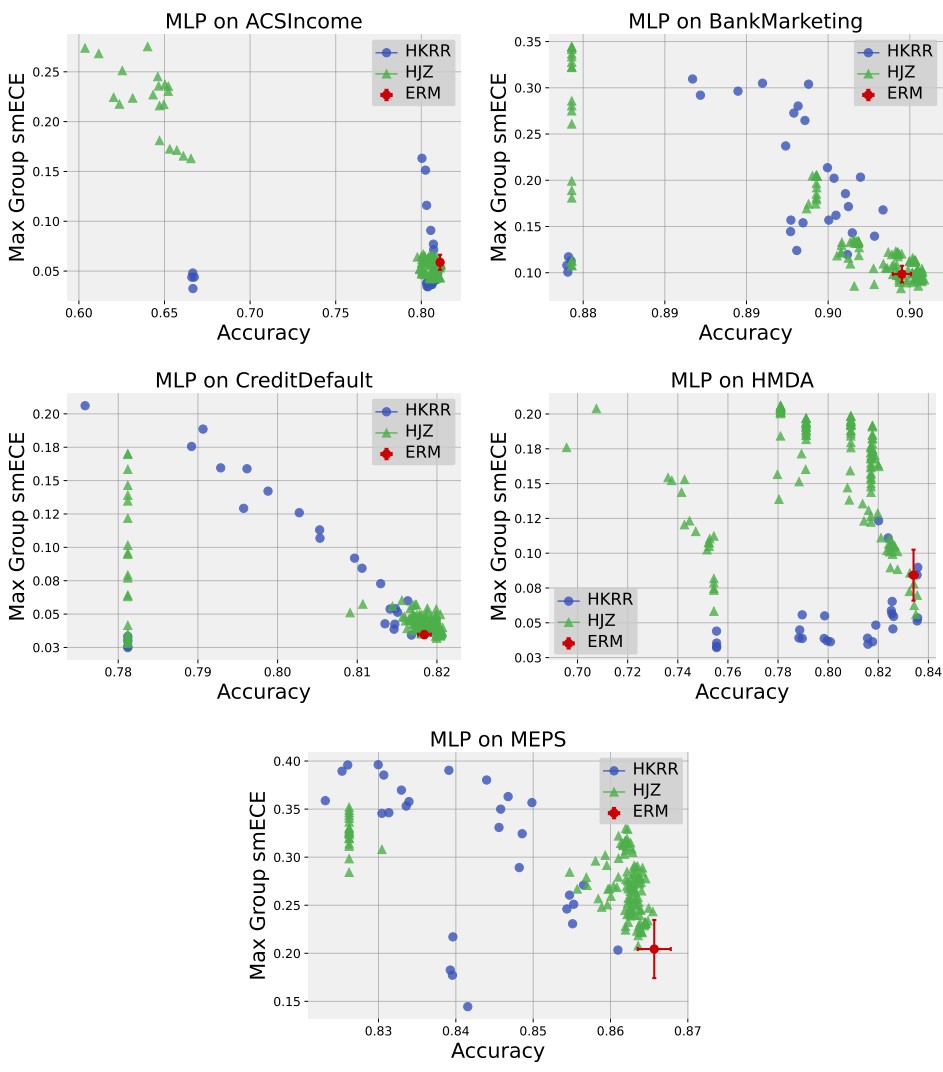

Figure 46: All multicalibration algorithms on MLPs. Alternate groups.

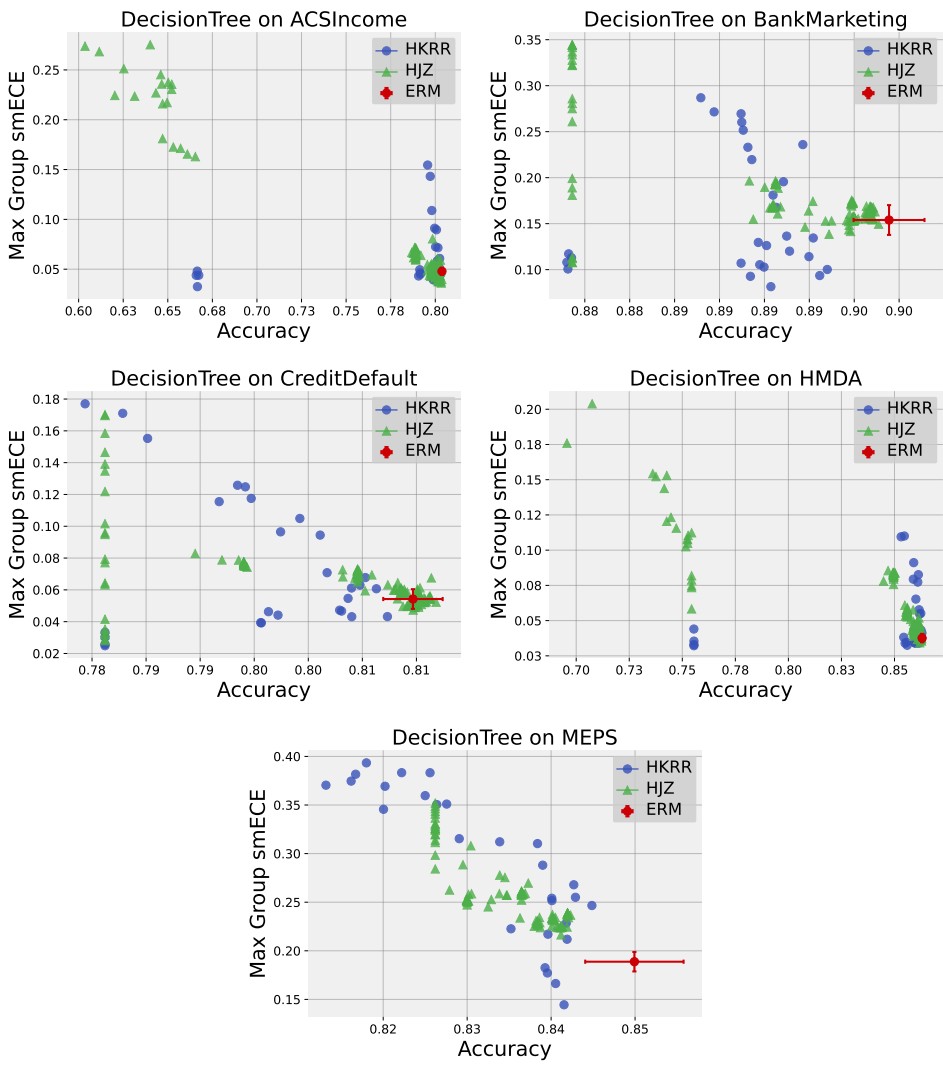

Figure 47: All multicalibration algorithms on Decision Trees. Alternate groups.

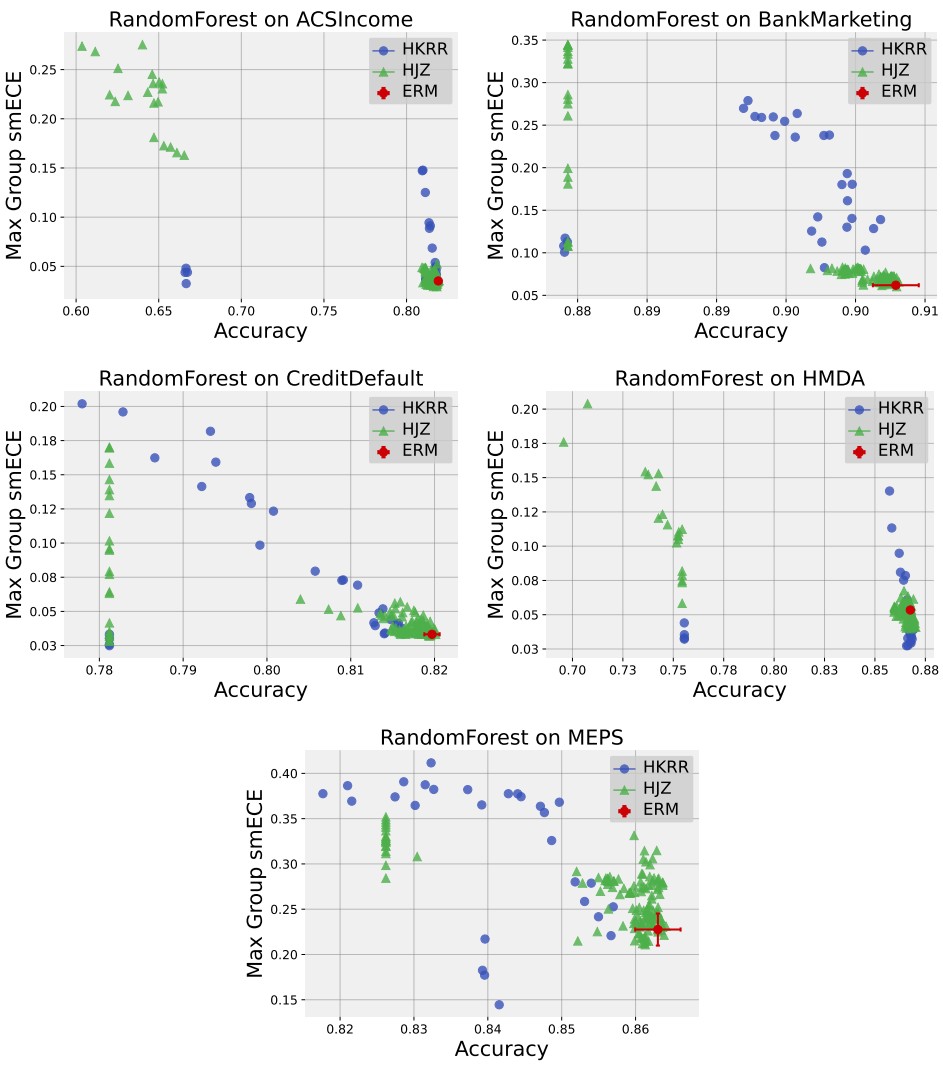

Figure 48: All multicalibration algorithms on Random Forest. Alternate groups.

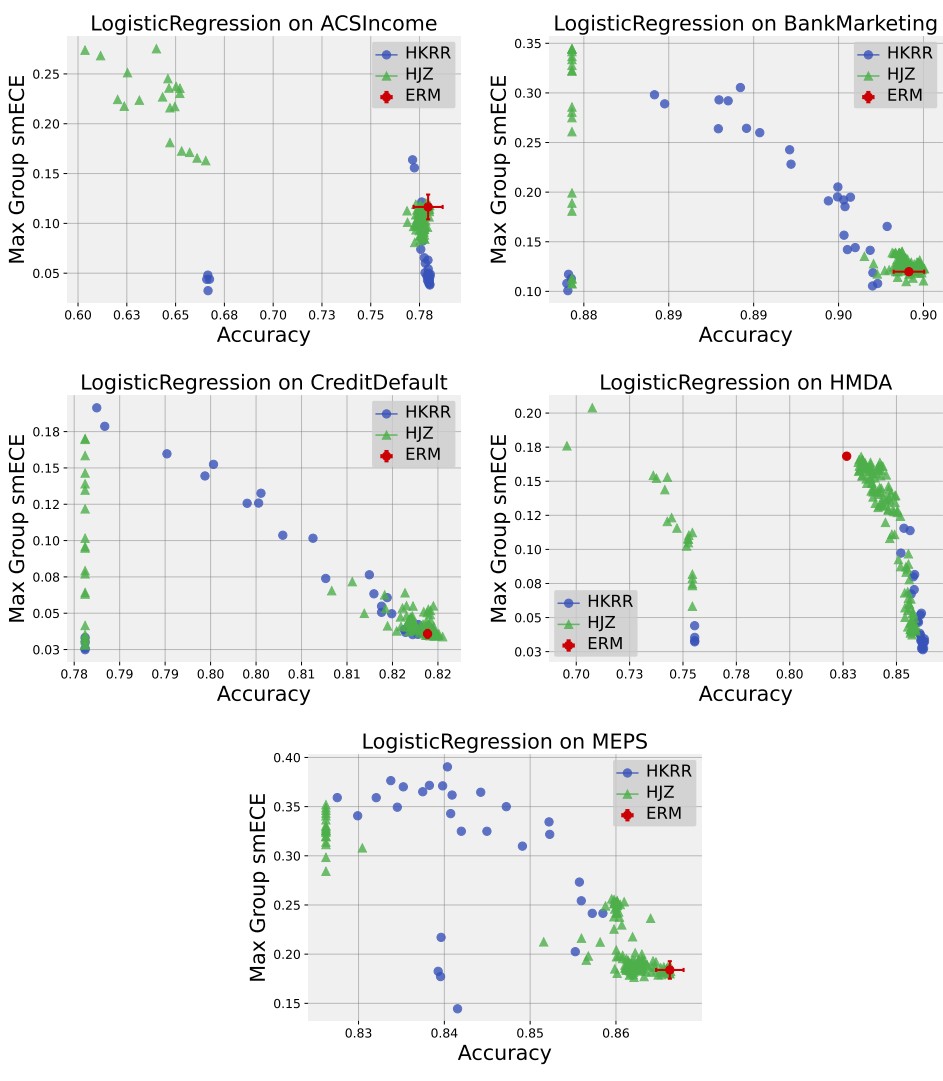

Figure 49: All multicalibration algorithms on Logistic Regression. Alternate groups.

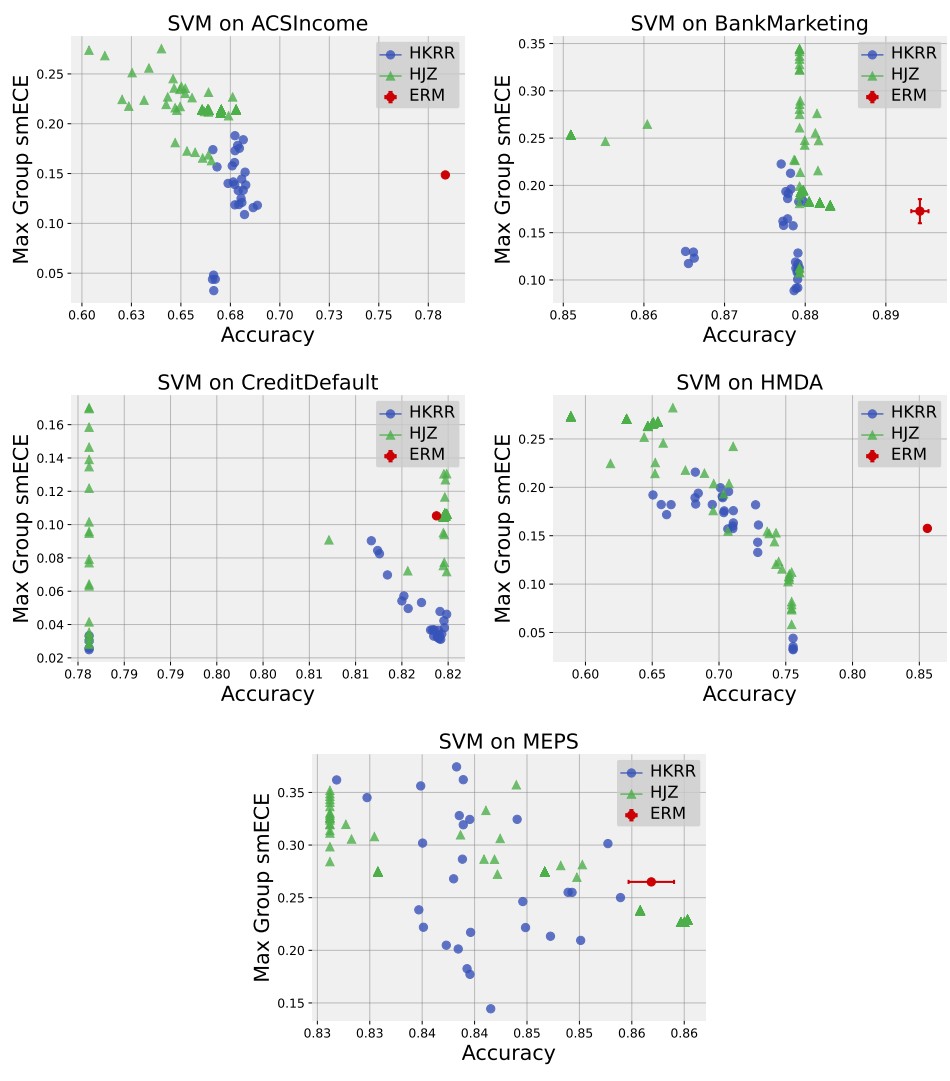

Figure 50: All multicalibration algorithms on SVMs. Alternate groups.

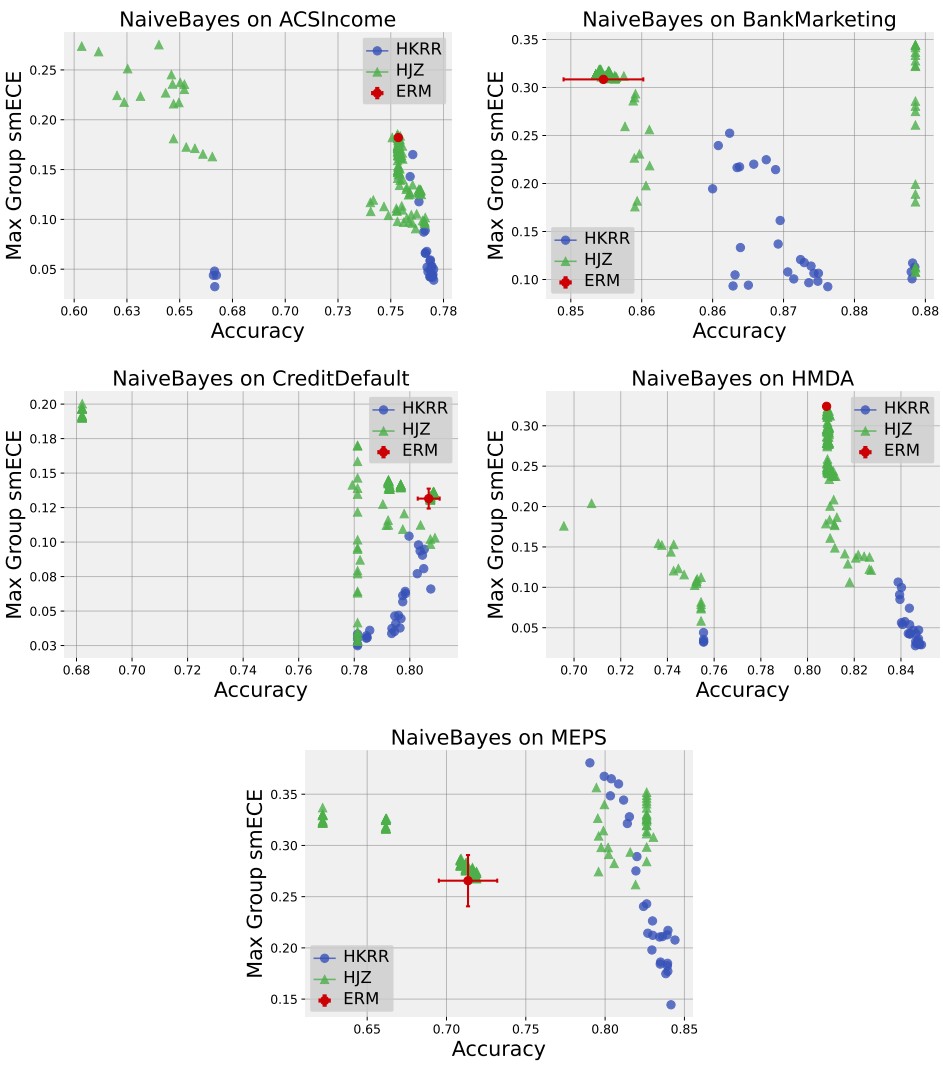

Figure 51: All multicalibration algorithms on Naive Bayes. Alternate groups.

## I.2 Tables Comparing Best-Performing Multicalibration Algorithms with ERM (Alternate Groups)

| Model | ECE ↓ | Max ECE ↓ | smECE ↓ | Max smECE ↓ | Acc ↑ |
|---|---|---|---|---|---|
| MLP ERM | 0.014 ± 0.005 | 0.059 ± 0.007 | 0.015 ± 0.004 | 0.059 ± 0.008 | 0.811 ± 0.002 |
| MLP HKRR | **0.009 ± 0.003** | **0.037 ± 0.005** | **0.009 ± 0.003** | **0.035 ± 0.003** | 0.803 ± 0.001 |
| MLP HJZ | 0.019 ± 0.003 | 0.053 ± 0.01 | 0.02 ± 0.002 | 0.051 ± 0.011 | 0.807 ± 0.004 |
| MLP Platt | 0.012 ± 0.004 | 0.046 ± 0.008 | 0.014 ± 0.002 | 0.044 ± 0.007 | 0.811 ± 0.001 |
| MLP Temp | 0.012 ± 0.005 | 0.056 ± 0.005 | 0.013 ± 0.004 | 0.056 ± 0.005 | 0.811 ± 0.001 |
| MLP Isotonic | 0.012 ± 0.002 | 0.055 ± 0.005 | 0.013 ± 0.001 | 0.054 ± 0.006 | **0.811 ± 0.001** |
| RandomForest ERM | 0.01 ± 0.001 | 0.036 ± 0.001 | 0.011 ± 0.0 | 0.035 ± 0.001 | 0.819 ± 0.001 |
| RandomForest HKRR | 0.007 ± 0.001 | 0.043 ± 0.011 | **0.007 ± 0.001** | 0.042 ± 0.01 | 0.818 ± 0.0 |
| RandomForest HJZ | **0.007 ± 0.001** | 0.032 ± 0.005 | 0.011 ± 0.001 | 0.031 ± 0.003 | 0.813 ± 0.007 |
| RandomForest Platt | 0.009 ± 0.001 | **0.031 ± 0.003** | 0.011 ± 0.001 | **0.029 ± 0.002** | 0.816 ± 0.004 |
| RandomForest Temp | 0.027 ± 0.001 | 0.074 ± 0.001 | 0.027 ± 0.001 | 0.073 ± 0.0 | **0.819 ± 0.001** |
| RandomForest Isotonic | 0.008 ± 0.001 | 0.033 ± 0.001 | 0.011 ± 0.0 | 0.031 ± 0.001 | 0.818 ± 0.001 |
| SVM ERM | 0.216 ± 0.001 | 0.292 ± 0.004 | 0.109 ± 0.0 | 0.149 ± 0.002 | **0.784 ± 0.001** |
| SVM HKRR | **0.008 ± 0.0** | **0.034 ± 0.005** | **0.008 ± 0.0** | **0.032 ± 0.004** | 0.667 ± 0.0 |
| SVM HJZ | 0.013 ± 0.002 | 0.164 ± 0.009 | 0.019 ± 0.001 | 0.163 ± 0.009 | 0.665 ± 0.0 |
| SVM Platt | 0.336 ± 0.007 | 0.438 ± 0.0 | 0.168 ± 0.004 | 0.214 ± 0.0 | 0.664 ± 0.007 |
| SVM Temp | 0.099 ± 0.006 | 0.216 ± 0.0 | 0.099 ± 0.006 | 0.211 ± 0.0 | 0.678 ± 0.006 |
| SVM Isotonic | 0.081 ± 0.012 | 0.263 ± 0.014 | 0.081 ± 0.012 | 0.25 ± 0.011 | 0.664 ± 0.007 |
| LogisticRegression ERM | 0.012 ± 0.002 | 0.123 ± 0.014 | 0.015 ± 0.002 | 0.116 ± 0.012 | 0.779 ± 0.007 |
| LogisticRegression HKRR | 0.006 ± 0.002 | **0.041 ± 0.003** | **0.006 ± 0.002** | **0.038 ± 0.002** | **0.78 ± 0.007** |
| LogisticRegression HJZ | 0.011 ± 0.001 | 0.085 ± 0.013 | 0.014 ± 0.001 | 0.083 ± 0.012 | 0.775 ± 0.009 |
| LogisticRegression Platt | 0.021 ± 0.004 | 0.108 ± 0.017 | 0.021 ± 0.004 | 0.105 ± 0.015 | 0.774 ± 0.012 |
| LogisticRegression Temp | 0.019 ± 0.0 | 0.115 ± 0.004 | 0.02 ± 0.0 | 0.111 ± 0.005 | 0.776 ± 0.009 |
| LogisticRegression Isotonic | **0.005 ± 0.001** | 0.109 ± 0.007 | 0.009 ± 0.001 | 0.105 ± 0.004 | 0.775 ± 0.009 |
| DecisionTree ERM | 0.017 ± 0.001 | 0.051 ± 0.004 | 0.016 ± 0.001 | 0.048 ± 0.004 | **0.804 ± 0.0** |
| DecisionTree HKRR | 0.008 ± 0.001 | 0.041 ± 0.004 | **0.008 ± 0.001** | 0.039 ± 0.003 | 0.799 ± 0.001 |
| DecisionTree HJZ | 0.019 ± 0.001 | 0.049 ± 0.002 | 0.017 ± 0.002 | 0.046 ± 0.002 | 0.802 ± 0.001 |
| DecisionTree Platt | 0.011 ± 0.001 | **0.041 ± 0.006** | 0.013 ± 0.001 | **0.037 ± 0.005** | 0.803 ± 0.0 |
| DecisionTree Temp | 0.028 ± 0.002 | 0.073 ± 0.001 | 0.027 ± 0.002 | 0.072 ± 0.001 | 0.803 ± 0.001 |
| DecisionTree Isotonic | **0.007 ± 0.002** | 0.054 ± 0.003 | 0.01 ± 0.001 | 0.051 ± 0.003 | 0.803 ± 0.001 |
| NaiveBayes ERM | 0.117 ± 0.0 | 0.201 ± 0.001 | 0.109 ± 0.0 | 0.182 ± 0.0 | 0.754 ± 0.0 |
| NaiveBayes HKRR | **0.006 ± 0.001** | **0.042 ± 0.005** | **0.006 ± 0.001** | **0.039 ± 0.004** | **0.77 ± 0.001** |
| NaiveBayes HJZ | 0.017 ± 0.002 | 0.093 ± 0.006 | 0.021 ± 0.001 | 0.091 ± 0.007 | 0.762 ± 0.004 |
| NaiveBayes Platt | 0.085 ± 0.004 | 0.165 ± 0.007 | 0.08 ± 0.004 | 0.161 ± 0.006 | 0.756 ± 0.002 |
| NaiveBayes Temp | 0.079 ± 0.002 | 0.182 ± 0.002 | 0.069 ± 0.001 | 0.18 ± 0.002 | 0.754 ± 0.001 |
| NaiveBayes Isotonic | 0.008 ± 0.002 | 0.105 ± 0.001 | 0.011 ± 0.002 | 0.103 ± 0.001 | 0.768 ± 0.0 |

Figure 52: ACS Income. Alternate groups.

| Model | ECE ↓ | Max ECE ↓ | smECE ↓ | Max smECE ↓ | Acc ↑ |
|---|---|---|---|---|---|
| MLP ERM | 0.008 ± 0.003 | 0.14 ± 0.018 | 0.012 ± 0.002 | 0.099 ± 0.009 | 0.9 ± 0.001 |
| MLP HKRR | 0.102 ± 0.003 | 0.127 ± 0.013 | 0.098 ± 0.001 | 0.117 ± 0.016 | 0.879 ± 0.0 |
| MLP HJZ | 0.019 ± 0.012 | 0.137 ± 0.016 | 0.021 ± 0.011 | 0.09 ± 0.008 | **0.901 ± 0.001** |
| MLP Platt | 0.011 ± 0.004 | 0.138 ± 0.022 | 0.014 ± 0.003 | 0.105 ± 0.017 | 0.896 ± 0.005 |
| MLP Temp | 0.042 ± 0.01 | **0.123 ± 0.024** | 0.042 ± 0.01 | **0.074 ± 0.007** | 0.901 ± 0.001 |
| MLP Isotonic | **0.008 ± 0.002** | 0.126 ± 0.024 | **0.01 ± 0.001** | 0.091 ± 0.009 | 0.9 ± 0.001 |
| RandomForest ERM | 0.014 ± 0.001 | 0.095 ± 0.01 | 0.015 ± 0.0 | 0.062 ± 0.001 | 0.903 ± 0.002 |
| RandomForest HKRR | 0.012 ± 0.002 | 0.106 ± 0.017 | 0.012 ± 0.002 | 0.082 ± 0.008 | 0.898 ± 0.001 |
| RandomForest HJZ | 0.011 ± 0.003 | 0.108 ± 0.022 | 0.013 ± 0.002 | 0.066 ± 0.014 | **0.903 ± 0.001** |
| RandomForest Platt | 0.009 ± 0.002 | **0.095 ± 0.022** | 0.012 ± 0.001 | **0.06 ± 0.006** | 0.903 ± 0.001 |
| RandomForest Temp | 0.057 ± 0.001 | 0.117 ± 0.019 | 0.054 ± 0.001 | 0.083 ± 0.002 | 0.903 ± 0.001 |
| RandomForest Isotonic | **0.008 ± 0.002** | 0.116 ± 0.024 | **0.011 ± 0.001** | 0.083 ± 0.009 | 0.901 ± 0.001 |
| SVM ERM | 0.106 ± 0.001 | 0.347 ± 0.027 | 0.053 ± 0.001 | 0.173 ± 0.013 | **0.894 ± 0.001** |
| SVM HKRR | 0.051 ± 0.014 | **0.108 ± 0.038** | 0.051 ± 0.014 | **0.091 ± 0.028** | 0.879 ± 0.0 |
| SVM HJZ | 0.106 ± 0.001 | 0.134 ± 0.003 | 0.098 ± 0.0 | 0.112 ± 0.004 | 0.879 ± 0.0 |
| SVM Platt | 0.117 ± 0.001 | 0.36 ± 0.009 | 0.059 ± 0.001 | 0.178 ± 0.004 | 0.883 ± 0.001 |
| SVM Temp | 0.152 ± 0.003 | 0.225 ± 0.0 | 0.151 ± 0.003 | 0.219 ± 0.0 | 0.88 ± 0.001 |
| SVM Isotonic | **0.023 ± 0.009** | 0.247 ± 0.025 | **0.023 ± 0.009** | 0.237 ± 0.021 | 0.88 ± 0.001 |
| LogisticRegression ERM | 0.032 ± 0.001 | 0.154 ± 0.006 | 0.03 ± 0.001 | 0.12 ± 0.002 | 0.899 ± 0.001 |
| LogisticRegression HKRR | 0.018 ± 0.002 | 0.132 ± 0.027 | 0.018 ± 0.002 | 0.108 ± 0.027 | 0.897 ± 0.002 |
| LogisticRegression HJZ | 0.106 ± 0.001 | 0.134 ± 0.003 | 0.098 ± 0.0 | 0.112 ± 0.004 | 0.879 ± 0.0 |
| LogisticRegression Platt | 0.023 ± 0.005 | 0.156 ± 0.009 | 0.023 ± 0.004 | 0.129 ± 0.012 | 0.897 ± 0.002 |
| LogisticRegression Temp | 0.061 ± 0.001 | 0.174 ± 0.02 | 0.056 ± 0.0 | 0.121 ± 0.005 | 0.899 ± 0.001 |
| LogisticRegression Isotonic | **0.008 ± 0.001** | **0.114 ± 0.012** | **0.011 ± 0.002** | **0.093 ± 0.006** | **0.9 ± 0.001** |
| DecisionTree ERM | 0.028 ± 0.002 | 0.213 ± 0.021 | 0.022 ± 0.001 | 0.154 ± 0.016 | **0.897 ± 0.002** |
| DecisionTree HKRR | 0.102 ± 0.003 | **0.127 ± 0.013** | 0.098 ± 0.001 | 0.117 ± 0.016 | 0.879 ± 0.0 |
| DecisionTree HJZ | 0.106 ± 0.001 | 0.134 ± 0.003 | 0.098 ± 0.0 | **0.112 ± 0.004** | 0.879 ± 0.0 |
| DecisionTree Platt | 0.025 ± 0.006 | 0.214 ± 0.044 | 0.021 ± 0.002 | 0.153 ± 0.019 | 0.897 ± 0.002 |
| DecisionTree Temp | 0.116 ± 0.001 | 0.173 ± 0.013 | 0.114 ± 0.001 | 0.163 ± 0.003 | 0.896 ± 0.002 |
| DecisionTree Isotonic | **0.01 ± 0.002** | 0.157 ± 0.011 | **0.011 ± 0.002** | 0.139 ± 0.014 | 0.896 ± 0.002 |
| NaiveBayes ERM | 0.122 ± 0.003 | 0.521 ± 0.002 | 0.093 ± 0.002 | 0.308 ± 0.002 | 0.857 ± 0.003 |
| NaiveBayes HKRR | 0.037 ± 0.005 | 0.121 ± 0.024 | 0.036 ± 0.004 | 0.106 ± 0.016 | 0.872 ± 0.003 |
| NaiveBayes HJZ | 0.106 ± 0.001 | 0.134 ± 0.003 | 0.098 ± 0.0 | 0.112 ± 0.004 | 0.879 ± 0.0 |
| NaiveBayes Platt | 0.122 ± 0.005 | 0.528 ± 0.01 | 0.094 ± 0.004 | 0.318 ± 0.012 | 0.857 ± 0.004 |
| NaiveBayes Temp | 0.217 ± 0.002 | 0.293 ± 0.009 | 0.212 ± 0.002 | 0.268 ± 0.009 | 0.857 ± 0.004 |
| NaiveBayes Isotonic | **0.007 ± 0.001** | **0.118 ± 0.011** | **0.01 ± 0.001** | **0.095 ± 0.012** | **0.885 ± 0.002** |

Figure 53: Bank Marketing. Alternate groups.

| Model | ECE ↓ | Max ECE ↓ | smECE ↓ | Max smECE ↓ | Acc ↑ |
|---|---|---|---|---|---|
| MLP ERM | 0.018 ± 0.005 | 0.039 ± 0.004 | 0.019 ± 0.004 | 0.035 ± 0.003 | 0.818 ± 0.001 |
| MLP HKRR | 0.028 ± 0.003 | **0.026 ± 0.003** | 0.026 ± 0.002 | **0.025 ± 0.002** | 0.781 ± 0.0 |
| MLP HJZ | 0.033 ± 0.002 | 0.036 ± 0.008 | 0.03 ± 0.001 | 0.028 ± 0.002 | 0.781 ± 0.0 |
| MLP Platt | 0.013 ± 0.004 | 0.043 ± 0.008 | 0.015 ± 0.002 | 0.038 ± 0.008 | **0.82 ± 0.0** |
| MLP Temp | 0.016 ± 0.004 | 0.043 ± 0.007 | 0.018 ± 0.003 | 0.033 ± 0.002 | 0.819 ± 0.001 |
| MLP Isotonic | **0.011 ± 0.003** | 0.043 ± 0.005 | **0.014 ± 0.001** | 0.035 ± 0.004 | 0.818 ± 0.001 |
| RandomForest ERM | 0.019 ± 0.001 | 0.035 ± 0.003 | 0.02 ± 0.001 | 0.033 ± 0.001 | **0.82 ± 0.001** |
| RandomForest HKRR | 0.028 ± 0.003 | **0.026 ± 0.003** | 0.026 ± 0.002 | **0.025 ± 0.002** | 0.781 ± 0.0 |
| RandomForest HJZ | 0.033 ± 0.002 | 0.036 ± 0.008 | 0.03 ± 0.001 | 0.028 ± 0.002 | 0.781 ± 0.0 |
| RandomForest Platt | **0.013 ± 0.002** | 0.04 ± 0.007 | 0.016 ± 0.002 | 0.035 ± 0.003 | 0.819 ± 0.001 |
| RandomForest Temp | 0.023 ± 0.003 | 0.04 ± 0.003 | 0.024 ± 0.002 | 0.037 ± 0.003 | 0.819 ± 0.001 |
| RandomForest Isotonic | 0.013 ± 0.003 | 0.035 ± 0.002 | **0.015 ± 0.002** | 0.031 ± 0.002 | 0.819 ± 0.001 |
| SVM ERM | 0.181 ± 0.0 | 0.21 ± 0.001 | 0.091 ± 0.0 | 0.105 ± 0.0 | 0.819 ± 0.0 |
| SVM HKRR | 0.02 ± 0.001 | 0.046 ± 0.004 | 0.016 ± 0.001 | 0.031 ± 0.003 | 0.819 ± 0.001 |
| SVM HJZ | 0.033 ± 0.002 | 0.036 ± 0.008 | 0.03 ± 0.001 | 0.028 ± 0.002 | 0.781 ± 0.0 |
| SVM Platt | 0.18 ± 0.001 | 0.213 ± 0.002 | 0.09 ± 0.0 | 0.106 ± 0.001 | **0.82 ± 0.001** |
| SVM Temp | 0.022 ± 0.0 | 0.052 ± 0.001 | 0.022 ± 0.0 | 0.051 ± 0.001 | 0.82 ± 0.0 |
| SVM Isotonic | **0.006 ± 0.001** | **0.026 ± 0.002** | **0.006 ± 0.001** | **0.026 ± 0.002** | **0.82 ± 0.001** |
| LogisticRegression ERM | **0.01 ± 0.001** | 0.042 ± 0.004 | 0.015 ± 0.001 | 0.036 ± 0.003 | 0.819 ± 0.0 |
| LogisticRegression HKRR | 0.028 ± 0.003 | **0.026 ± 0.003** | 0.026 ± 0.002 | **0.025 ± 0.002** | 0.781 ± 0.0 |
| LogisticRegression HJZ | 0.033 ± 0.002 | 0.036 ± 0.008 | 0.03 ± 0.001 | 0.028 ± 0.002 | 0.781 ± 0.0 |
| LogisticRegression Platt | 0.014 ± 0.001 | 0.045 ± 0.004 | 0.016 ± 0.002 | 0.037 ± 0.003 | 0.819 ± 0.0 |
| LogisticRegression Temp | 0.023 ± 0.001 | 0.041 ± 0.006 | 0.023 ± 0.001 | 0.037 ± 0.001 | 0.819 ± 0.0 |
| LogisticRegression Isotonic | 0.01 ± 0.002 | 0.04 ± 0.009 | **0.014 ± 0.001** | 0.034 ± 0.006 | **0.82 ± 0.001** |
| DecisionTree ERM | 0.04 ± 0.003 | 0.078 ± 0.011 | 0.031 ± 0.001 | 0.054 ± 0.006 | 0.81 ± 0.003 |
| DecisionTree HKRR | 0.028 ± 0.003 | **0.026 ± 0.003** | 0.026 ± 0.002 | **0.025 ± 0.002** | 0.781 ± 0.0 |
| DecisionTree HJZ | 0.033 ± 0.002 | 0.036 ± 0.008 | 0.03 ± 0.001 | 0.028 ± 0.002 | 0.781 ± 0.0 |
| DecisionTree Platt | 0.033 ± 0.003 | 0.062 ± 0.005 | 0.028 ± 0.002 | 0.05 ± 0.006 | 0.81 ± 0.003 |
| DecisionTree Temp | 0.031 ± 0.004 | 0.062 ± 0.009 | 0.029 ± 0.003 | 0.058 ± 0.01 | **0.811 ± 0.003** |
| DecisionTree Isotonic | **0.007 ± 0.001** | 0.04 ± 0.006 | **0.009 ± 0.003** | 0.036 ± 0.007 | 0.797 ± 0.004 |
| NaiveBayes ERM | 0.187 ± 0.006 | 0.226 ± 0.005 | 0.108 ± 0.005 | 0.132 ± 0.007 | 0.807 ± 0.004 |
| NaiveBayes HKRR | 0.028 ± 0.013 | **0.035 ± 0.008** | 0.026 ± 0.013 | 0.03 ± 0.006 | 0.784 ± 0.02 |
| NaiveBayes HJZ | 0.033 ± 0.002 | 0.036 ± 0.008 | 0.03 ± 0.001 | **0.028 ± 0.002** | 0.781 ± 0.0 |
| NaiveBayes Platt | 0.197 ± 0.011 | 0.236 ± 0.013 | 0.119 ± 0.015 | 0.143 ± 0.023 | 0.792 ± 0.014 |
| NaiveBayes Temp | 0.037 ± 0.012 | 0.075 ± 0.007 | 0.037 ± 0.012 | 0.073 ± 0.008 | 0.809 ± 0.003 |
| NaiveBayes Isotonic | **0.013 ± 0.004** | 0.039 ± 0.007 | **0.016 ± 0.003** | 0.037 ± 0.005 | **0.809 ± 0.001** |

Figure 54: Credit Default. Alternate groups.

| Model | ECE ↓ | Max ECE ↓ | smECE ↓ | Max smECE ↓ | Acc ↑ |
|---|---|---|---|---|---|
| MLP ERM | 0.045 ± 0.015 | 0.087 ± 0.018 | 0.043 ± 0.014 | 0.084 ± 0.018 | 0.834 ± 0.002 |
| MLP HKRR | **0.004 ± 0.001** | **0.034 ± 0.006** | **0.004 ± 0.001** | **0.033 ± 0.006** | 0.756 ± 0.0 |
| MLP HJZ | 0.012 ± 0.001 | 0.062 ± 0.021 | 0.016 ± 0.001 | 0.056 ± 0.019 | **0.835 ± 0.009** |
| MLP Platt | 0.184 ± 0.009 | 0.272 ± 0.025 | 0.14 ± 0.002 | 0.206 ± 0.017 | 0.781 ± 0.002 |
| MLP Temp | 0.047 ± 0.033 | 0.097 ± 0.02 | 0.046 ± 0.032 | 0.094 ± 0.02 | 0.817 ± 0.008 |
| MLP Isotonic | 0.013 ± 0.002 | 0.1 ± 0.009 | 0.014 ± 0.002 | 0.092 ± 0.008 | 0.824 ± 0.006 |
| RandomForest ERM | 0.038 ± 0.002 | 0.054 ± 0.001 | 0.038 ± 0.002 | 0.053 ± 0.001 | 0.868 ± 0.001 |
| RandomForest HKRR | **0.006 ± 0.001** | **0.028 ± 0.003** | **0.006 ± 0.001** | **0.027 ± 0.003** | 0.866 ± 0.001 |
| RandomForest HJZ | 0.007 ± 0.002 | 0.044 ± 0.008 | 0.011 ± 0.002 | 0.041 ± 0.007 | 0.869 ± 0.001 |
| RandomForest Platt | 0.009 ± 0.001 | 0.06 ± 0.007 | 0.011 ± 0.001 | 0.054 ± 0.005 | 0.867 ± 0.001 |
| RandomForest Temp | 0.037 ± 0.001 | 0.072 ± 0.004 | 0.039 ± 0.001 | 0.07 ± 0.004 | 0.866 ± 0.001 |
| RandomForest Isotonic | 0.009 ± 0.002 | 0.048 ± 0.002 | 0.01 ± 0.002 | 0.046 ± 0.003 | **0.869 ± 0.001** |
| SVM ERM | 0.144 ± 0.001 | 0.31 ± 0.004 | 0.072 ± 0.0 | 0.158 ± 0.002 | **0.856 ± 0.001** |
| SVM HKRR | 0.004 ± 0.001 | **0.034 ± 0.006** | 0.004 ± 0.001 | **0.033 ± 0.006** | 0.756 ± 0.0 |
| SVM HJZ | 0.008 ± 0.002 | 0.058 ± 0.009 | 0.011 ± 0.001 | 0.058 ± 0.009 | 0.754 ± 0.0 |
| SVM Platt | 0.008 ± 0.002 | 0.079 ± 0.002 | 0.012 ± 0.001 | 0.079 ± 0.001 | 0.754 ± 0.0 |
| SVM Temp | 0.266 ± 0.001 | 0.324 ± 0.005 | 0.253 ± 0.001 | 0.295 ± 0.003 | 0.647 ± 0.003 |
| SVM Isotonic | **0.002 ± 0.001** | 0.115 ± 0.001 | **0.002 ± 0.001** | 0.115 ± 0.001 | 0.754 ± 0.0 |
| LogisticRegression ERM | 0.016 ± 0.001 | 0.171 ± 0.002 | 0.016 ± 0.001 | 0.168 ± 0.002 | 0.827 ± 0.001 |
| LogisticRegression HKRR | 0.007 ± 0.002 | **0.028 ± 0.005** | 0.007 ± 0.002 | **0.026 ± 0.005** | **0.862 ± 0.0** |
| LogisticRegression HJZ | 0.006 ± 0.002 | 0.039 ± 0.007 | 0.011 ± 0.002 | 0.037 ± 0.006 | 0.857 ± 0.003 |
| LogisticRegression Platt | 0.008 ± 0.002 | 0.079 ± 0.002 | 0.012 ± 0.001 | 0.079 ± 0.001 | 0.754 ± 0.0 |
| LogisticRegression Temp | 0.062 ± 0.006 | 0.17 ± 0.039 | 0.058 ± 0.005 | 0.166 ± 0.042 | 0.833 ± 0.011 |
| LogisticRegression Isotonic | **0.002 ± 0.001** | 0.115 ± 0.001 | **0.002 ± 0.001** | 0.115 ± 0.001 | 0.754 ± 0.0 |
| DecisionTree ERM | 0.019 ± 0.001 | 0.039 ± 0.005 | 0.018 ± 0.002 | 0.038 ± 0.003 | 0.863 ± 0.001 |
| DecisionTree HKRR | **0.005 ± 0.001** | **0.033 ± 0.008** | **0.005 ± 0.001** | 0.032 ± 0.007 | 0.856 ± 0.001 |
| DecisionTree HJZ | 0.016 ± 0.003 | 0.042 ± 0.004 | 0.015 ± 0.002 | 0.036 ± 0.003 | 0.862 ± 0.0 |
| DecisionTree Platt | 0.02 ± 0.002 | 0.051 ± 0.007 | 0.018 ± 0.002 | 0.046 ± 0.007 | 0.861 ± 0.001 |
| DecisionTree Temp | 0.06 ± 0.002 | 0.08 ± 0.002 | 0.052 ± 0.001 | 0.069 ± 0.003 | 0.863 ± 0.002 |
| DecisionTree Isotonic | 0.006 ± 0.001 | 0.034 ± 0.004 | 0.009 ± 0.002 | **0.032 ± 0.003** | **0.863 ± 0.001** |
| NaiveBayes ERM | 0.134 ± 0.001 | 0.416 ± 0.006 | 0.126 ± 0.0 | 0.324 ± 0.003 | 0.808 ± 0.001 |
| NaiveBayes HKRR | 0.006 ± 0.001 | **0.032 ± 0.005** | 0.006 ± 0.001 | **0.03 ± 0.003** | **0.848 ± 0.001** |
| NaiveBayes HJZ | 0.008 ± 0.002 | 0.058 ± 0.009 | 0.011 ± 0.001 | 0.058 ± 0.009 | 0.754 ± 0.0 |
| NaiveBayes Platt | 0.008 ± 0.002 | 0.079 ± 0.002 | 0.012 ± 0.001 | 0.079 ± 0.001 | 0.754 ± 0.0 |
| NaiveBayes Temp | 0.185 ± 0.002 | 0.271 ± 0.004 | 0.184 ± 0.002 | 0.257 ± 0.003 | 0.809 ± 0.0 |
| NaiveBayes Isotonic | **0.002 ± 0.001** | 0.115 ± 0.001 | **0.002 ± 0.001** | 0.115 ± 0.001 | 0.754 ± 0.0 |

Figure 55: HMDA. Alternate groups.

| Model | ECE ↓ | Max ECE ↓ | smECE ↓ | Max smECE ↓ | Acc ↑ |
|---|---|---|---|---|---|
| MLP ERM | **0.017 ± 0.005** | 0.3 ± 0.057 | 0.021 ± 0.004 | 0.204 ± 0.03 | **0.866 ± 0.002** |
| MLP HKRR | 0.019 ± 0.002 | 0.311 ± 0.08 | **0.018 ± 0.002** | 0.217 ± 0.043 | 0.84 ± 0.002 |
| MLP HJZ | 0.025 ± 0.006 | **0.28 ± 0.049** | 0.024 ± 0.003 | 0.208 ± 0.031 | 0.864 ± 0.003 |
| MLP Platt | 0.02 ± 0.004 | 0.314 ± 0.048 | 0.023 ± 0.002 | 0.239 ± 0.039 | 0.863 ± 0.002 |
| MLP Temp | 0.063 ± 0.037 | 0.282 ± 0.061 | 0.058 ± 0.032 | **0.195 ± 0.038** | 0.864 ± 0.003 |
| MLP Isotonic | 0.025 ± 0.004 | 0.29 ± 0.047 | 0.025 ± 0.004 | 0.234 ± 0.027 | 0.864 ± 0.003 |
| RandomForest ERM | 0.019 ± 0.001 | 0.297 ± 0.038 | 0.021 ± 0.001 | 0.228 ± 0.018 | **0.863 ± 0.003** |
| RandomForest HKRR | 0.019 ± 0.002 | 0.311 ± 0.08 | 0.018 ± 0.002 | 0.217 ± 0.043 | 0.84 ± 0.002 |
| RandomForest HJZ | 0.016 ± 0.002 | **0.239 ± 0.036** | 0.019 ± 0.001 | **0.211 ± 0.015** | 0.861 ± 0.002 |
| RandomForest Platt | 0.019 ± 0.005 | 0.267 ± 0.03 | 0.021 ± 0.001 | 0.229 ± 0.02 | 0.86 ± 0.003 |
| RandomForest Temp | 0.041 ± 0.004 | 0.283 ± 0.057 | 0.039 ± 0.004 | 0.236 ± 0.014 | 0.861 ± 0.002 |
| RandomForest Isotonic | **0.015 ± 0.002** | 0.248 ± 0.019 | **0.017 ± 0.001** | 0.235 ± 0.012 | 0.862 ± 0.002 |
| SVM ERM | 0.143 ± 0.002 | 0.565 ± 0.0 | 0.072 ± 0.001 | 0.265 ± 0.0 | 0.857 ± 0.002 |
| SVM HKRR | **0.019 ± 0.002** | **0.311 ± 0.08** | **0.018 ± 0.002** | **0.217 ± 0.043** | 0.84 ± 0.002 |
| SVM HJZ | 0.027 ± 0.005 | 0.311 ± 0.037 | 0.026 ± 0.002 | 0.284 ± 0.027 | 0.826 ± 0.0 |
| SVM Platt | 0.14 ± 0.001 | 0.47 ± 0.051 | 0.07 ± 0.001 | 0.229 ± 0.02 | **0.86 ± 0.001** |
| SVM Temp | 0.117 ± 0.011 | 0.323 ± 0.022 | 0.116 ± 0.011 | 0.277 ± 0.018 | 0.847 ± 0.01 |
| SVM Isotonic | 0.048 ± 0.023 | 0.317 ± 0.076 | 0.048 ± 0.023 | 0.275 ± 0.052 | 0.847 ± 0.017 |
| LogisticRegression ERM | 0.022 ± 0.002 | 0.26 ± 0.02 | 0.022 ± 0.001 | 0.184 ± 0.009 | **0.866 ± 0.002** |
| LogisticRegression HKRR | 0.019 ± 0.002 | 0.311 ± 0.08 | **0.018 ± 0.002** | 0.217 ± 0.043 | 0.84 ± 0.002 |
| LogisticRegression HJZ | 0.017 ± 0.003 | 0.256 ± 0.064 | 0.02 ± 0.001 | 0.177 ± 0.028 | 0.863 ± 0.003 |
| LogisticRegression Platt | 0.02 ± 0.001 | 0.254 ± 0.032 | 0.022 ± 0.0 | 0.176 ± 0.031 | 0.862 ± 0.004 |
| LogisticRegression Temp | 0.102 ± 0.002 | **0.207 ± 0.045** | 0.094 ± 0.001 | **0.159 ± 0.025** | 0.862 ± 0.003 |
| LogisticRegression Isotonic | **0.016 ± 0.002** | 0.233 ± 0.03 | 0.019 ± 0.002 | 0.188 ± 0.037 | 0.861 ± 0.005 |
| DecisionTree ERM | 0.067 ± 0.004 | 0.328 ± 0.036 | 0.047 ± 0.004 | **0.189 ± 0.01** | **0.85 ± 0.006** |
| DecisionTree HKRR | **0.019 ± 0.002** | 0.311 ± 0.08 | **0.018 ± 0.002** | 0.217 ± 0.043 | 0.84 ± 0.002 |
| DecisionTree HJZ | 0.03 ± 0.007 | **0.286 ± 0.082** | 0.032 ± 0.007 | 0.225 ± 0.057 | 0.838 ± 0.006 |
| DecisionTree Platt | 0.106 ± 0.007 | 0.529 ± 0.064 | 0.059 ± 0.003 | 0.261 ± 0.03 | 0.836 ± 0.006 |
| DecisionTree Temp | 0.098 ± 0.003 | 0.3 ± 0.026 | 0.091 ± 0.002 | 0.232 ± 0.016 | 0.841 ± 0.005 |
| DecisionTree Isotonic | 0.055 ± 0.024 | 0.292 ± 0.039 | 0.042 ± 0.017 | 0.235 ± 0.046 | 0.836 ± 0.006 |
| NaiveBayes ERM | 0.277 ± 0.019 | 0.469 ± 0.031 | 0.164 ± 0.013 | 0.266 ± 0.025 | 0.714 ± 0.018 |
| NaiveBayes HKRR | 0.019 ± 0.002 | 0.311 ± 0.08 | **0.018 ± 0.002** | 0.217 ± 0.043 | **0.84 ± 0.002** |
| NaiveBayes HJZ | 0.027 ± 0.005 | 0.311 ± 0.037 | 0.026 ± 0.002 | 0.284 ± 0.027 | 0.826 ± 0.0 |
| NaiveBayes Platt | 0.268 ± 0.008 | 0.452 ± 0.012 | 0.164 ± 0.005 | 0.268 ± 0.01 | 0.719 ± 0.007 |
| NaiveBayes Temp | 0.298 ± 0.003 | 0.343 ± 0.006 | 0.276 ± 0.002 | 0.307 ± 0.004 | 0.719 ± 0.007 |
| NaiveBayes Isotonic | **0.017 ± 0.002** | **0.216 ± 0.06** | 0.019 ± 0.001 | **0.205 ± 0.055** | 0.829 ± 0.006 |

Figure 56: MEPS. Alternate groups.

# J Results on Language and Image Datasets

## J.1 Plots for All Multicalibration Algorithms

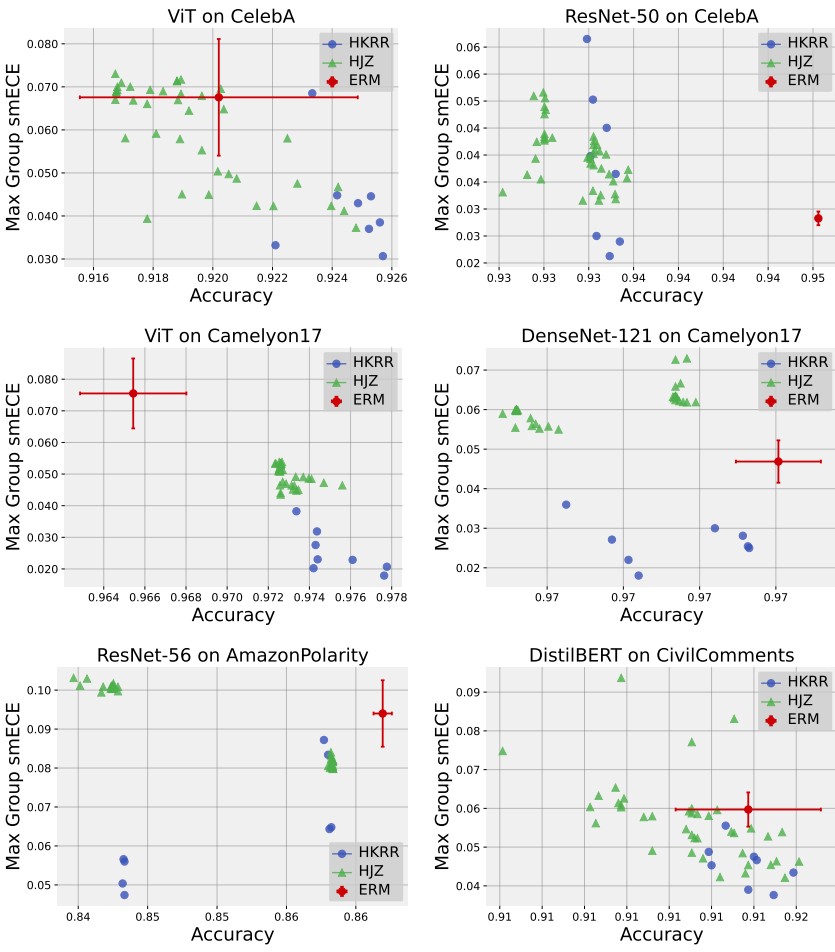

Figure 57: All multicalibration runs for image and language models. Note the small x-axis scale in some plots.

## J.2 Result Tables for Image and Language Data

| Model | ECE ↓ | Max ECE ↓ | smECE ↓ | Max smECE ↓ | Acc ↑ |
|---|---|---|---|---|---|
| ViT ERM | 0.021 ± 0.008 | 0.076 ± 0.011 | 0.022 ± 0.007 | 0.076 ± 0.011 | 0.965 ± 0.003 |
| ViT HKRR | 0.003 ± 0.0 | **0.018 ± 0.003** | 0.003 ± 0.0 | **0.018 ± 0.003** | **0.978 ± 0.0** |
| ViT HJZ | 0.006 ± 0.001 | 0.047 ± 0.001 | 0.007 ± 0.001 | 0.044 ± 0.003 | 0.973 ± 0.002 |
| ViT Platt | 0.014 ± 0.008 | 0.049 ± 0.014 | 0.016 ± 0.008 | 0.049 ± 0.014 | 0.973 ± 0.003 |
| ViT Temp | 0.025 ± 0.004 | 0.046 ± 0.01 | 0.019 ± 0.003 | 0.037 ± 0.008 | 0.973 ± 0.003 |
| ViT Isotonic | **0.001 ± 0.0** | 0.031 ± 0.007 | **0.002 ± 0.0** | 0.031 ± 0.007 | 0.977 ± 0.001 |
| DenseNet-121 ERM | 0.006 ± 0.003 | 0.047 ± 0.005 | 0.006 ± 0.002 | 0.047 ± 0.005 | **0.974 ± 0.001** |
| DenseNet-121 HKRR | 0.003 ± 0.002 | **0.018 ± 0.002** | **0.003 ± 0.002** | **0.018 ± 0.002** | 0.97 ± 0.003 |
| DenseNet-121 HJZ | 0.005 ± 0.001 | 0.056 ± 0.014 | 0.006 ± 0.001 | 0.055 ± 0.014 | 0.967 ± 0.003 |
| DenseNet-121 Platt | 0.006 ± 0.001 | 0.062 ± 0.015 | 0.007 ± 0.001 | 0.062 ± 0.015 | 0.971 ± 0.002 |
| DenseNet-121 Temp | 0.015 ± 0.002 | 0.052 ± 0.009 | 0.015 ± 0.002 | 0.05 ± 0.008 | 0.967 ± 0.003 |
| DenseNet-121 Isotonic | **0.002 ± 0.0** | 0.047 ± 0.006 | 0.003 ± 0.0 | 0.047 ± 0.006 | 0.972 ± 0.001 |

Figure 58: Camelyon17.

| Model | ECE ↓ | Max ECE ↓ | smECE ↓ | Max smECE ↓ | Acc ↑ |
|---|---|---|---|---|---|
| ViT ERM | 0.016 ± 0.006 | 0.069 ± 0.013 | 0.016 ± 0.006 | 0.068 ± 0.014 | 0.92 ± 0.005 |
| ViT HKRR | 0.008 ± 0.003 | **0.031 ± 0.003** | 0.008 ± 0.003 | **0.031 ± 0.003** | **0.926 ± 0.001** |
| ViT HJZ | 0.006 ± 0.001 | 0.038 ± 0.002 | 0.009 ± 0.0 | 0.037 ± 0.002 | 0.925 ± 0.001 |
| ViT Platt | 0.009 ± 0.002 | 0.047 ± 0.005 | 0.012 ± 0.002 | 0.047 ± 0.005 | 0.924 ± 0.001 |
| ViT Temp | 0.028 ± 0.008 | 0.072 ± 0.012 | 0.029 ± 0.009 | 0.07 ± 0.014 | 0.917 ± 0.006 |
| ViT Isotonic | **0.005 ± 0.001** | 0.057 ± 0.005 | **0.007 ± 0.001** | 0.057 ± 0.005 | 0.922 ± 0.001 |
| ResNet-50 ERM | 0.008 ± 0.001 | 0.028 ± 0.001 | 0.009 ± 0.001 | 0.028 ± 0.001 | **0.945 ± 0.0** |
| ResNet-50 HKRR | 0.006 ± 0.001 | **0.024 ± 0.004** | 0.006 ± 0.001 | **0.024 ± 0.004** | 0.934 ± 0.006 |
| ResNet-50 HJZ | 0.006 ± 0.001 | 0.032 ± 0.003 | 0.008 ± 0.0 | 0.033 ± 0.003 | 0.934 ± 0.007 |
| ResNet-50 Platt | 0.005 ± 0.001 | 0.037 ± 0.004 | 0.007 ± 0.0 | 0.037 ± 0.004 | 0.935 ± 0.006 |
| ResNet-50 Temp | 0.017 ± 0.006 | 0.046 ± 0.006 | 0.017 ± 0.006 | 0.045 ± 0.006 | 0.933 ± 0.007 |
| ResNet-50 Isotonic | **0.003 ± 0.001** | 0.051 ± 0.009 | **0.006 ± 0.001** | 0.051 ± 0.009 | 0.933 ± 0.007 |

Figure 59: CelebA.

| Model | ECE ↓ | Max ECE ↓ | smECE ↓ | Max smECE ↓ | Acc ↑ |
|---|---|---|---|---|---|
| DistilBERT ERM | 0.021 ± 0.001 | 0.065 ± 0.005 | 0.021 ± 0.001 | 0.06 ± 0.004 | 0.915 ± 0.001 |
| DistilBERT HKRR | 0.013 ± 0.0 | 0.047 ± 0.005 | 0.013 ± 0.0 | 0.043 ± 0.004 | 0.915 ± 0.001 |
| DistilBERT HJZ | 0.004 ± 0.001 | 0.043 ± 0.008 | 0.007 ± 0.001 | 0.043 ± 0.007 | 0.915 ± 0.001 |
| DistilBERT Platt | 0.004 ± 0.001 | 0.047 ± 0.008 | 0.007 ± 0.0 | 0.045 ± 0.007 | 0.915 ± 0.001 |
| DistilBERT Temp | 0.025 ± 0.005 | 0.044 ± 0.004 | 0.025 ± 0.005 | 0.044 ± 0.004 | 0.914 ± 0.001 |
| DistilBERT Isotonic | **0.002 ± 0.0** | **0.032 ± 0.006** | **0.005 ± 0.0** | **0.032 ± 0.006** | **0.916 ± 0.0** |

Figure 60: Civil Comments.

| Model | ECE ↓ | Max ECE ↓ | smECE ↓ | Max smECE ↓ | Acc ↑ |
|---|---|---|---|---|---|
| ResNet-56 ERM | 0.039 ± 0.013 | 0.094 ± 0.009 | 0.039 ± 0.013 | 0.094 ± 0.009 | **0.867 ± 0.001** |
| ResNet-56 HKRR | 0.015 ± 0.001 | **0.059 ± 0.01** | 0.015 ± 0.001 | **0.047 ± 0.005** | 0.848 ± 0.004 |
| ResNet-56 HJZ | 0.013 ± 0.005 | 0.081 ± 0.012 | 0.014 ± 0.005 | 0.081 ± 0.012 | 0.863 ± 0.002 |
| ResNet-56 Platt | 0.009 ± 0.003 | 0.082 ± 0.01 | 0.01 ± 0.002 | 0.082 ± 0.01 | 0.863 ± 0.002 |
| ResNet-56 Temp | 0.024 ± 0.01 | 0.07 ± 0.003 | 0.024 ± 0.01 | 0.069 ± 0.003 | 0.863 ± 0.002 |
| ResNet-56 Isotonic | **0.005 ± 0.001** | 0.079 ± 0.009 | **0.007 ± 0.0** | 0.078 ± 0.008 | 0.863 ± 0.002 |

Figure 61: Amazon Polarity.

