# OpenReview forum: "When is Multicalibration Post-Processing Necessary?"
_NeurIPS.cc/2024/Conference — NeurIPS 2024 poster_

### Official Review · Reviewer_rHYn · 2024-07-02

**Soundness:** 2
**Presentation:** 4
**Contribution:** 3
**Rating:** 6
**Confidence:** 3

**Summary:**

The paper investigates the necessity and effectiveness of multicalibration post-processing across data of different modalities and machine learning models. It finds that models which are calibrated out of the box tend to be multicalibrated without further post-processing, while multicalibration can help inherently uncalibrated models. Traditional calibration methods, such as Platt scaling, sometimes achieve multicalibration implicitly. The study, conducted on a wide range of datasets (tabular, image, language) and models (from decision trees to large language models), reveals that empirical risk minimization (ERM) often achieves multicalibration. It also notes that multicalibration post-processing is sensitive to hyperparameters and works best with large datasets. Additionally, traditional calibration methods can sometimes match the performance of multicalibration algorithms.

**Strengths:**

Writing is excellent and the paper is well organized, easy to read, and understandable without an effort. The message is clear and there are actionable recommendations. The topic is timely and the experiments done by the authors are fairly comprehensive; lots of models for tabular data, fewer for language and vision. This is one of the largest set of experiments I've seen in the literature. Overall, the work has the potential to have a meaningful impact on the community.

**Weaknesses:**

Not all the conclusions from the authors are well supported by their empirical findings. More details below.

Regarding observation 3:
* The association between the accuracy and the calibration fraction is somewhat clear for HMDA but it’s definitely unclear of the other datasets (Appendix F.3). Accuracy seems to even increase as the calibration fraction increases on certain datasets. Thus, the authors conclusion in observation 3 is only based on HMDA. I suggest dropping it or providing more support for that. The current experimental results do not support it.
* Max smECE of ERM is about twice as large as the max smECE after calibration in case of Naive Bayes, SVM, and decision trees. Then how do the authors justify their observation?
* Along the same lines, observation 7 suggests that there is a substantial difference between the results for language/vision data and tabular data because in the latter we do not see such gains. As mentioned right above, Figure 2 shows such gains. It just depends on which algorithms we look at.

Other comments:
* Observation 5 about the robustness of HJZ to hyperparameter choices does not hold for language and vision data? How do the authors explain this finding?

Minor details:
* What are the error bars in Figure 3? Standard deviations?
* Do the \pm values in Figure 3 represent standard error or deviation?

**Questions:**

I hope that the authors can solve my concerns noted above.

**Limitations:**

Limitations are discussed.

---

> ### Author Rebuttal · Authors · 2024-08-06
>
> We address the weaknesses and questions below.
>
> >**W1**: Regarding observation 3. The association between the accuracy and the calibration fraction is somewhat clear for HMDA but it’s definitely unclear of the other datasets (Appendix F.3). Accuracy seems to even increase as the calibration fraction increases on certain datasets. Thus, the authors conclusion in observation 3 is only based on HMDA. I suggest dropping it or providing more support for that. The current experimental results do not support it.
>
> Note that in Observation 3, we say “a practitioner utilizing multicalibration post-processing **may** have to trade off worst-group calibration error and total accuracy.” We do not claim that this tradeoff exists in all cases. This is an important note to practitioners, in particular because they may not have been previously aware that they may need to make such a tradeoff. For example, we would not want to make the claim that “a practitioner utilizing multicalibration post-processing may NOT have to trade off worst-group calibration error and total accuracy.”, even though such a claim is supported in some cases by the data.
>
> The tradeoff we observe here tends to occur for larger calibration fractions. We do often observe accuracy improve as calibration fraction moves from 0 to 0.4, but that accuracy then deteriorates as calibration fraction goes from .4 to 1.0. See the following examples in Appendix F.3: MLP on Credit Default (p. 33), MLP on MEPS (p. 33), MLP on HMDA (p. 33), Decision Tree on Credit Default (p. 34), Decision Tree on Bank Marketing (p. 34), Random Forest on Bank Marketing (p. 35), SVM on MEPS (p. 37), Naive Bayes on Credit Default (p. 38). We thank the reviewer for raising this point, however, and we will make all this discussion more explicit in the next version.
>
> >**W2**: Regarding observation 3. Max smECE of ERM is about twice as large as the max smECE after calibration in case of Naive Bayes, SVM, and decision trees. Then how do the authors justify their observation?
>
> We assume that the reviewer is referring to “after [multi]calibration” for NB, SVM, and DTs. Indeed, we agree with the reviewer that multicalibration can be helpful; we mention this in Observation 2: “HKRR or HJZ post-processing can help un-calibrated models like SVMs or Naive Bayes”, where we also provide empirical justification. In Observation 1, we point out that models which are multicalibrated out of the box also tend to be multicalibrated (on tabular data), but the converse also has merit: mis-calibrated models are also likely mis-multicalibrated. We observe that Naive Bayes, SVMs, and decision trees tend to be miscalibrated out of the box (see Fig. 2, Appendix F.2), and that multicalibration post-processing can help them. We will improve clarity and expand this discussion in the next version.
>
> >**W3**: Regarding observation 3. Along the same lines, observation 7 suggests that there is a substantial difference between the results for language/vision data and tabular data because in the latter we do not see such gains. As mentioned right above, Figure 2 shows such gains. It just depends on which algorithms we look at.
>
> We apologize for the confusion, and will work to improve clarity. As you observe, the NNs trained on vision/language data really do benefit from multicalibration post-processing. Some models trained on tabular data also do benefit from multicalibration post-processsing. Note, however, that the MLPs trained on tabular data do not tend to benefit from multicalibration post-processing in a statistically meaningful way. In any case where we draw a direct comparison between vision/language models and tabular models, we meant to compare only the NNs trained on these different types of data.
>
> More broadly, we mean to convey that multicalibration outcomes are both model and data dependent. We will improve the clarity in the next draft and remove this confusion for future readers. Thank you for pointing this out!
>
> >**W4**: Other comments. Observation 5 about the robustness of HJZ to hyperparameter choices does not hold for language and vision data? How do the authors explain this finding?
>
> We kindly refer to our global response for this discussion.
>
> >**W5**: Minor details. What are the error bars in Figure 3? Standard deviations? Do the \pm values in Figure 3 represent standard error or deviation?
>
> The error bars and $\pm$ values throughout our paper always represent standard deviation. We will make this explicit in the next version. We thank the reviewer for the helpful comments and feedback to our work. We will clarify the mentioned discussion in the next version.
>
> We thank the reviewer for their time and thoughtful feedback. If the pressing concerns have been sufficiently addressed, we kindly ask the reviewer to consider adjusting their score by taking the rebuttal into account. Thank you!

---

> > ### Comment · Reviewer_rHYn · 2024-08-09
> >
> > Thanks for the reply. I still believe that the paper, in its current version, presents several takeaways about algorithms/procedures that "can"/"may" or "cannot"/"may not" be helpful. However, in most cases, it's unclear why and when these statements hold true. While I acknowledge that the authors have conducted extensive experiments, more are needed, and I am concerned that practitioners may misinterpret these takeaways. Therefore, I suggest the authors carefully reconsider how each statement is phrased.
> >
> > I also agree with the other reviewers' comments regarding the lack of discussion on the specific nature of the groups, which were chosen arbitrarily. Different groups might lead to different conclusions about multicalibration. Although I don't disagree with the authors' response on this topic, I believe more should be said, and other groups should be explored on the same datasets to check if the results are consistent. This further adds to my concerns about how readers might interpret the paper's takeaways.
> >
> > Regarding W1: I believe the finding is not significant and could mislead practitioners, who are likely to skip the Appendix and focus only on the main body of the paper—or worse, just the takeaway without noticing the "may." As I mentioned in my initial rebuttal, this takeaway seems to overfit a specific dataset. If the trade-off only exists for larger calibration fractions, as highlighted in the examples, this should be clearly discussed in the takeaway. If the takeaway remains (and I don't think it should), the authors should emphasize the "may" in bold and explicitly state that the findings are supported by just one dataset or larger calibration fractions. This concern applies to all other questions/answers as well. The statements need to be crystal clear.
> >
> > Regarding W5: The authors likely mean standard error, not standard deviation.

---

> > > ### Author Response · Authors · 2024-08-12
> > >
> > > We sincerely thank the reviewer for their engagement and for acknowledging that we have conducted extensive experiments. You requested additional experiments with “more complex” groups, which we have been running since your last response, and hope to have uploaded by the end of today.
> > >
> > > You previously had some input for the phrasing of our Observations. We will modify them to make the takeaways clearer. In particular, you mentioned Observations 3, 5, and 7. Here are the old and newly proposed versions:
> > >
> > > >Observation 3: A practitioner utilizing multicalibration post-processing may have to trade off worst group calibration error and overall accuracy.
> > >
> > > New: A practitioner utilizing multicalibration post-processing could potentially be faced with a trade-off between worst group calibration error and overall accuracy. This is most salient in high calibration fraction regimes (40-80%).
> > >
> > > >Observation 5: There is no clearly dominant algorithm between HKRR and HJZ on tabular data but HJZ is more robust to choice of hyperparameters.
> > >
> > > New: When considering statistical significance, there is no clearly dominant algorithm between HKRR and HJZ on tabular data. However, HJZ is more robust to our hyperparameter sweep. This may allow practitioners to find good solutions faster than using HKRR when, for example, post-processing Naive Bayes or Decision Tree predictors.
> > >
> > > >Observation 7: Multicalibration post-processing can improve worst group smECE relative to an ERM baseline by 50% or more.
> > >
> > > New: On language and vision data, multicalibration post-processing can improve worst group calibration error relative to neural network ERM baselines by 50% or more. This stands in contrast to multicalibration post-processing for neural network ERM on tabular data, where we found no statistically significant improvements.

---

> > > > ### Author Response · Authors · 2024-08-12
> > > >
> > > > In addition, we will also add the following explicit discussion section to discuss groups. Note that although portions of this are taken from previous responses, there are new discussions sprinkled throughout. We would further like to highlight that the way we construct groups is standard, used by large production systems such as LinkedIn [4], in the measurement of bias in NLP systems [6], and in the auditing of deployed ML systems [2, 3].
> > > >
> > > > **Group Design and Limitations of Findings**
> > > >
> > > > Multicalibration requires the selection of “groups” or subsets of the population of interest. In most practical applications, these subgroups are given by features of the input, or simple conjunctions of features. This way of constructing groups is standard, used by large production systems such as LinkedIn [4], in the measurement of bias in ML [7, 8] and NLP systems [1, 6], and in the auditing of deployed ML systems like Facebook and LinkedIn [2, 3]. Furthermore, this approach still takes advantage of the group intersectionality notions (Ovalle et al.) that multigroup fairness is known for (Hébert-Johnson et al.).
> > > >
> > > > For our tested tabular datasets, we determined groups by such “sensitive” attributes—individual characteristics against which practitioners would not want to discriminate. In many cases, such attributes naturally include race, gender, and age, and vary with available information. For example, on ACS Income, we include groups such as “Multiracial”, “American Indian”, and “Seniors”. We also include some groups which are conjunctions of two attributes, for example “Black Women” or “White Women”. On Credit Default, for example, we include groups such as “Education = High School, Age > 40”, or “Married, Age > 60”. Similar groups are used throughout all the tabular data, as well as CelebA.
> > > >
> > > > On datasets where samples are not in correspondence with individuals—Camelyon17, Amazon Polarity, and Civil Comments—we define groups based on available information that can be viewed as “sensitive” with respect to the underlying task. In other words, we define groups such that an individual or institution using a predictor which is miscalibrated on this group may be seen as discriminating against the group. Ideally, a social media service will not be underconfident when predicting the toxicity of posts mentioning a minority identity group; such predictions may allow hate speech to remain on the platform. Thinking along this vein, we include “Muslim”, “Black”, and various other phrases defining protected groups in the Civil Comments dataset. Similarly, one would want a shopping platform to promote product listings fairly: a predictor which outputs mis-multicalibrated product ratings may boost listings for technology products proportionally to their true ratings but unfairly ignore the positive reviews of book listings. Thus, some groups from the Amazon Polarity dataset we used are defined as reviews containing the phrases “book”, “food”, “exercise”, or “tech”.
> > > >
> > > > There are (at least) two important properties of groups: group size and group “complexity”. The minimum group size $\gamma \in [0,1]$ is a parameter which has implications for the overall sample complexity of multicalibration (in particular, it introduces a $1/\gamma$ factor into known sample complexity upper / lower bounds. $\gamma$ is the size --- as a fraction of the dataset — of the smallest group in the collection). For this reason, we restricted ourselves to groups which were >0.5% of the entire dataset: without enough samples from a particular group, known multicalibration algorithms are prone to overfitting the training set and not providing any generalization performance. The increasing complexity of intersectional groups is a known issue, even at the scale of companies like LinkedIn [4]. Note that we consider groups spanning 0.5% all the way to 70-80% of the data. We deem this variety (details in appendix C.4) sufficient to capture the varying sizes. Group “size” can also be thought of as somewhat correlated with group complexity: one can imagine that some (but not all) sufficiently small groups are defined by more complex (boolean) functions of the features.
> > > >
> > > > (... continued in next response ...)

---

> > > > > ### Author Response · Authors · 2024-08-12
> > > > >
> > > > > (...continued from previous response...)
> > > > >
> > > > > More broadly, group “complexity” is more difficult to capture. For imperfect predictors, we can (nearly) always construct groups for which the predictor is not multicalibrated against. These groups may be as complex as the underlying predictor (since they can, for example, capture where the predictor misclassified test points). However, it is not clear whether this is a meaningful set of groups to multicalibrate against. To avoid such discussion, we intentionally determine groups by available features which we hypothesize practitioners may deem important or “sensitive” to the underlying prediction task. This is standard in the literature (see, e.g., [1,2,3,5]).
> > > > >
> > > > > As a consequence of these group selection criteria, our results only apply to these sensitive groups about which a practitioner would be concerned in practice. Theoretical analyses consider multicalibration on worst-case groups from an infinite set; in practice, this perspective is overly pessimistic. Observations of poor multicalibration with respect to our “simple” groups are not fruitless either; conditioned on such observations, multicalibration with respect to the more complex groups seems unlikely. If sample complexity is already an issue with simpler  groups, it will only become more of a problem as group complexity increases (and group size shrinks even further).
> > > > >
> > > > > It is entirely possible that defining more complex, but perhaps less practically motivated, group collections — e.g., the collection of all halfspaces — would yield different results than ours, and this remains an interesting direction for future work. We do not believe, however, that such possibilities detract from our overall message meant for practitioners using finite collections of groups defined from potentially sensitive features.
> > > > >
> > > > > [1]: Baldini et al. Your fairness may vary: Pretrained language model fairness in toxic text classification.
> > > > > [2]: Ali et  al. Discrimination through Optimization: How Facebook’s Ad Delivery Can Lead to Biased Outcomes.
> > > > > [3]: Imana et al. Auditing for Racial Discrimination in the Delivery of Education Ads.
> > > > > [4]: Quiñonero-Candela et al. Disentangling and Operationalizing AI Fairness at LinkedIn.
> > > > > [5]: Czarnowska et al. Quantifying Social Biases in NLP: A Generalization and Empirical Comparison of Extrinsic Fairness Metrics.
> > > > > [6]: Li et al. A Survey on Fairness in Large Language Models.
> > > > > [7]: Tifrea et al. FRAPPE: A Group Fairness Framework for Post-Processing Everything
> > > > > [8]: Atwood et al. Inducing Group Fairness in LLM-Based Decisions
> > > > >
> > > > > >W5: The authors likely mean standard error, not standard deviation.
> > > > >
> > > > > We explicitly compute standard deviation in line 191-201 of our attached supplemental code file scripts/results.py using the empirical standard deviation formula:
> > > > >
> > > > >                 for m in metric_suffixes:
> > > > >                     # mean
> > > > >                     s = sum([run[m] for run in metrics[cf][alg_type][param_idx]])
> > > > >                     n = len(metrics[cf][alg_type][param_idx])
> > > > >                     mean[m] = s / n
> > > > >
> > > > >                     # standard deviation
> > > > >                     sq_diff = sum([(run[m] - mean[m])**2 for run in metrics[cf][alg_type][param_idx]])
> > > > >                     sd[m] = (sq_diff / n) ** 0.5
> > > > >
> > > > > Further, we plot using matplotlib.axes.Axes.errorbar, using the (explicitly computed) standard deviation as the yerr parameter. To generate Figure 3, for example, please see the function `increasing_CF_helps_mcb_algs’ in generate_figures.py line 37. Here, we call the code from results.py to get the means and standard deviations, then use matplotlib.axes.Axes.errorbar to plot the error bars in lines 96 and 108.
> > > > >
> > > > > We really appreciate the reviewer taking time and engaging with our discussion, and believe that the paper will greatly benefit from their input. We can certainly incorporate any additional feedback if requested!

---

> > > > > > ### Comment · Reviewer_rHYn · 2024-08-12
> > > > > >
> > > > > > I appreciate the additional work the authors have done. The rephrasing and experiments on groups have resolved my concerns. I will raise my score.

---

> > > > > > > ### Author Response · Authors · 2024-08-12
> > > > > > >
> > > > > > > We thank the reviewer for their continued engagement and input in our work!

---

### Official Review · Reviewer_nkHR · 2024-07-13

**Soundness:** 2
**Presentation:** 2
**Contribution:** 2
**Rating:** 5
**Confidence:** 4

**Summary:**

The authors provide a large and broad set of evaluations of how useful it is to supplement empirical risk minimization procedures with multicalibration (and/or calibration) post-processing. Their experiments span a wide variety of settings and datasets, broadly falling into the tabular data, image data, and language data category. For all these categories of tasks, they evaluate whether or not one/both/neither of two existing multicalibration post-processing methods (referred to as HKRR and HJZ) are useful to apply to an ERM-trained predictor in order to ensure the resulting predictor is calibrated over a pre-specified family of subgroups in the data; and they do the same with respect to post-processing by applying simpler, marginal calibration methods such as isotonic regression.

They distill the findings into many practical insights that should be useful to future practitioners wishing to apply multicalibration in their settings. In broad strokes, it is found that in many cases, especially for tabular data, full multicalibration postprocessing may not make as much practical sense, as pure ERM (possibly with marginal calibration) may already get you close enough to calibration; but at the same time, in image/language processing settings, the usefulness of multicalibration post processing generally increases and may become worth it. Along with these general insights, the authors provide many other guidelines and observations on aspects such as hyperparameter tuning for multicalibration, the comparison in performance between HKRR and HJZ, and other empirical tradeoffs.

**Strengths:**

In general, the multicalibration literature has suffered from a lack of well-concerted assessments of the empirical validity of its undisputably significant theoretical achievements (both when it comes to its “multigroup fairness” motivation and in other applications). Such assessments are even more needed given that known theoretical bounds on multicalibration’s sample complexity, as well as the overfitting effects observed in limited empirical assessments, point to potentially pessimistic empirical performance (especially compared to non-multigroup, classic calibration algorithms like Platt or temperature scaling or isotonic regression). So one important merit of this paper is that it can serve as a good starting point for further empirical evaluation of multicalibration methods.

The setup itself, which is to examine how to partition available data into training and calibration sets (+ validation set as necessary to tune the multicalibration algorithms’ hyperparameters), and to ask whether or not the calibration data would be better used for the training / ERM step, is a clean and reasonable question to ask; especially so given the many hints found in recent prior work suggesting that ERM procedures may already encapsulate some (multi)calibration guarantees even without explicitly enforcing them.

In all, the authors have indeed managed to execute, to a fair degree (and modulo some concerns below), on their promise of a first comprehensive evaluation of multicalibration post-processing. As listed in the summary section above, they have also provided various useful rules of thumb along the way, which should benefit practitioners and also serve as conversation starters, and in several cases as cautionary notes, when it comes to applying multicalibration in practice.

**Weaknesses:**

The prior work results about ERM being sufficient for obtaining various guarantees are mentioned, but not discussed in enough depth to provide the full context for the question being asked in the present manuscript. For instance, an important thing that [Blasiok et al, 2023]’s note in their intro, before providing their result on NNs giving multicalibration, is that one might imagine that ERM-based solutions might a-priori need to rely not just on a larger-sized NN, but also on a certain larger family of subsets that encapsulate/imply the subgroups that the practitioner wants to calibrate their data on — and their results show that in fact, aside from increasing the size of the neural net, it is (in that setting) not necessary to design any special enlarged family of subsets. So, put simply, the current manuscript needs to note much more explicitly that designing an enlarged group family for the ERM could a-priori be an important design choice for practitioners, but may not be as important (as enlarging the size of the model), as hinted on by prior work. Furthermore, in the same vein as, and aside from, [Kirichenko et al, 2023], please also cite/mention [Rosenfeld et al, 2023] — and for both these works, please discuss the overlap between robustness/multigroup fairness considerations.

Group families with respect to which to multicalibrate, especially for tabular datasets, are defined in various “intuitive” ways, such as demographic groups. In that way, consistent with observations from [Kirichenko et al] and [Rosenfeld et al], it appears that explicit multicalibration not being necessary may be due to the fact that the predictors already learn the requisite features that are both prediction-relevant and also significantly correlate with the chosen protected groups. The extent to which this is an important factor — and that multicalibration may indeed be necessary for more “unusual” groups (worst-case, one can almost always hand-craft groups that, at least on training data, show miscalibration) —is not studied in this manuscript. For instance, the worst-group calibration reported only ranges over chosen groups of interest, but of course other groups not considered may have much worse calibration guarantees. So I’m not convinced the results that say multicalibration may not be necessary are robust to defining groups that are somehow non-standard.

Another example of potential discrepancy in what is being studied here is the smECE vs ECE distinction — to my knowledge, the HKRR and HJZ algorithms were developed prior to the smECE paper, and can in their various instantiations explicitly target metrics like ECE or L2 or L_infty calibration, rather than smECE. Thus, at least to a less-familiar reader, this must at least be clarified e.g. in the form of a statement that conclusions based on smECE performance are of a purely empirical character.

There are other presentation issues from my perspective. Besides not enunciating group choices in the main part as just discussed, the algorithms that are actually used are not explained in any sort of pseudocode, nor are their objectives/claimed bounds ever stated. This is an important writing issue given that HKRR and HJZ are introduced in their respective papers using very different notation, with differing guarantees, and with finer distinctions, such as e.g. that HJZ can output both deterministic and non-deterministic recalibrated predictors.

**Questions:**

Please see the “Weaknesses” section above; some concerns I listed there (notably as far as writing/presentation is concerned) could be resolved within a revision.

---

> ### Author Rebuttal · Authors · 2024-08-06
>
> We agree that recent theoretical consequences of multicalibration have provided more than sufficient motivation for an empirical analysis. In light of this observation, we point out that another contribution of our work is the benchmarking repository containing all experimental code (submitted with the supplemental material). We believe this will lower several barriers to future work in this direction.
>
> We address the weaknesses and questions below.
>
> We agree that a more thorough treatment of [1] is needed; this will be included in the next version. The connection to our work is subtle. In [1], the authors consider performing ERM over some C’ (superset of C), so that the resulting model is multicalibrated with respect to C. In their case, C’ is a family of neural networks (NNs), and C is some family of smaller NNs. In many of our experiments, we indeed perform (approximate) ERM over some family of NNs, and one might expect this to result in multicalibration with respect to our finite collection of groups G, which are easily computable by some class of smaller NNs. We find this to largely be the case in our tabular experiments (Observation 1). This is, as you said, one unique aspect of neural networks vs other model families: for NNs, it is therefore possible to obtain multicalibration guarantees over arbitrarily-complex groups by simply taking "large enough" networks (without making design choices specific to the group structure). We will clarify these points in our revision.
>
> We also agree that a more faithful account of connections to robustness is in order, especially in light of recent theoretical work drawing some connections between these areas [2, 3]. We note that [2] considers corruptions in subgroups of the training data rather than general distribution shift. In general, we view multi-group robustness as a notion of robustness which, like multicalibration, aims to “respect” multiple groups simultaneously. Perhaps the closest connection between multigroup robustness and multicalibration is that they intuitively may share similar mechanisms (as the reviewer noted). That is, the mechanistic question in both cases is to understand which groups can be easily computed from the predictor's "features" – ie, which groups are easy for the predictor to distinguish. We will elaborate on this point (and the relations to the mentioned works) in our revision.
>
> Apart from this similarity, we think it is necessary to delineate practical multicalibration concerns from practical robustness concerns. As noted in Appendix C.2, it is common for empirical works in subgroup robustness to consider groups of a fixed label, and to consider worst group accuracy. These groups alone are not meaningful from the perspective of multicalibration, however, as multicalibration requires more refined probability estimates even when expected labels lie in (0,1). We will explain this relationship and cite [4] (with further discussion) in the next version of our paper.
>
> [1]: Błasiok et al. Loss minimization yields multicalibration for large neural networks.
> [2]: Hu et al. Multigroup Robustness
> [3]: Wu et al. Bridging multicalibration and out-of-distribution generalization beyond covariate shift.
> [4]: Rosenfeld et al. (Almost) provable distribution shift via Disagreement Discrepancy.
>
> >**W2**: Findings may not hold for "non-standard" group definitions.
>
> We give a detailed discussion of this in our **global reviewer response**, to which we kindly refer.
>
> Briefly, the fact that our results don't hold for non-standard groups is inherent to the problem formulation. It is impossible for a fixed predictor to be multicalibrated with respect to all groups (as you point out), and thus we must decide which groups to consider when studying multicalibration. Certain theory works chose to consider "all groups computable by a small circuit/DNN" (e.g. [Blasiok et al, 2023] and [Hébert-Johnson et al. 2017]). Since our focus is on the application side, we instead chose groups that are relevant to fairness applications, which are often "intuitive" groups as you said.
>
> At a high level, the prior theory works and our empirical work use group definitions at two ends of the spectrum (from "all circuits" to "intuitive groups"). We could hope to eventually understand exactly *which* groups our algorithms are multicalibrated on, but this goal does not yet appear in reach; our work is just the first step.
>
> We will be sure to clarify these limitations, and the importance of group definitions, in our revision.
>
> >**W3**: Discrepancy in HKRR / HJZ guarantees and calibration measurement.
>
> We agree that the distinction between the theoretical guarantees of HKRR / HJZ (via ECE / L-p calibration) and what we measure (smECE) is important. We report both ECE and smECE for this reason, but will do better to specify which of our takeaways / observations are particular to the metric of interest.
>
> >**W4**: Other issues: "Besides not enunciating group choices in the main part as just discussed, the algorithms that are actually used are not explained in any sort of pseudocode, nor are their objectives/claimed bounds ever stated"
>
> We intend to add two more discussions to our next version: (1) a clear high-level description of the two families of multicalibration algorithms and their guarantees (including what kind of calibration metric they guarantee performance on), as well as psuedo-code for the algorithms (most likely relegated to the appendix); and (2) a description of how and why we selected the groups for each dataset. We will also include a more explicit discussion about the limitations of our work, regarding the questions you posed in W2.
>
> We thank the reviewer for their time and thoughtful feedback. If the pressing concerns have been sufficiently addressed, we kindly ask the reviewer to consider adjusting their score by taking the rebuttal into account. Thank you!

---

> > ### Comment · Reviewer_nkHR · 2024-08-13
> > **Response to Authors**
> >
> > Thank you for your responses to me and to the other reviewers, as well as for the general response. There, you gave several more experiments that are good to see, and provided several detailed discussions of the various moving parts involved in evaluating the benefits of multicalibration on various kinds of data; I agree with most of your points there.
> >
> > One "fine-print" point that I would like to highlight is that when you say that you will clarify that your work focuses on feature-defined groups that would be useful to practitioners --- as opposed to various potentially more convoluted group families such as ones that might predict which region in the data the predictor might be miscalibrated on --- it is important to further stress that these two categories of group families are not entirely disconnected from each other. Indeed, for some less-fancy (and even some fancy) predictors that people may be using in practice, it could well be the case that one might be able to "eyeball" (or deduce on a small holdout set) some feature-defined groups that would also explicitly target the model's inaccurate predictions region --- e.g. one might suspect that the model mispredicts a lot on some unprotected but still natural demographic group, and include that group in the family. Such a scenario would fall squarely into the "practical" category of applications of multicalibration, and thus cannot be discounted.
> >
> > Therefore, I urge you to be careful in your formulation of what kinds of groups your paper considers --- given my above point, I would refrain from saying that your statements are made about the general class of intuitive feature-based groups that practitioners might use, and steer it more into the direction of saying that you consider group families that may seem inherently important to protect regardless of which predictor is used on the data. (Again, to reiterate, my point is: just like the groups you experiment with, groups that capture inefficiencies of the predictor don't have to be complex or unnatural; they may well be feature-defined and relatively natural, and practitioners may intentionally or unintentionally come across such groups.)
> >
> > Also, I would place extra emphasis in the revision on the fact that "practitioners" refers to something like "fairness practitioners"; indeed empirical research in other areas may lead to other groups being used, that this paper was not focusing on.
> >
> > Finally, regarding raising my score, as it stands, there are many key updates needed in terms of the phrasing of the paper's setting and of its takeaways and of its literature connections etc., and while I believe the authors are on the right track in terms of this, judging by the rebuttal discussions, but the result of incorporating so many of these updates will inherently be high-variance in terms of the ensuing readability, focus etc of the updated paper. In this way, I am feeling more positive about this paper now but I am still in the "borderline" region; so I hope this explains why I would prefer to keep my current score.

---

> > > ### Author Response · Authors · 2024-08-13
> > >
> > > Thank you again for these points and all of your thoughtful feedback. They will certainly help improve the clarity of discussion and limitations in our next draft!

---

### Official Review · Reviewer_M2Di · 2024-07-13

**Soundness:** 4
**Presentation:** 4
**Contribution:** 4
**Rating:** 8
**Confidence:** 4

**Summary:**

This work presents an empirical analysis of multi-calibration post-processing applied to a variety of models for binary classification, ranging from decision trees to transformers. They perform experiments with tabular, image, and text data on datasets of varying sizes. They compare model group calibration to overall calibration after a) only ERM b) calibration and c) multicalbration. They observe that, while multicalibration postprocessing does improve multicalibration, in many cases, simple calibration improves it similarly. Additionally, they find that for models that are already calibrated after ERM, they are often multicalibrated as well. They observe differences in metrics used to calculate multicalibration, varying with dataset size that have implications for the situations in which ECE and smECE are appropriate to use. Finally, they find that for large datasets, such as those for vision and language, HJZ shows bigger improvements over HKRR.

**Strengths:**

This is a very expansive and thorough empirical study that has lots of practical takeaways for the ML fairness community that could be very valuable. Particularly in the context of larger models, where post-processing can be very costly, knowing what methods are likely to work is very important. The paper's organization is clear, results are easy to understand, and relevant takeaways are clearly highlighted. The paper builds well on prior work while providing novel key insights about how different methods perform in different settings.

**Weaknesses:**

The paper could be more self-contained. While it's understandable for some results to be in the appendix given the number of experiments run in this paper, I worry that too much may have been pushed to the appendix (though this may be unavoidable).
Some details, such as how groups are chosen for language tasks, are also not clear when consulting the appendix, particularly for readers who are not familiar with how these groups are normally chosen. Details like this would help readers to better understand the contribution of the paper.

**Questions:**

1. What are the groups for text experiments?
2. Figure 2 claims that MLPs are among the models that do not benefit from post-processing, however the table appears to show better multicalibration after post-processing. Is this claim correct?

**Limitations:**

The limitations of the paper seemed well addressed to me.

---

> ### Author Rebuttal · Authors · 2024-08-06
>
> We thank the reviewer for the detailed comments and feedback. We will address the weaknesses and questions.
>
> >**W1**: The paper could be more self-contained. While it's understandable for some results to be in the appendix given the number of experiments run in this paper, I worry that too much may have been pushed to the appendix (though this may be unavoidable).
>
> We agree with this point, and will include more figures in the body of our next version, and experiment with compressing some of the data presented in the main paper. Under each figure, we will also include a direct reference to related appendices. We believe that we can further compress some of the experimental results and (especially) figures in order to include more in the main paper.
>
> >**W2**: Some details, such as how groups are chosen for language tasks, are also not clear when consulting the appendix, particularly for readers who are not familiar with how these groups are normally chosen. Details like this would help readers to better understand the contribution of the paper.
>
> We also agree with this point. We will add a clear description of our group selection criteria to the body, as well as a high-level description (and pseudocode in the appendix) of the multicalibration algorithms considered.
>
> We determined groups by “sensitive” attributes—individual characteristics against which practitioners would not want to discriminate. In many cases, such attributes naturally included race, gender, and age, and then varied with available information. This included all tabular datasets as well as the CelebA dataset (image). On the other datasets, samples are not necessarily in correspondence with individuals, (e.g. postings on an internet platform, or slides of tissue).
>
> On datasets where samples are not in correspondence with individuals—Camelyon17, Amazon Polarity, and Civil Comments—we define groups based on available information that can be viewed as “sensitive” with respect to the underlying task. In other words, we define groups such that an individual or institution using a predictor which is miscalibrated on this group may be seen as discriminating against the group. Ideally, a social media service will not be underconfident when predicting the toxicity of posts mentioning a minority identity group; such predictions may allow hate speech to remain on the platform. Similarly, we would want a shopping platform to promote product listings fairly: a predictor which outputs mis-multicalibrated product ratings may boost listings for technology products proportionally to their true ratings but unfairly ignore the positive reviews of book listings.
>
> Aside from these heuristics, we also required that groups composed at least a 0.005-fraction (.5%) of the underlying dataset. This imposed a degree of uniformity in our group selection across all datasets.
>
> >**Q1**: What are the groups for text experiments?
>
> The text datasets are Amazon Polarity and Civil Comments. Groups for these datasets can be seen in Appendix C.6 (page 19) but briefly can be described as the presence of certain relevant words or phrases. We will also improve the group explanations in the main text to avoid confusion.
>
> >**Q2**: Figure 2 claims that MLPs are among the models that do not benefit from post-processing, however the table appears to show better multicalibration after post-processing. Is this claim correct?
>
> In the table of Figure 2, in the first group of 4 rows, it can be seen that MLP HJZ obtains a max smECE of 0.076 ± 0.018, while MLP ERM obtains a max smECE of 0.086 ± 0.015. While we agree that the max smECE of HJZ is less than that of ERM here, this difference is not statistically meaningful: each of these means is within its standard deviation of the other.
>
> We give a more detailed discussion of this point in response to Q4 of reviewer Zyam, to which we kindly refer.

---

> > ### Comment · Reviewer_M2Di · 2024-08-13
> >
> > Apologies for the late reply. Thank you for these clarifying points! I'll be keeping my score, but believe that adding the additional experiments and improved group descriptions to the paper would greatly improve the clarity.

---

### Official Review · Reviewer_Zyam · 2024-07-16

**Soundness:** 2
**Presentation:** 3
**Contribution:** 3
**Rating:** 5
**Confidence:** 4

**Summary:**

The paper investigates the effectiveness of multicalibration post-processing across various datasets and machine learning models. The study finds that models which are inherently calibrated often exhibit multicalibration without post-processing, while uncalibrated models benefit from multicalibration techniques. Traditional heuristics calibration methods can sometimes achieve similar outcomes as multicalibration algorithms. The research provides empirical evidence and practical insights for applying multicalibration in real-world scenarios, emphasizing the sensitivity of multicalibration algorithms to hyperparameter choices and the varying necessity of multicalibration depending on the model and dataset.

**Strengths:**

1. The writing is clear: The introduction effectively establishes the need for this empirical study, and the algorithms and results are presented clearly and concisely.
2. This is the first comprehensive study to evaluate different multicalibration methods.
3. The experimental results are thoroughly discussed, and the observations are detailedly listed.

**Weaknesses:**

1. The paper does not introduce any new algorithms, datasets, or theoretical analysis, but instead focuses on evaluating existing algorithms.
2. The experimental results are primarily observations and lack a deeper discussion of the reasons behind the observed phenomena.
3. Including experiments for all datasets in the main paper would help readers better understand the common findings and differences.

**Questions:**

1. For lines 179-181, in what sense is HKRR sample inefficient? I thought multicalibration for binary problems was known to be sample efficient.
2. Could you provide more reasons why HKRR outperforms HJZ in language and vision datasets?
3. What is the rationale behind choosing a large fraction of data for calibration? From my understanding, it should be about 10% or even smaller when the overall dataset is small.
4. How do the results differ across different datasets for tabular data? I noticed that in the HMDA dataset, the post-processing algorithms perform strictly better than ERM. Do MC algorithms work better for large datasets? Could you add more experiments if this hypothesis makes sense?
5. Did you try to create more complicated groups for the tabular experiments? It is possible that the groups are simple enough that even ERM works well.
6. For Figure 3, could you explain why the max group smECE increases when the calibration fraction gets larger?

**Limitations:**

Yes, the authors adequately addressed the limitations.

---

> ### Author Rebuttal · Authors · 2024-08-06
>
> We thank the reviewer for the detailed comments and feedback. Please kindly refer to the global response for **Q2** and **Q5**. We address the other weaknesses and questions below.
>
> **W1**: We agree with the reviewer that we do not introduce any new algorithms. The expressed purpose of our work is to empirically study existing algorithms and attempt to determine their potential for practical use in a host of important (existing) regimes. There are already a number of theoretical works examining and proposing multicalibration (MCB) algorithms which we cite and discuss (Hébert-Johnson et al., Gopalan et al. 2022a/b, Bastani et al. 2022 etc.). We believe that determining when and where these algorithms are useful in practice is an extremely important next step for the research area.
>
> More broadly, we respectfully disagree that focusing on existing algorithms is a weakness of papers in machine learning. It is worth noting the existence of several important empirical studies whose main contributions were not novel methods or theoretical results. For example, the main contributions of the important works [1, 2, 3] are almost purely focused on examining existing algorithms from an empirical lens. Such evaluation research is especially important for algorithms involved in high-stakes decision making in the real world (including multicalibration e.g., Barda et al. 2021).
>
> [1]: Grinsztajn et al., Why do tree-based models still outperform deep learning on typical tabular data?
> [2]: Zhang et al., Understanding Deep Learning Requires Rethinking Generalization.
> [3]: Minderer et al., Revisiting the Calibration of Modern Neural Networks.
>
> **W2**: We agree that we do not provide much explanation for the reasons behind the observed phenomena. However, we envision our work as an important step to understand when and where to focus additional study in order to better understand the applicability of MCB algorithms. Before our work, it was not at all obvious when and where MCB may or may not be helpful to the standard machine learning practitioner. Importantly, we believe that we already provide numerous helpful insights and words of caution.
>
> **W3**: We agree that many experiments are relegated to the appendix. This was due to space constraints: our goal was to have a cohesive message from over 40K MCB runs in the main body of the paper. We hoped to distill practical take-aways and best-practices for practitioners wishing to apply and measure the effectiveness of MCB post-processing algorithms. If accepted, we plan to better compress experimental results from the appendix and include them in the main body, with in-line references to appendix links in the captions.
>
> **Q1**: HKRR / HJZ are efficient in that the number of samples required is polynomial in the parameters of the algorithm. However, the degree of this polynomial may be large. For example, sample complexity bounds of the HKRR algorithm are $O(1/(\alpha^4 \cdot \lambda^{1.5}))$. Unfortunately, this quickly grows to the order of millions of samples for moderately small choices of $\alpha$ (multicalibration error) and $\lambda$ (bucket width). We view our work as investigating whether we really need this large number of samples to empirically provide reasonable multicalibration levels in practice.
>
> **Q3**: We vary the amount of (hold-out) calibration data widely due to the sample complexity issues discussed in Q1. In particular, the required samples to guarantee theoretical convergence is very large, and a priori, it was not clear at all how much data should be given to MCB algorithms in practice. This is because MCB algorithms generally require many more samples than simple calibration (which often performs well with only 10% of the data). Indeed, we find that in some cases for tabular data, MCB performs best with 40-80% sized calibration sets. More generally, this is due to a trade-off between the effectiveness of MCB post-processing and the (latent) multicalibration properties of ERM. We will include this discussion in the revision.
>
> **Q4**: While results differ in minor ways between the different tabular datasets, we observe that models which are calibrated “out of the box” tend to not benefit from multicalibration post-processing in a statistically meaningful way (Observation 1). On four of the five tabular datasets, looking only at the max-smECE metric of MLP ERM, the standard deviations overlap with the opposing mean (MEPS, ACS Income), or (mean ± sd) values come within 0.003 of each other (Bank Marketing, Credit Default). As you point out, on HMDA (the remaining fifth dataset), max-smECE of MLP ERM is improved drastically by post-processing. On HMDA, however, MLP ERM is poorly calibrated to begin with: relative to the smECE of MLP HKRR (0.005 ± 0.001), or even the smECE of MLP Temp (0.022 ± 0.006), the smECE of MLP ERM (0.049 ± 0.006) is quite large. Therefore, observation 1 does not apply (as strongly) to the MLPs trained on HMDA. On the four other tabular datasets, MLP ERM achieves smECE much closer to that of the post-processed MLPs. Outside of MLPs, we find that miscalibrated models (e.g. Naive Bayes, SVMs) tend to benefit from multicalibration post-processing throughout our tabular experiments (Observation 2). We can provide specific examples if necessary (omitted here due to space).
>
> **Q6**: Note that an increase in calibration fraction implies less data is used to train the base predictor, and more data saved for post-processing. As you point out, max smECE increases as we increase the calibration fraction in Fig 3. This is likely since the base predictor has poorer generalization when trained on less data, and this performance is not rectified by multicalibration post-processing. Please also see **Q3**.
>
> We thank the reviewer for their time and thoughtful feedback. If the pressing concerns have been sufficiently addressed, we kindly ask the reviewer to consider adjusting their score by taking the rebuttal into account. Thank you!

---

> > ### Comment · Reviewer_Zyam · 2024-08-12
> >
> > Thank you for your detailed response!
> >
> > Regarding Q1: I believe it would be beneficial to make this point clear in the final version.
> >
> > Regarding Q3: Yes, I definitely think it would be useful to include this in the discussion.
> >
> > Regarding Q4: It's important to be careful when stating observations and to list results across all datasets in the main paper so that readers can easily verify them.
> >
> > I agree with reviewer rHYn’s point that "However, in most cases, it's unclear why and when these statements hold true," and "Therefore, I suggest the authors carefully reconsider how each statement is phrased." I'm glad to see the authors provided a detailed response to this.
> >
> > Overall, while some of the reasoning behind the observations remains unclear, given that this paper is the first comprehensive study to evaluate different multicalibration methods and includes a large number of experiments, and considering that the authors plan to include additional discussion based on feedback from the reviewers, I will raise my score to 5.

---

> > > ### Author Response · Authors · 2024-08-12
> > >
> > > Thank you for your response and engagement!
> > >
> > > To address your Q4, we will better compress the results from all datasets in the main body as discussed in M2Di weakness 1. We appreciate the feedback and believe it will result in improvements to the presentation!

---

### Author Rebuttal · Authors · 2024-08-07

**Global author response**

We thank all the reviewers for the thorough reviews and detailed comments. We respond to two common reviewer comments here.

1. Firstly, reviewers **Zyam Q5**, **nkHR W2**, and **M2Di W2** all had questions and comments about group selection. We have combined our discussion here. We also have some additional discussion in the response to **nkHR W2**.

There are (at least) two important properties of groups: group size and group “complexity”. We will certainly provide further discussion of both for all datasets in the next version of the paper.

The minimum group size is a parameter which has implications for the overall sample complexity of multicalibration (in particular, it introduces a $1/\gamma$ factor into known sample complexity upper / lower bounds. $\gamma \in [0,1]$ is the size --- as a fraction of the dataset --- of the smallest group in the collection). For this reason, we restricted ourselves to groups which were >0.5% of the entire dataset. Note that we consider groups spanning 0.5% all the way to 70-80% of the data. We deem this variety (details in appendix C.4) sufficient to capture the varying sizes. Group “size” can also be thought of as somewhat correlated with group complexity: one can imagine that some (but not all) sufficiently small groups are defined by more complex (boolean) functions of the features.

More broadly, group “complexity” is more difficult to capture. As reviewer **nkHR** points out, we can (nearly) always construct groups for which our predictor is not multicalibrated against. These groups may be as complex as the underlying predictor (since they can, for example, capture where the predictor misclassified test points). However, it is not clear whether this is a meaningful set of groups to multicalibrate against. To avoid such discussion, we intentionally determine groups by available features which we hypothesize practitioners may deem important or “sensitive” to the underlying prediction task. In our tabular data and CelebA, such attributes naturally include race, gender, and age, and further vary with the available features already in the data. On the other datasets, however, samples are not necessarily in correspondence with individuals, (e.g. postings on an internet platform, or slides of biological tissue). In these cases, we define groups such that an individual or institution using a predictor which is miscalibrated on this group may be seen as discriminating against the group.

As a consequence of these selection criteria, our results only apply to these sensitive groups about which a practitioner would be concerned in practice. We will make this limitation clearer in the next draft of the paper. It is entirely possible that defining more complex — but perhaps less practically motivated — groups would yield different results than ours, and this remains an interesting direction for future work. We do not believe, however, that such possibilities detract from our overall message. We will further emphasize in our next draft that our findings hold only with respect to how we have defined groups "simply" or "intuitively".

2. Secondly, reviewers **Zyam Q2** and **rHYn** have questions about 1) Why HKRR outperforms HJZ in language and vision datasets; and 2) Why HJZ is not robust to hyperparameter choices for language and vision data (potentially counter to Observation 5). These are both questions about dynamics of the multicalibration post-processing algorithms, and we address them with the following discussion.

The dynamics of both of these algorithms are complex, and an answer to these questions is difficult to support from our results alone. In particular, the performance of the algorithms depend on _at least_ the following parameters:

1. Distribution of initial predictions output by the models (i.e. input to post-processing algorithm).
2. Choice of hyperparameters for HKRR and HJZ.
3. “Complexity” or “expressiveness” of the groups.
4. Amount of samples.

In our work, we focus mainly on (2) and (4). We discuss (3) at length above, but teasing apart exactly how (1) and (3) contribute to the performance of the multicalibration algorithm is certainly an interesting avenue for future work. We believe that part of the reason for the superiority of HKRR in language/vision data may be explained within the lens of (2) and (4).

As stated in the paper (line 296), due to computational constraints and the added dimension of choosing how much data to save for calibration, we search a large — but not all-encompassing — collection of hyperparameters for each of the multicalibration algorithms tested. With regards to dataset size, three of the four vision/language datasets are noticeably larger than the tabular datasets (by at least 100k samples). It is possible that HKRR generally performs better on such dataset sizes, or that the optimal hyperparameters for HJZ change significantly in this larger sample regime.

Stability of HJZ is a more challenging question to answer, since it likely has to do with internal game dynamics in the learning algorithm. In particular, by choosing various online learning algorithms, HJZ implements a _family_ of multicalibration post-processing. We test all algorithms from this family with varying parameters. It is possible that the family of algorithms itself somehow has a shift in stability as we scale to a large data regime (4). However, as analyzing even a singular algorithm (e.g. HKRR) is challenging, we are not sure that speculating about the stability of a family of algorithms is currently possible, and hence, leave this to future work.

We thank all the reviewers for the helpful feedback, and we will certainly incorporate additional discussions (as indicated) in future versions of our work. Thank you!

---

> ### Author Response · Authors · 2024-08-12
> **Additional Experiments with Different and More "Complex" Groups**
>
> As requested, we ran some additional experiments for MLPs on the tabular datasets with different --- and in some cases, more complex --- groups. We present a few of the experiments here, and echo takeaways relative to the old experiments (with different group definitions). Unfortunately, we can no longer submit a PDF anywhere, so have included the results as markdown tables. Due to character limits, these tables may have to be split among multiple respones.
>
> Overall, the takeaway for MLPs largely remains the same: it is difficult to find instances where multicalibration helps in a statistically significant way over ERM or some form of simple calibration (Observations 1 and 2). Further, where multicalibration does help, this help sometimes comes at a cost to accuracy (Observation 3).
>
> For each dataset, we include the "initial" results which are in our submitted draft, and new results which we ran at the reviewers requests.
>
> ## ACS Income
> ### Group definitions: initial
> | group name      |   n samples |   fraction |   y mean |
> |:----------------|------------:|-----------:|---------:|
> | Black Adults    |        8508 |     0.0435 |   0.3461 |
> | Black Females   |        4353 |     0.0222 |   0.3193 |
> | Women           |       92354 |     0.472  |   0.3491 |
> | Never Married   |       68408 |     0.3496 |   0.2344 |
> | American Indian |        1294 |     0.0066 |   0.2836 |
> | Seniors         |       14476 |     0.074  |   0.541  |
> | White Women     |       55856 |     0.2855 |   0.3729 |
> | Multiracial     |        8206 |     0.0419 |   0.3572 |
> | Asian           |       32709 |     0.1672 |   0.4805 |
> | Dataset         |      195665 |     1      |   0.4106 |
>
> ### Group definitions: new
> | group name               |   n samples |   fraction |   y mean |
> |:-------------------------|------------:|-----------:|---------:|
> | associates degree male   |        7331 |     0.0375 |   0.4957 |
> | associates degree female |        8372 |     0.0428 |   0.3186 |
> | divorced female          |       10652 |     0.0544 |   0.4415 |
> | under part time          |       16525 |     0.0845 |   0.1025 |
> | part time                |       55269 |     0.2825 |   0.1408 |
> | full time                |      135989 |     0.695  |   0.5214 |
> | over full time           |       11471 |     0.0586 |   0.6441 |
> | not white                |       74659 |     0.3816 |   0.3574 |
> | government employee      |       29121 |     0.1488 |   0.5337 |
> | private employee         |      166544 |     0.8512 |   0.389  |
> | under 21                 |       10166 |     0.052  |   0.0106 |
> | middle aged              |       81582 |     0.4169 |   0.5064 |
> | Dataset                  |      195665 |     1      |   0.4106 |

---

> ### Author Response · Authors · 2024-08-12
>
> ### ACSIncome Results: initial
> | Model        | ECE                                             | Max ECE                                          | smECE                                            | Max smECE                                        | Acc                                              |
> |:-------------|:------------------------------------------------|:-------------------------------------------------|:-------------------------------------------------|:-------------------------------------------------|:-------------------------------------------------|
> | MLP ERM      | 0.01 ± 0.003                                    | 0.069 ± 0.011                                    | 0.012 ± 0.003                                    | 0.058 ± 0.005                                    | **0.812 ± 0.001** |
> | MLP HKRR     | 0.023 ± 0.001                                   | **0.065 ± 0.004** | 0.023 ± 0.001                                    | 0.063 ± 0.002                                    | 0.615 ± 0.0                                      |
> | MLP HJZ      | 0.01 ± 0.002                                    | 0.069 ± 0.008                                    | 0.013 ± 0.001                                    | **0.055 ± 0.004** | 0.81 ± 0.002                                     |
> | MLP Platt    | 0.017 ± 0.009                                   | 0.076 ± 0.008                                    | 0.018 ± 0.008                                    | 0.064 ± 0.008                                    | 0.809 ± 0.003                                    |
> | MLP Temp     | 0.011 ± 0.005                                   | 0.068 ± 0.01                                     | 0.013 ± 0.004                                    | 0.059 ± 0.007                                    | 0.811 ± 0.001                                    |
> | MLP Isotonic | **0.01 ± 0.001**  | 0.067 ± 0.008                                    | **0.011 ± 0.001**  | 0.057 ± 0.002                                    | 0.811 ± 0.001                                    |
>
> ### Results: new
> | Model        | ECE                                              | Max ECE                                          | smECE                                            | Max smECE                                        | Acc                                              |
> |:-------------|:-------------------------------------------------|:-------------------------------------------------|:-------------------------------------------------|:-------------------------------------------------|:-------------------------------------------------|
> | MLP ERM      | 0.014 ± 0.005                                    | 0.059 ± 0.007                                    | 0.015 ± 0.004                                    | 0.059 ± 0.008                                    | 0.811 ± 0.002                                    |
> | MLP HKRR     | **0.009 ± 0.003**  | **0.037 ± 0.005**  | **0.009 ± 0.003**  | **0.035 ± 0.003**  | 0.803 ± 0.001                                    |
> | MLP HJZ      | 0.019 ± 0.003                                    | 0.053 ± 0.01                                     | 0.02 ± 0.002                                     | 0.051 ± 0.011                                    | 0.807 ± 0.004                                    |
> | MLP Platt    | 0.012 ± 0.004                                    | 0.046 ± 0.008                                    | 0.014 ± 0.002                                    | 0.044 ± 0.007                                    | 0.811 ± 0.001                                    |
> | MLP Temp     | 0.012 ± 0.005                                    | 0.056 ± 0.005                                    | 0.013 ± 0.004                                    | 0.056 ± 0.005                                    | 0.811 ± 0.001                                    |
> | MLP Isotonic | 0.012 ± 0.002                                    | 0.055 ± 0.005                                    | 0.013 ± 0.001                                    | 0.054 ± 0.006                                    | **0.811 ± 0.001**  |
>
> ## Takeaways:
>
> Old groups: (Observation 1) Multicalibration does not improve upon ERM in a statistically significant way. Note here also the poor performance in accuracy of HKRR in the results included in the paper. This is because we selected the best performing hyperparameter based on worst group smECE, which does not prioritize accuracy. However, there is another choice of hyperparameters which performs slightly worse for worst group smECE on the validation (and test) set, but achieves competitive accuracy of around 0.80. Furthermore, this relates to tradeoffs discussed in Observation 3. We will note this in the next draft as well.
>
> New  groups: Multicalibration does not improve upon ERM + Platt scaling in a statistically significant way (Observation 1 and 2).

---

> > ### Author Response · Authors · 2024-08-12
> >
> > ## Bank Marketing
> > ### Group definitions: initial
> > | group name              |   n samples |   fraction |   y mean |
> > |:------------------------|------------:|-----------:|---------:|
> > | Job $=$ Management      |        9458 |     0.2092 |   0.1376 |
> > | Job $=$ Technician      |        7597 |     0.168  |   0.1106 |
> > | Job $=$ Entrepreneur    |        1487 |     0.0329 |   0.0827 |
> > | Job $=$ Blue-Collar     |        9732 |     0.2153 |   0.0727 |
> > | Job $=$ Retired         |        2264 |     0.0501 |   0.2279 |
> > | Marital $=$ Married     |       27214 |     0.6019 |   0.1012 |
> > | Marital $=$ Single      |       12790 |     0.2829 |   0.1495 |
> > | Education $=$ Primary   |        6851 |     0.1515 |   0.0863 |
> > | Education $=$ Secondary |       23202 |     0.5132 |   0.1056 |
> > | Education $=$ Tertiary  |       13301 |     0.2942 |   0.1501 |
> > | Housing $=$ Yes         |       25130 |     0.5558 |   0.077  |
> > | Housing $=$ No          |       20081 |     0.4442 |   0.167  |
> > | Age $<$ 30              |        3050 |     0.0675 |   0.1951 |
> > | 30 $\leq$ Age $<$ 40    |       17359 |     0.384  |   0.1129 |
> > | Age $\geq$ 50           |       12185 |     0.2695 |   0.1287 |
> > | Dataset                 |       45211 |     1      |   0.117  |
> >
> > ### Group definitions: new
> > | group name                      |   n samples |   fraction |   y mean |
> > |:--------------------------------|------------:|-----------:|---------:|
> > | Job $=$ Management, Age $<$ 50  |        7091 |     0.1568 |   0.1393 |
> > | Job $=$ Technician, Age $<$ 30  |         436 |     0.0096 |   0.1858 |
> > | Job $=$ Blue-Collar, Age $>$ 50 |        1075 |     0.0238 |   0.0679 |
> > | Married, Education $=$ Primary  |        5246 |     0.116  |   0.0755 |
> > | Single, Education $=$ Tertiary  |        4792 |     0.106  |   0.1836 |
> > | Housing $=$ Yes, Age $<$ 30     |        1621 |     0.0359 |   0.0993 |
> > | Housing $=$ No, Age $<$ 30      |        1429 |     0.0316 |   0.3037 |
> > | Under 21                        |         305 |     0.0067 |   0.3115 |
> > | Middle Age                      |       23841 |     0.5273 |   0.0977 |
> > | Senior Age                      |         961 |     0.0213 |   0.4225 |
> > | Dataset                         |       45211 |     1      |   0.117  |

---

> ### Author Response · Authors · 2024-08-12
>
> ### Bank Marketing Results: initial
> | Model        | ECE                                              | Max ECE                                          | smECE                                            | Max smECE                                        | Acc                                              |
> |:-------------|:-------------------------------------------------|:-------------------------------------------------|:-------------------------------------------------|:-------------------------------------------------|:-------------------------------------------------|
> | MLP ERM      | 0.009 ± 0.004                                    | 0.048 ± 0.012                                    | 0.012 ± 0.002                                    | 0.046 ± 0.01                                     |  **0.901 ± 0.002**  |
> | MLP HKRR     |  **0.007 ± 0.001**  | 0.044 ± 0.006                                    |  **0.007 ± 0.002**  |  **0.039 ± 0.003**  | 0.879 ± 0.0                                      |
> | MLP HJZ      | 0.01 ± 0.002                                     |  **0.043 ± 0.011**  | 0.013 ± 0.002                                    | 0.039 ± 0.007                                    | 0.9 ± 0.003                                      |
> | MLP Platt    | 0.01 ± 0.002                                     | 0.048 ± 0.012                                    | 0.012 ± 0.001                                    | 0.044 ± 0.01                                     | 0.899 ± 0.001                                    |
> | MLP Temp     | 0.021 ± 0.006                                    | 0.047 ± 0.005                                    | 0.022 ± 0.005                                    | 0.041 ± 0.003                                    | 0.9 ± 0.002                                      |
> | MLP Isotonic | 0.014 ± 0.003                                    | 0.044 ± 0.009                                    | 0.015 ± 0.002                                    | 0.04 ± 0.007                                     | 0.9 ± 0.0                                        |
>
> ### Results: new
> | Model        | ECE                                              | Max ECE                                          | smECE                                            | Max smECE                                        | Acc                                              |
> |:-------------|:-------------------------------------------------|:-------------------------------------------------|:-------------------------------------------------|:-------------------------------------------------|:-------------------------------------------------|
> | MLP ERM      | 0.012 ± 0.006                                    | 0.124 ± 0.023                                    | 0.014 ± 0.004                                    | 0.088 ± 0.018                                    | 0.9 ± 0.002                                      |
> | MLP HKRR     | 0.102 ± 0.003                                    | 0.127 ± 0.013                                    | 0.098 ± 0.001                                    | 0.117 ± 0.016                                    | 0.879 ± 0.0                                      |
> | MLP HJZ      | 0.009 ± 0.003                                    | 0.134 ± 0.032                                    | 0.013 ± 0.002                                    | 0.088 ± 0.014                                    | 0.899 ± 0.001                                    |
> | MLP Platt    |  **0.008 ± 0.002**  | 0.122 ± 0.021                                    |  **0.011 ± 0.001**  | 0.094 ± 0.019                                    | 0.9 ± 0.0                                        |
> | MLP Temp     | 0.042 ± 0.008                                    |  **0.116 ± 0.019**  | 0.042 ± 0.008                                    |  **0.072 ± 0.007**  | 0.899 ± 0.002                                    |
> | MLP Isotonic | 0.01 ± 0.002                                     | 0.121 ± 0.009                                    | 0.012 ± 0.002                                    | 0.085 ± 0.008                                    |  **0.901 ± 0.001**  |
>
> ## Takeaways:
>
> Old group definitions: (Observation 1 and 2) Multicalibration does not outperform Isotonic / Temperature scaling in a statistically significant way.
>
> New: (Observation 1 and 2) Multicalibration does not outperform ERM or Temperature scaling in a statistically significant way.

---

> > ### Author Response · Authors · 2024-08-12
> >
> > ## Credit Default
> > ### Group definitions: initial
> > | group name                            |   n samples |   fraction |   y mean |
> > |:--------------------------------------|------------:|-----------:|---------:|
> > | Male, Age $<$ 30                      |        3281 |     0.1094 |   0.2405 |
> > | Single                                |       15964 |     0.5321 |   0.2093 |
> > | Single, Age $>$ 30                    |        6888 |     0.2296 |   0.1992 |
> > | Female                                |       18112 |     0.6037 |   0.2078 |
> > | Married, Age $<$ 30                   |        1482 |     0.0494 |   0.2611 |
> > | Married, Age $>$ 60                   |         225 |     0.0075 |   0.2667 |
> > | Education $=$ High School             |        4917 |     0.1639 |   0.2516 |
> > | Education $=$ High School, Married    |        2861 |     0.0954 |   0.2635 |
> > | Education $=$ High School, Age $>$ 40 |        2456 |     0.0819 |   0.2577 |
> > | Education $=$ University, Age $<$ 25  |        1610 |     0.0537 |   0.2795 |
> > | Female, Education $=$ University      |        8656 |     0.2885 |   0.222  |
> > | Education $=$ Graduate School         |       10585 |     0.3528 |   0.1923 |
> > | Female, Education $=$ Graduate School |        6231 |     0.2077 |   0.1814 |
> > | Dataset                               |       30000 |     1      |   0.2212 |
> >
> > ### Group definitions: new
> > | group name                         |   n samples |   fraction |   y mean |
> > |:-----------------------------------|------------:|-----------:|---------:|
> > | Single, Male                       |        6553 |     0.2184 |   0.2266 |
> > | Single, Female                     |       15964 |     0.5321 |   0.2093 |
> > | Young Adult                        |        9618 |     0.3206 |   0.2284 |
> > | Middle Aged                        |        8872 |     0.2957 |   0.2356 |
> > | Education $=$ High School, Female  |        2927 |     0.0976 |   0.2364 |
> > | Education $=$ University, Female   |        8656 |     0.2885 |   0.222  |
> > | Education $=$ High School, Single  |        1909 |     0.0636 |   0.2368 |
> > | Education $=$ High School, Married |        2861 |     0.0954 |   0.2635 |
> > | Education $=$ University, Single   |        7020 |     0.234  |   0.2306 |
> > | Education $=$ University, Married  |        6842 |     0.2281 |   0.2435 |
> > | Education $=$ Graduate, Single     |        6809 |     0.227  |   0.1842 |
> > | Dataset                            |       30000 |     1      |   0.2212 |

---

> ### Author Response · Authors · 2024-08-12
>
> ### Credit Default Results: initial
> | Model        | ECE                                              | Max ECE                                          | smECE                                            | Max smECE                                        | Acc                                              |
> |:-------------|:-------------------------------------------------|:-------------------------------------------------|:-------------------------------------------------|:-------------------------------------------------|:-------------------------------------------------|
> | MLP ERM      | 0.018 ± 0.006                                    | 0.116 ± 0.035                                    | 0.02 ± 0.005                                     | 0.061 ± 0.005                                    | 0.819 ± 0.001                                    |
> | MLP HKRR     | 0.029 ± 0.002                                    |  **0.057 ± 0.005**  | 0.026 ± 0.001                                    |  **0.049 ± 0.005**  | 0.781 ± 0.0                                      |
> | MLP HJZ      | 0.029 ± 0.003                                    | 0.086 ± 0.003                                    | 0.028 ± 0.002                                    | 0.073 ± 0.001                                    | 0.781 ± 0.0                                      |
> | MLP Platt    | 0.028 ± 0.002                                    | 0.086 ± 0.001                                    | 0.027 ± 0.001                                    | 0.075 ± 0.002                                    | 0.781 ± 0.0                                      |
> | MLP Temp     | 0.035 ± 0.008                                    | 0.098 ± 0.015                                    | 0.035 ± 0.008                                    | 0.064 ± 0.004                                    |  **0.821 ± 0.001**  |
> | MLP Isotonic |  **0.003 ± 0.001**  | 0.059 ± 0.001                                    |  **0.003 ± 0.001**  | 0.059 ± 0.001                                    | 0.781 ± 0.0                                      |
>
> ### Results: new
> | Model        | ECE                                              | Max ECE                                          | smECE                                            | Max smECE                                        | Acc                                             |
> |:-------------|:-------------------------------------------------|:-------------------------------------------------|:-------------------------------------------------|:-------------------------------------------------|:------------------------------------------------|
> | MLP ERM      | 0.015 ± 0.005                                    | 0.04 ± 0.005                                     | 0.018 ± 0.004                                    | 0.035 ± 0.003                                    | 0.819 ± 0.002                                   |
> | MLP HKRR     | 0.028 ± 0.003                                    |  **0.026 ± 0.003**  | 0.026 ± 0.002                                    |  **0.025 ± 0.002**  | 0.781 ± 0.0                                     |
> | MLP HJZ      |  **0.012 ± 0.003**  | 0.038 ± 0.005                                    |  **0.015 ± 0.002**  | 0.033 ± 0.003                                    | 0.82 ± 0.001                                    |
> | MLP Platt    | 0.014 ± 0.005                                    | 0.045 ± 0.006                                    | 0.016 ± 0.002                                    | 0.038 ± 0.008                                    | 0.818 ± 0.001                                   |
> | MLP Temp     | 0.019 ± 0.006                                    | 0.044 ± 0.007                                    | 0.02 ± 0.005                                     | 0.037 ± 0.004                                    |  **0.82 ± 0.001**  |
> | MLP Isotonic | 0.013 ± 0.003                                    | 0.047 ± 0.008                                    | 0.015 ± 0.002                                    | 0.035 ± 0.003                                    | 0.82 ± 0.001                                    |
>
>
> ## Takeaways
> Old: (Observation 3) Statistically significant improvement of HKRR to worst group smECE over ERM and calibration. However, comes at a cost to accuracy. HJZ sees no significant improvements.
>
> New: Unchanged

---

> > ### Author Response · Authors · 2024-08-12
> >
> > ## MEPS
> > ### Group definitions: initial
> > | group name         |   n samples |   fraction |   y mean |
> > |:-------------------|------------:|-----------:|---------:|
> > | Age 0-18           |        3308 |     0.2986 |   0.0605 |
> > | Age 19-34          |        2468 |     0.2228 |   0.1021 |
> > | Age 35-50          |        2186 |     0.1973 |   0.1404 |
> > | Age 51-64          |        1813 |     0.1636 |   0.267  |
> > | Age 65-79          |         977 |     0.0882 |   0.4637 |
> > | Not White          |        7121 |     0.6427 |   0.1227 |
> > | Northeast          |        1553 |     0.1402 |   0.226  |
> > | Midwest            |        2020 |     0.1823 |   0.204  |
> > | South              |        4325 |     0.3904 |   0.1487 |
> > | West               |        3181 |     0.2871 |   0.1481 |
> > | Poverty Category 1 |        2435 |     0.2198 |   0.1577 |
> > | Poverty Category 2 |         704 |     0.0635 |   0.1378 |
> > | Poverty Category 3 |        1941 |     0.1752 |   0.1484 |
> > | Poverty Category 4 |        3100 |     0.2798 |   0.1519 |
> > | Dataset            |       11079 |     1      |   0.1694 |
> >
> > ### Group definitions: new
> > | group name          |   n samples |   fraction |   y mean |
> > |:--------------------|------------:|-----------:|---------:|
> > | Under 21            |        3772 |     0.3405 |   0.0607 |
> > | Middle Age          |        2874 |     0.2594 |   0.199  |
> > | Senior Age          |        1304 |     0.1177 |   0.4862 |
> > | Sex $=$ 1           |        5281 |     0.4767 |   0.1274 |
> > | Sex $=$ 2           |        5798 |     0.5233 |   0.2077 |
> > | White               |        3958 |     0.3573 |   0.2534 |
> > | Active Duty Group 2 |        6454 |     0.5825 |   0.1432 |
> > | Marriage Group 1    |        3645 |     0.329  |   0.2222 |
> > | Marriage Group 2    |         450 |     0.0406 |   0.4867 |
> > | Pregnancy Group 1   |         124 |     0.0112 |   0.4274 |
> > | Pregnancy Group 2   |        2167 |     0.1956 |   0.1398 |
> > | Insurance Group 1   |        5926 |     0.5349 |   0.179  |
> > | Insurance Group 2   |        3890 |     0.3511 |   0.1979 |
> > | Dataset             |       11079 |     1      |   0.1694 |

---

> ### Author Response · Authors · 2024-08-12
>
> ### MEPS Results: initial
> | Model        | ECE                                              | Max ECE                                          | smECE                                            | Max smECE                                        | Acc                                              |
> |:-------------|:-------------------------------------------------|:-------------------------------------------------|:-------------------------------------------------|:-------------------------------------------------|:-------------------------------------------------|
> | MLP ERM      | 0.022 ± 0.006                                    | 0.106 ± 0.009                                    | 0.024 ± 0.002                                    | 0.086 ± 0.015                                    | 0.864 ± 0.001                                    |
> | MLP HKRR     | 0.019 ± 0.005                                    | 0.122 ± 0.008                                    |  **0.019 ± 0.004**  | 0.104 ± 0.002                                    | 0.835 ± 0.003                                    |
> | MLP HJZ      | 0.019 ± 0.003                                    |  **0.088 ± 0.011**  | 0.021 ± 0.002                                    |  **0.076 ± 0.018**  | 0.864 ± 0.003                                    |
> | MLP Platt    |  **0.017 ± 0.005**  | 0.1 ± 0.019                                      | 0.019 ± 0.003                                    | 0.088 ± 0.02                                     | 0.865 ± 0.003                                    |
> | MLP Temp     | 0.019 ± 0.007                                    | 0.091 ± 0.016                                    | 0.02 ± 0.004                                     | 0.081 ± 0.02                                     |  **0.866 ± 0.001**  |
> | MLP Isotonic | 0.02 ± 0.006                                     | 0.108 ± 0.021                                    | 0.02 ± 0.004                                     | 0.089 ± 0.021                                    | 0.864 ± 0.003                                    |
>
> ### Results: new
> | Model        | ECE                                              | Max ECE                                         | smECE                                            | Max smECE                                        | Acc                                              |
> |:-------------|:-------------------------------------------------|:------------------------------------------------|:-------------------------------------------------|:-------------------------------------------------|:-------------------------------------------------|
> | MLP ERM      |  **0.017 ± 0.005**  | 0.3 ± 0.057                                     | 0.021 ± 0.004                                    | 0.204 ± 0.03                                     |  **0.866 ± 0.002**  |
> | MLP HKRR     | 0.019 ± 0.002                                    | 0.311 ± 0.08                                    |  **0.018 ± 0.002**  | 0.217 ± 0.043                                    | 0.84 ± 0.002                                     |
> | MLP HJZ      | 0.025 ± 0.006                                    |  **0.28 ± 0.049**  | 0.024 ± 0.003                                    | 0.208 ± 0.031                                    | 0.864 ± 0.003                                    |
> | MLP Platt    | 0.02 ± 0.004                                     | 0.314 ± 0.048                                   | 0.023 ± 0.002                                    | 0.239 ± 0.039                                    | 0.863 ± 0.002                                    |
> | MLP Temp     | 0.063 ± 0.037                                    | 0.282 ± 0.061                                   | 0.058 ± 0.032                                    |  **0.195 ± 0.038**  | 0.864 ± 0.003                                    |
> | MLP Isotonic | 0.025 ± 0.004                                    | 0.29 ± 0.047                                    | 0.025 ± 0.004                                    | 0.234 ± 0.027                                    | 0.864 ± 0.003                                    |
>
> ## Takeaways
> Old: (Observation 1) No statistically significant improvement by multicalibration algorithms.
>
> New: Unchanged.

---

### Decision · Program_Chairs · 2024-09-25

**Decision:**

Accept (poster)

**Comment:**

We thank the authors for their submission as well as their thorough and constructive engagement throughout the rebuttal period. The additional clarification the authors provided through this latter period helped address some important outstanding questions and concerns raised by the reviewers (and AC).

The reviewers point to several strengths of this work, including and importantly, that it does the hard (and sometimes not-adequately-rewarded) work of conducting an expansive evaluation for an existing method. This is an important gap in the multi-calibration literature and, as one of the reviewers pointed out, such an empirical evaluation is even more needed given recent theoretical contributions pointing to pessimistic real-world performance.

The reviewers have put considerable work into providing detailed feedback, that will significantly improve the quality of this work. As this work (we hope) may serve as a motivation for future work by the conference attendees, we strongly encourage the authors to incorporate this feedback in their revision of the paper (especially the feedback related to the experimental results and adding some discussion about the "why").